# Democratising deep learning for microscopy with ZeroCostDL4Mic

Lucas von Chamier[1,19], Romain F. Laine[1,2,19], Johanna Jukkala[3,4], Christoph Spahn [5], Daniel Krentzel [6,7], Elias Nehme[8,9], Martina Lerche[3], Sara Hernández-Pérez [3,10], Pieta K. Mattila [3,10], Eleni Karinou [11], Séamus Holden [11], Ahmet Can Solak[12], Alexander Krull [13,14,15], Tim-Oliver Buchholz[13,14], Martin L. Jones [6], Loïc A. Royer [12], Christophe Leterrier [16], Yoav Shechtman [9], Florian Jug [13,14,17], Mike Heilemann[5], Guillaume Jacquemet [3,4✉] & Ricardo Henriques [1,2,18✉]

Deep Learning (DL) methods are powerful analytical tools for microscopy and can outperform conventional image processing pipelines. Despite the enthusiasm and innovations fuelled by DL technology, the need to access powerful and compatible resources to train DL networks leads to an accessibility barrier that novice users often find difficult to overcome. Here, we present ZeroCostDL4Mic, an entry-level platform simplifying DL access by leveraging the free, cloud-based computational resources of Google Colab. ZeroCostDL4Mic allows researchers with no coding expertise to train and apply key DL networks to perform tasks including segmentation (using U-Net and StarDist), object detection (using YOLOv2), denoising (using CARE and Noise2Void), super-resolution microscopy (using Deep-STORM), and image-to-image translation (using Label-free prediction - fnet, pix2pix and CycleGAN). Importantly, we provide suitable quantitative tools for each network to evaluate model performance, allowing model optimisation. We demonstrate the application of the platform to study multiple biological processes.

[1] MRC-Laboratory for Molecular Cell Biology, University College London, London, UK. [2] The Francis Crick Institute, London, UK. [3] Turku Bioscience Centre, University of Turku and Åbo Akademi University, Turku, Finland. [4] Faculty of Science and Engineering, Cell Biology, Åbo Akademi University, Turku, Finland. [5] Institute of Physical and Theoretical Chemistry, Goethe-University Frankfurt, Frankfurt, Germany. [6] Electron Microscopy Science Technology Platform, The Francis Crick Institute, London, UK. [7] Department of Bioengineering, Imperial College London, London, UK. [8] Department of Electrical Engineering, Technion—Israel Institute of Technology, Haifa, Israel. [9] Department of Biomedical Engineering, Technion—Israel Institute of Technology, Haifa, Israel. [10] Institute of Biomedicine, and MediCity Research Laboratories, University of Turku, Turku, Finland. [11] Centre for Bacterial Cell Biology, Biosciences Institute, Faculty of Medical Sciences, Newcastle University, Newcastle, UK. [12] Chan Zuckerberg Biohub, San Francisco, CA, USA. [13] Center for Systems Biology Dresden (CSBD), Dresden, Germany. [14] Max Planck Institute for Molecular Cell Biology and Genetics, Dresden, Germany. [15] Max Planck Institute for Physics of Complex Systems, Dresden, Germany. [16] Aix Marseille Université, CNRS, INP UMR7051, NeuroCyto, Marseille, France. [17] Fondatione Human Technopole, Milano, Italy. [18] Instituto Gulbenkian de Ciência, Oeiras, Portugal. [19] These authors contributed equally: Lucas von Chamier, Romain F. Laine. ✉email: guillaume.jacquemet@abo.fi; rjhenriques@igc.gulbenkian.pt

Over the past decade, the amount and complexity of bioimages have increased exponentially, translating into a need for more complex image analysis. To combat this data challenge, new processing tools, including artificial intelligence (AI) methods, have been developed. In particular, deep learning (DL), a subset of AI that is capable of independently extracting relevant features from images to perform specific tasks, now often outperforms conventional image-processing strategies, as was demonstrated for image classification[1] or segmentation[2]. Lately, DL is increasingly employed for high-performance image-analysis tasks such as object detection[3,4], image segmentation[2,5,6] and image restoration (improvement in image resolution or denoising)[7,8]. In particular, the ability to automatically recognise objects and features in images (for instance, cancer cells in biopsy samples) is well underway to revolutionise how clinical samples are analysed (digital pathology)[4,9]. These capabilities have also led to an increased interest in DL in standard image-analysis workflows such as nuclear segmentation[5,10,11], a common task that can be a significant challenge if done manually[12]. However, the potentially game-changing capabilities of DL have, to date, remained out of reach for most researchers without a strong background in computer sciences.

A classical DL pipeline requires a computer algorithm (called "artificial neural network" or "DL network" here for simplicity) to be trained on a training dataset to generate a "model" that can perform a specific task. The training step is crucial as it will dictate the specificity and performance of the model. Generally, there are two different approaches to this step, using either supervised or self-supervised training[13]. In the supervised approach, paired input and output images are required. The network learns by attempting to find a mapping from the input to the desired output images, on an image-by-image basis. In the self-supervised approach, networks learn implicit patterns in the data and therefore, do not require paired input and output images. For instance, in a supervised image restoration task, a DL network, such as content-aware image restoration (CARE)[7], is trained using a dataset containing many examples of noisy and paired high-quality images. Once trained, the model can be used to denoise images similar to the noisy images encountered during training. Training these DL networks is computationally expensive and often requires coding and computational expertise. In many biomedical research laboratories, neither of these requirements are readily available, hindering the adoption of an ever-growing number of powerful DL methods.

To train DL networks, computer scientists typically set up local servers with high computational power or purchase expensive DL-ready workstations (Fig. 1a). This constitutes a robust way to develop and deploy DL approaches but requires technical know-how and financial commitment to set up and maintain. An alternative is to purchase computational resources provided by cloud services to train DL models. Trained DL models can then be used directly in the cloud or downloaded and used locally (Fig. 1a). Several software packages taking these approaches have been developed for the bioimaging community, especially trying to simplify the interface with DL hardware (i.e., U-Net[2,14], cDeep3m[10], DeepCell Kiosk[15], DeepMIB[16], NucleAIzer[17], YAPiC (https://yapic.github.io/yapic/), ImJoy[18], ilastik[19], CellProfiler[20] and Noise2Void[8] and DenoiSeg Fiji plugin[11,21]). Another solution is to take advantage of model "zoos" which provide trained DL models with high reusability and versatility potential (Fig. 1a). These can be used directly to obtain predictions from new data using web interfaces or ImageJ plugins (i.e., CellPose[22], DeepImageJ[23], StarDist Fiji plugin[5,6]). Using pretrained models alleviates the computational requirement of training and has therefore become popular for tasks such as cellular and nuclear

segmentation[6,17,22]. However, using pretrained models bears the risk of predicting incorrect structures and artefacts if they are used on unseen data too dissimilar to the data they were originally trained on[24,25]. This is also supported by observations that models perform best on datasets very similar to the training datasets[7,25–27] (see Supplementary Note 1 and Supplementary Fig. 1). Training will therefore often be necessary for users to achieve optimal performance of models on their own datasets. Given these considerations, there is a need for a tool that seamlessly enables users to train, validate and experiment with DL tools for various image-analysis tasks without the constraints of expensive resources and the potential drawbacks of pretrained models.

Here, we developed ZeroCostDL4Mic, an entry-level, cloud-based DL-deployment platform (Fig. 1b) that simplifies DL use for microscopy (Supplementary Movie 1). ZeroCostDL4Mic is a unified collection of self-explanatory Jupyter Notebooks, featuring an easy-to-use graphical user interface (GUI) (Supplementary Fig. 2) that requires only a web browser and a Google account for a user to run any of our DL-based tasks (Fig. 1c). All calculations are performed in the cloud using Google Colaboratory (Colab for short), circumventing the need to purchase or install graphical processing units (GPUs) and associated software. Using ZeroCostDL4Mic does not require prior knowledge in coding. Researchers can, in a few mouse clicks and aided by a common workflow, install all needed software dependencies, upload their imaging data and run networks for training and prediction (Supplementary Movie 1). Within our framework, models are comprehensively evaluated for performance and reliability and can be directly applied to new data automatically. Models can also be ported to local machines to obtain predictions for which accessible interfaces are available (DeepImageJ[23], StarDist[5,6], CARE[7]). Additionally, ZeroCostDL4Mic guides researchers on how to generate the training data necessary for DL, allowing them to develop a deeper understanding of DL methods while experimenting with parameter optimisation. We integrated a broad range of powerful DL bioimage analysis tasks within ZeroCostDL4Mic (Fig. 1c and Supplementary Movie 2). These include several supervised and self-supervised DL networks for image segmentation and object detection (using U-Net[2,14,28], StarDist[5,6] and YOLOv2[3]), image denoising and restoration (using CARE[7] and Noise2Void[8]), super-resolution microscopy (using Deep-STORM[29]) and image-to-image translations (using label-free prediction—fnet[26], pix2pix[30] and CycleGAN[31]).

## Results

**The ZeroCostDL4Mic framework**. The ZeroCostDL4Mic platform is built around the availability of cloud computing and the versatility of Jupyter Notebooks (Fig. 1b). Jupyter Notebooks can efficiently and interactively run Python code, currently the default language to deploy DL applications. For cloud computing, we focused on developing the platform around Google Colab as it provides an appropriate range of resources for free, e.g. GPU, random access memory (RAM) and disk space. These online resources provide researchers interested in DL an entrance to the field without the need to purchase and maintain a local infrastructure dedicated to DL and allow the versatility to easily run multiple networks without reconfiguring the infrastructure. To make ZeroCostDL4Mic as easy to use as possible, we exploited the Jupyter Notebooks' code readability and Colab's integration of code input via a graphical user interface (GUI). While ZeroCostDL4Mic relies on Google Colab to run, our notebooks can be adapted to run on other cloud-based platforms such as Deepnote (https://deepnote.com/) or FloydHub (https://www.floydhub.com/).

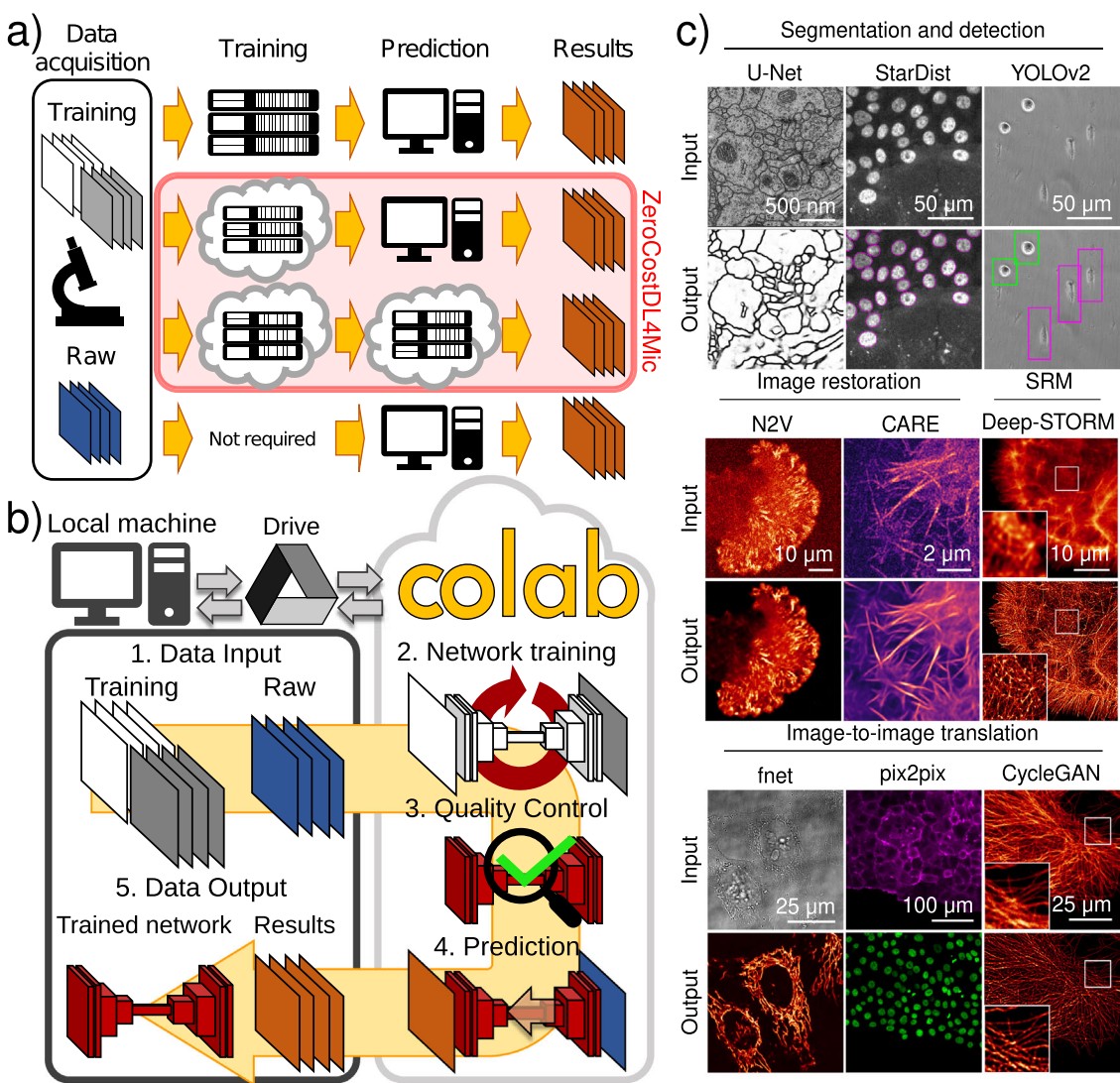

**Fig. 1 Using DL for microscopy. a** Paths to exploiting DL. Training on local servers and inference on local machines (or servers) (first row), cloud-based training and local inference (second row), cloud-based training and inference (third row) and pretrained networks on standard machines (fourth row). **b** Overview of ZeroCostDL4Mic. The workflow of ZeroCostDL4Mic, featuring data transfer through Google Drive, training, quality control and prediction via Google Colab. After running a network, trained models, quality control and prediction results can then be downloaded to the user's machine. **c** Overview of the bioimage analysis tasks currently implemented within the ZeroCostDL4Mic platform. Datasets from top left to bottom right: U-Net—ISBI 2012 Neuronal Segmentation Dataset[78,79], StarDist—nuclear marker (SiR-DNA) in DCIS.COM cells, YOLOv2—bright field in MDA-MB-231 cells, N2V—actin label (paxillin-GFP) in U-251-glioma cells, CARE—actin label Lifeact-RFP in DCIS.COM cells, Deep-STORM—actin-labelled glial cell, fnet—bright-field and mitochondrial label TOM20-Alexa Fluor 594 in HeLa cells, pix2pix—actin label Lifeact-RFP and nuclear labels in DCIS.COM cells, CycleGAN—tubulin label in U2OS cells. All datasets are available through Zenodo (see "Data availability") or as indicated in the GitHub repository.

For each provided DL network, the original network architecture is packaged in our standard workflow that every ZeroCostDL4Mic notebook follows (Supplementary Fig. 2) and which recapitulates crucial steps in DL: from data loading and training to model evaluation and inference on new data. To ensure that users with little to no coding expertise can interactively work through the pipeline, we added a textual introduction to each analytical step, explaining the basis for the procedure and providing instructions. To do this, we engaged with colleagues from the imaging and biomedical research community as beta testers. While the underlying code is hidden by default, it remains accessible, allowing users to learn, explore and edit the notebooks' programmatic structure.

A clear issue with DL is the need to validate the performance of each model on data for which the desired output is known (ground truth). With ZeroCostDL4Mic, we wanted to streamline the evaluation of model performance as much as possible, so we integrated the quality control (QC) step where quantitative metrics estimate a model's prediction accuracy by comparing it to ground-truth data. These metrics vary from one network to another to cater to the varied data output from the networks that we implemented (Supplementary Note 2). Also, to make this assessment easier, we decided to spatially map the potential discrepancies, allowing a visual way to observe artefacts linked to specific structures in the image.

In each notebook, we also included two strategies commonly used to improve the training of DL models. Firstly, we added a range of data augmentation steps, using the Augmentor[32] or imgaug (https://github.com/aleju/imgaug) python packages that can be applied to the training data to effectively increase the diversity of the training data without requiring the user to produce and provide more data (Supplementary Note 3). Secondly, we also included the possibility to perform transfer

learning[33] via the loading of a pretrained model as a starting model rather than initialising training with a blank model (Supplementary Note 4). This powerful approach allows the platform to benefit from the growing availability of pretrained models from model zoos without compromising on the quality of the performance of a model on the specific data type provided by the user. This can also have several advantages in terms of shortening training times and reducing the amount of required training data. Both of these are demonstrated and discussed in detail in a later section of this paper.

**Using Google Colab for DL for microscopy.** By using Google Colab, ZeroCostDL4Mic provides free access to the high-performance computing resources needed to run the broad range of DL networks implemented here (Fig. 1). Google Colab is widely used in the data science community for developing DL projects[34–36]. However, to productively make use of these resources, users typically need to possess expert knowledge which has drastically limited its uptake by the biomedical research community. By establishing a user-friendly and efficient interface with Google Colab, we aim to leverage this cloud-computing system to deploy state-of-the-art DL models for microscopy.

Google Colab provides access to remote virtual machines with free but finite resources which are made available for a specific runtime duration (maximum 12 h, see Supplementary Note 5 for details). The runtime duration is limited as Google Colab is intended for short, interactive use and not for long-running background computations (such as cryptocurrency mining). These resources are well suited for running simple DL pipelines. They include disk space to store training data, trained models and inference results (~68 GB), RAM to store variables and partial blocks of data (12 or 25 GB, depending on the session), and access to a high-end GPU (typically Tesla P100, T4 or K80). With these resources, training DL models in ZeroCostDL4Mic only takes a few minutes to a few hours and allows users to produce the results shown in Figs. 2–9 (see Supplementary Table 1 for details of the dataset used and Supplementary Table 2 for training parameters and times). Installing the necessary libraries takes less than a minute. Training times vary depending on the network used and the assigned GPU (see Supplementary Table 3 for install times and training speeds), but providing a speed improvement ranging between 5× and 200× over central processing unit (CPU) computation time. Once a model is trained, and its performance validated, it can be applied to new data (see Supplementary Table 4 for inference speeds).

Resources are typically sufficient to train models on real biological data and to provide the results presented in this paper. We do, however, also discuss strategies to overcome potential limitations (Supplementary Note 5). In particular, training a model with Colab may impose an upper limit on the number of images that can be used during training (typically dictated by the availability of sufficient RAM, see Supplementary Table 5 for breaking points on training capabilities within Google Colab, and Supplementary Table 6 for inference throughput). Again, these limitations will strongly depend on the network and the size of the provided images. When generating predictions, users can easily and quickly batch process thousands of images, limited by the available storage on Google Drive (15 GB for a free account). Considering the resources available with Colab, we believe that ZeroCostDL4Mic is well suited for:

1. Prototyping image-analysis workflows and pipelines without financial investment.
2. Executing small-to-medium-size projects (a few 10's of GB of data) compared to large-scale projects often encountered in machine vision research.
3. Short-term projects not requiring a permanent investment in DL infrastructure.
4. As a resource for DL enthusiasts and students to learn about DL methods and state-of-the-art architectures, such as U-Net[2,28] or (generative adversarial networks) GANs[30,31].

However, larger-scale (>20 GB of data) and longer-term analysis pipelines may benefit from the investment in paid-for cloud-based platforms (like Paperspace (https://www.paperspace.com/), Amazon Web Services (AWS) (https://aws.amazon.com/) or FloydHub) or local infrastructure, therefore tuning the resources to the needs of the specific in-house application.

In addition, ZeroCostDL4Mic is adjustable to run outside Google Colab (see Supplementary Note 6 and Supplementary Fig. 4 for running ZeroCostDL4Mic notebooks within Deepnote and FloydHub).

Within ZeroCostDL4Mic, we implemented several DL-based image-analysis tools, including networks for image segmentation and object detection, image denoising and restoration, super-resolution microscopy and image-to-image translations. In the following sections, we introduce each of the tasks and DL networks that are currently available on our platform, showcasing results obtained by using Colab.

**Image segmentation and object detection.** Manual segmentation is a time-consuming and challenging task that typically requires expert knowledge and can be a bottleneck for studies that aim to quantify large datasets[12,37]. Hence, DL tools are of great interest in this field, as they can combine expert-level performance with high-throughput analysis[4]. Within ZeroCostDL4Mic, we implemented two networks, U-Net[2,28] and StarDist[5,6], which perform state-of-the-art image segmentation, both handling 2D and 3D imaging datasets. To illustrate the applicability of our ZeroCostDL4Mic U-Net 2D and 3D notebooks, we trained U-Net networks to segment membranes from 2D electron microscopy (EM) images and mito-chondria from 3D EM images (Fig. 2a, b and Supplementary Movie 3). To showcase our StarDist notebooks, we first trained a StarDist model to segment the nuclei of densely packed cell monolayers (Fig. 2c). Interestingly, the trained model could detect extra nuclei that were missed when these images were manually labelled (Fig. 2c). In addition, to facilitate the analysis of live-cell imaging data, the outputs generated by the StarDist notebook are directly compatible with the popular tracking software TrackMate which consequently enables automated cell tracking (Fig. 3d and Supplementary Movie 4)[38].

Object detection tasks have become of interest in microscopy studies as they allow the identification of multiple classes/objects in an image, e.g., counting pathogens or identifying cell types in bioimages[9,39]. To provide object detection capabilities within ZeroCostDL4Mic, we implemented the popular DL network YOLOv2[3]. To demonstrate how our YOLOv2 implementation is applicable to the analysis of microscopy data, we annotated time-lapses of cells migrating on cell-derived matrices in the function of the shape taken by the cells as they migrate ("elongated" cells, "rounded" cells, "dividing" cells and "spread-out" cells) (Fig. 3 and Supplementary Movie 5). The YOLOv2 model, trained using ZeroCostDL4Mic, correctly classified and labelled a large fraction of the cells in the images (as indicated by the mAP score).

**Image restoration and denoising.** Fluorescence live-cell imaging has become the primary strategy to directly and dynamically observe pathways with high molecular specificity. However, when performing live-cell imaging experiments, low laser intensities need to be combined with low expression of the molecule(s) of interest to ensure the physiological relevance of the phenomenon observed[40]. Therefore, live-cell imaging experiments often lead to

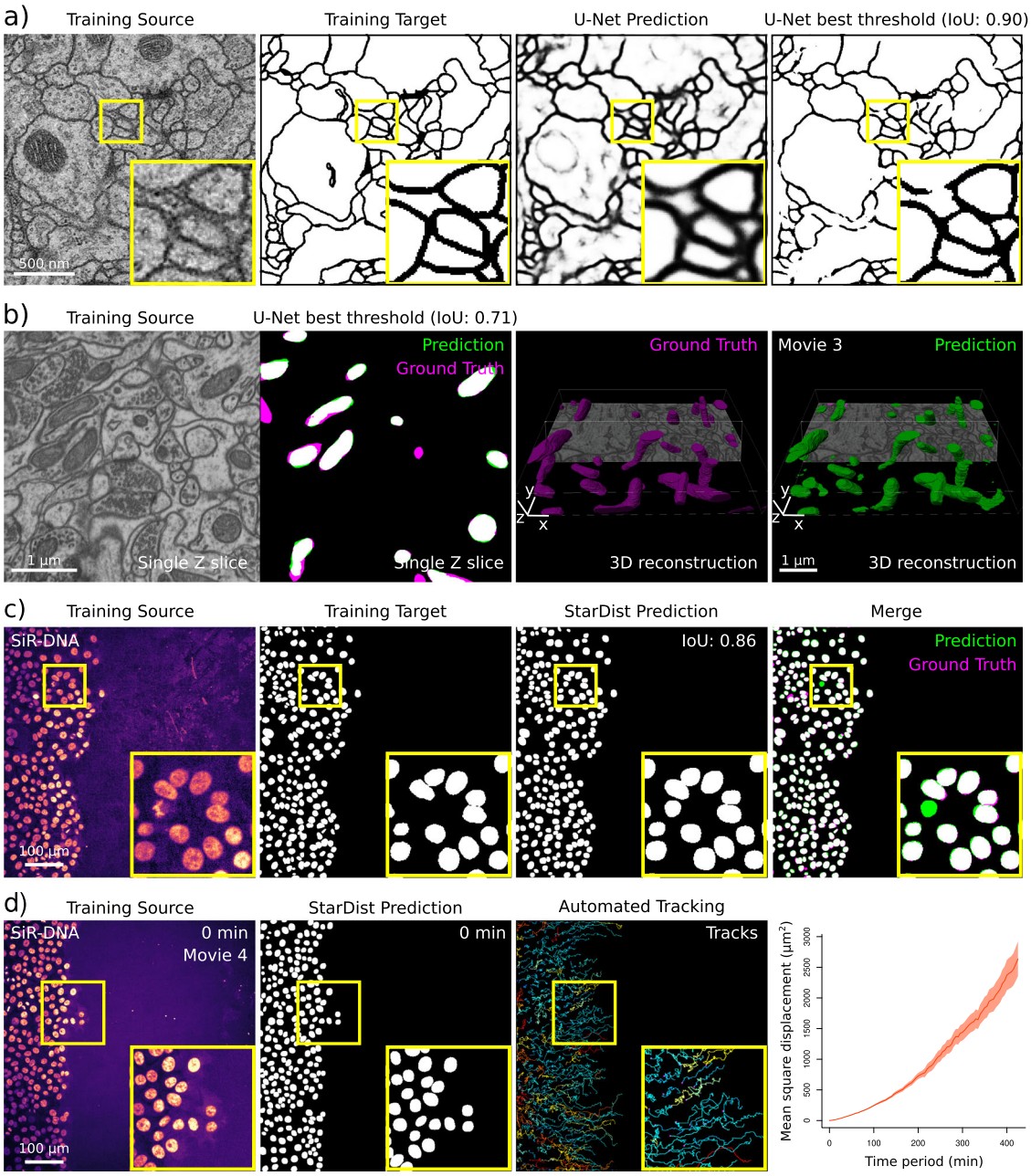

**Fig. 2 Image-segmentation networks (U-Net and StarDist). a**, **b** Example of data generated using the ZeroCostDL4Mic U-Net and StarDist notebooks.
**a** A 2D U-Net model was trained to segment neuronal membranes from EM images. This training dataset is from the 2012 ISBI segmentation challenge[78].
Training source (raw data), training targets (hand-annotated binary masks), predictions (raw output of the notebook after training) and U-Net image
thresholded output are displayed, achieving an Intersection over Union (IoU) of 0.90 (see Supplementary Note 2 for details). The optimal threshold was
assessed automatically using the Quality Control section of the notebook (see Supplementary Note 3). **b** A 3D U-Net network was trained to segment
mitochondria from EM images. The training dataset was made available by EPFL and consists of EM images of $5 \times 5 \times 5 \, \mu m^3$ sections taken from the CA1
hippocampus region of the brain. A representative single Z slice, as well as an overlay displaying U-Net prediction and the ground truth, are displayed. 3D
reconstructions displayed were performed from U-Net predictions using Imaris (Supplementary Movie 3). **c**, **d** Example of data generated using the
ZeroCostDL4Mic StarDist notebooks. **c**, **d** A StarDist model was trained (**c**) to automatically detect nuclei in movies of migrating DCIS.COM cells, labelled
with SiR-DNA, to track their movement automatically (**d**). **c** Example of Training source (DCIS.COM cells labelled with SiR-DNA), Training targets
(Ground-truth masks) and StarDist prediction (IoU of 0.86) are displayed. **d** StarDist outputs were used to automatically track cell movement over time in
TrackMate (Supplementary Movie 4). Cell tracks were further analysed using the online platform motilitylab.net, indicating a directed movement that is
expected for such migration assays (error bars represent the standard deviation). IoU Intersection over Union.

the acquisition of noisy images, and denoising strategies are
becoming increasingly essential for the interpretation of data and
its further analysis.

Content-aware image restoration (CARE)[7] can denoise and
improve the resolution of 2D and 3D images, using supervised

training. To illustrate the applicability of our ZeroCostDL4Mic
CARE notebooks, we trained a 3D CARE network to denoise
live-cell structured illumination microscopy (SIM) imaging
data (Fig. 4a, b and Supplementary Movie 6). To generate
a suitable training dataset, both high and low SNR images of

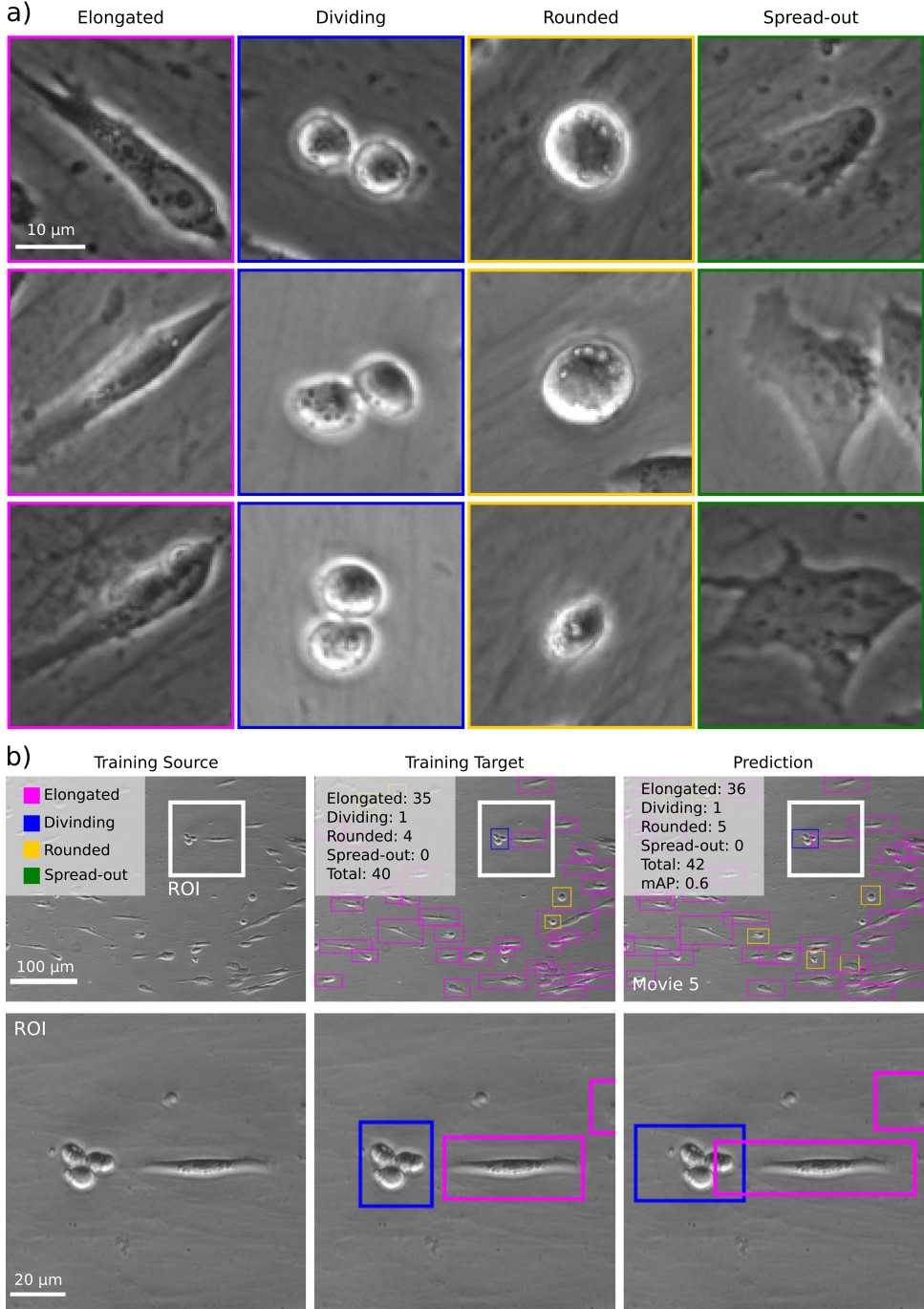

**Fig. 3 Object detection (YOLOv2).** Example of data generated using the ZeroCostDL4Mic YOLOv2 notebook, detecting and identifying cell shape classification from a cell migration bright-field time-lapse dataset. **a** Identified cell shapes and representative examples that were hand-labelled in the training dataset. **b** Input, ground truth and prediction obtained from object detection, highlighting the identification of the presence of three classes in the field-of-view (see also Supplementary Movie 5) and an mAP of 0.60 for this field of view. mAP: mean average precision (see Supplementary Note 2 for details). mAP mean average precision.

the actin cytoskeleton were acquired from fixed samples. The network trained using these images was then used to restore live-cell imaging data (Fig. 4a, b and Supplementary Movie 6). The approach employed here is especially useful as it can be very challenging to obtain high-quality live-cell imaging by other means while using SIM for extended periods of time[41,42].

Noise2Void[8] is a DL method capable of denoising microscopy images using self-supervised learning, therefore in the absence of a dedicated paired training dataset (Supplementary Movies 7–9). This has the advantage that the network can be trained directly on the data that needs processing and therefore makes this approach very easy and powerful to use. We first demonstrated the capabilities of our Noise2Void notebook to denoise the movie of an ovarian carcinoma cell migrating on cell-derived matrices (Supplementary Movie 7). Here, a single Z stack (time point) is used to train Noise2Void, and the resulting model is applied to the rest of the movie (Fig. 4c and

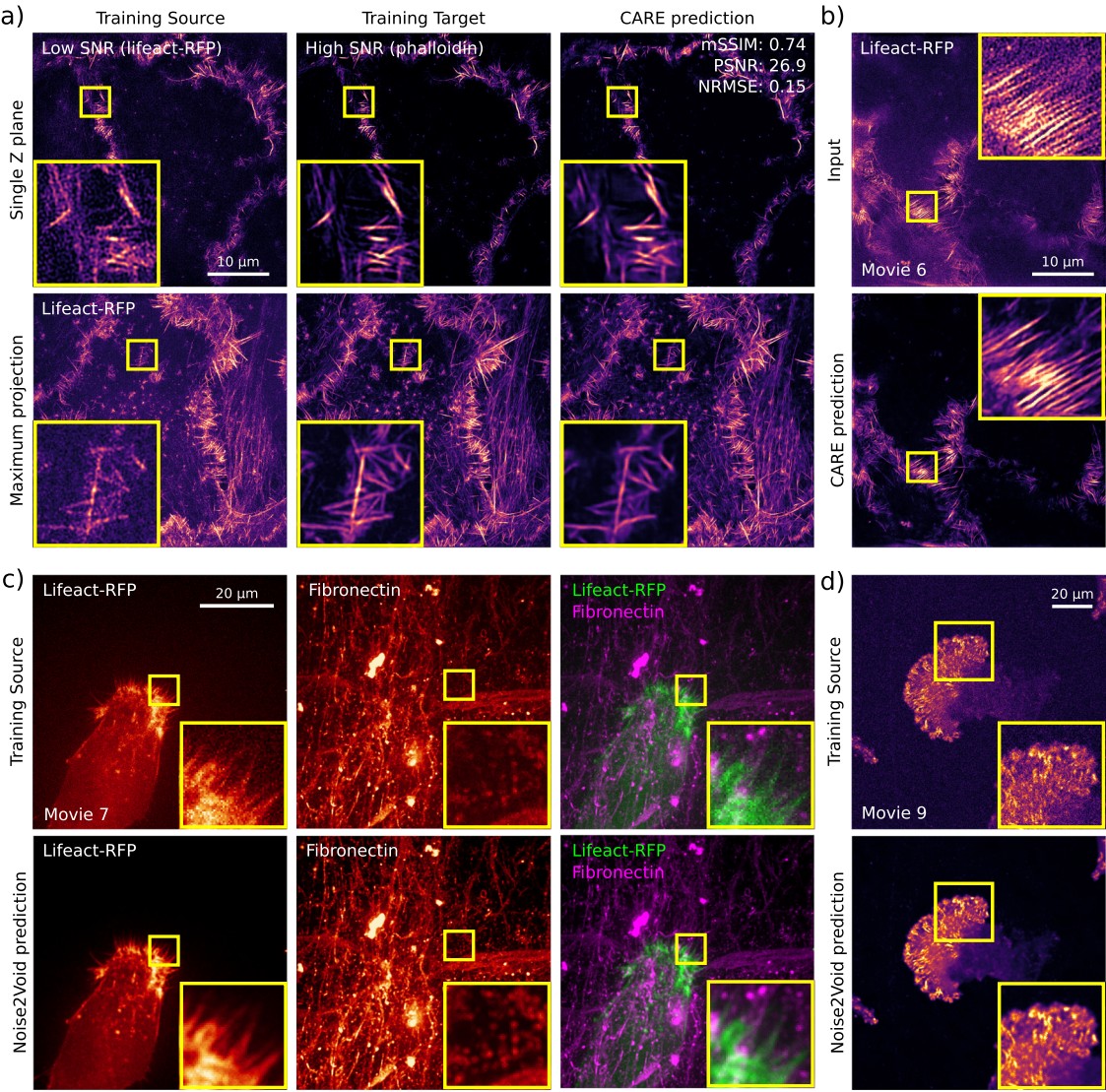

**Fig. 4 Image denoising and restoration networks (CARE and Noise2Void).** Example of data generated using ZeroCostDL4Mic CARE and Noise2Void notebooks. **a, b** A 3D CARE network was trained using SIM images of the actin cytoskeleton of DCIS.COM cells using fixed samples (**a**) to denoise live-cell imaging data (**b**). Quality control metrics are as follows: mSSIM: 0.74, PSNR: 26.9 and NRMSE: 0.15. **c** Fixed samples were imaged using SIM to obtain low signal-to-noise images (lifeact-RFP, Training Source) and matching high signal-to-noise (Phalloidin staining, Training Target) images, and this paired dataset was used to train CARE. Input, ground truth and a CARE prediction are displayed (both single $Z$ plane and maximal projections). The QC metrics values computed directly in the CARE notebook are indicated. **b** The network trained in (**a**) was then used to restore live-cell imaging data (Supplementary Movie 6). The low SNR image (input) and the associated CARE predictions are displayed (single plane). **c** Movie of an ovarian carcinoma cell labelled with lifeact-RFP migrating on cell-derived matrices (labelled for fibronectin) denoised using Noise2Void. Both training source and Noise2Void predictions are displayed (Supplementary Movie 7). For each channel, a single $Z$ stack (time point) was used to train noise2Void, and the resulting model was applied to the rest of the movie. **d** Movie of a glioma cell endogenously labelled for paxillin-GFP, migrating on 9.6 kPa polyacrylamide hydrogel, and imaged using an SDC. Both training source and Noise2Void prediction are displayed (Supplementary Movie 9). A single image (time point) was used to train Noise2Void, and the resulting model was applied to the rest of the movie. For all panels, yellow squares highlight a region of interest that is magnified.

Supplementary Movie 7). Next, we used the ZeroCostDL4Mic notebook to denoise data capturing the endogenous expression levels of a glioma cell endogenously labelled for paxillin-GFP, migrating on polyacrylamide hydrogel (Fig. 4d and Supplementary Movie 9).

In all the displayed examples, the Noise2Void models trained in ZeroCostDL4Mic performed very well and significantly improved the quality of the images. However, it is important to note that, in our hands, Noise2Void did not perform well when used to denoise the actin dataset used to train CARE (Fig. 4a). This is likely due to the noise in this particular dataset not being homogeneous and containing imprinted patterns from the SIM

reconstruction process. This highlights the importance of testing a range of networks to identify which approach is most suited for a specific dataset, underlining the need for a platform integrating a broad range of networks.

**Super-resolution microscopy.** Over the last two decades, super-resolution microscopy (SRM) has enabled cellular structures to be observed down to the nanoscale[43]. SRM methods often require the post-processing analysis of images, which can be aided by DL strategies[29,44,45]. Within ZeroCostDL4Mic, we implemented Deep-STORM[29], a DL network capable of reconstructing single-molecule localisation microscopy (SMLM) data from dense

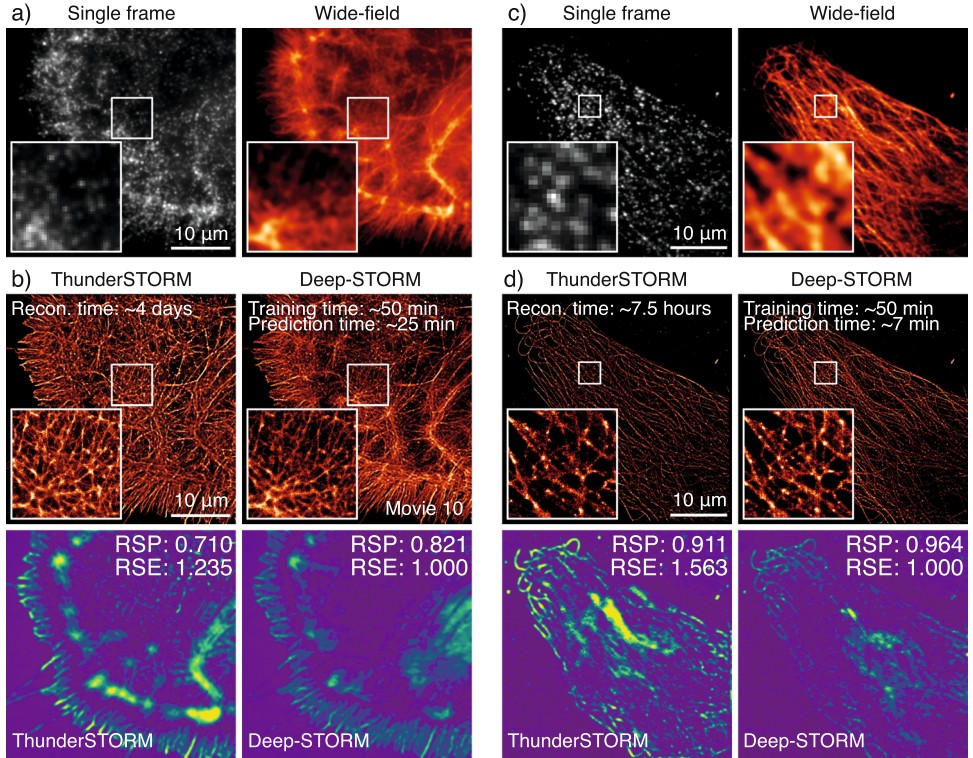

**Fig. 5 Super-resolution microscopy network (Deep-STORM).** Example of data that can be generated using the ZeroCostDL4Mic Deep-STORM notebook. **a** Single frame of the raw BIN10 dataset, phalloidin labelling of a glial cell, and the wide-field image. **b** Top: Comparison of ThunderSTORM[51] Multi-Emitter Maximum likelihood estimation (ME-MLE) and Deep-STORM reconstructions (see also Supplementary Movie 10). ME-MLE processing times were estimated using an Intel Core i7-8700 CPU @ 3.2 GHz, 64GB RAM machine. Bottom: SQUIRREL[52] analysis comparing reconstructions from ThunderSTORM ME-MLE and Deep-STORM, highlighting better linearity of the reconstruction with respect to the equivalent wide-field dataset for Deep-STORM. **c** Single frame of the raw of a DNA-PAINT[50] dataset of a U2OS cell immuno-labelled for tubulin and the wide-field image. **d** Top: Comparison of ThunderSTORM[51] Multi-Emitter Maximum likelihood estimation (ME-MLE) and Deep-STORM reconstructions. ME-MLE processing times were estimated using an Intel Core i7-8700 CPU @ 3.2 GHz, 32 GB RAM. Bottom: SQUIRREL[52] analysis comparing reconstructions of ThunderSTORM ME-MLE and Deep-STORM, highlighting better linearity of the reconstruction with respect to the equivalent wide-field dataset for Deep-STORM. The reconstruction times shown for Deep-STORM were obtained with the NVIDIA Tesla P100 PCIe 16 GB RAM available on Google Colab. RSP resolution -scaled Pearson coefficient, RSE root-squared error.

emitter datasets. Performing high-density SMLM reconstruction has the advantage of achieving high-performance SMLM imaging in poorly blinking conditions and offers the possibility of significantly shortening the image acquisitions.

Training Deep-STORM requires raw SMLM data accompanied by ground-truth localisation coordinates. Importantly, Deep-STORM can be trained using simulated data. For this purpose, we included a simulator within the ZeroCostDL4Mic notebook to directly generate SMLM training (and test) data that can be tuned to mimic the experimental data type that needs to be subsequently analysed. Unlike the original Deep-STORM network, our notebook also allows us to extract the localisation coordinates from the reconstructed image, enabling further analysis such as drift correction (available within the notebook) and spatial point pattern analysis[46–48].

To illustrate the capabilities of our ZeroCostDL4Mic Deep-STORM notebook, we reconstructed the image of a glial cell, labelled for actin and imaged using *d*STORM[49] (Fig. 5a, b and Supplementary Movie 10), and the image of a cancer cell stained for tubulin and imaged using DNA-PAINT[50] (Fig. 5c, d). For comparison, the same data were processed using a Multi-Emitter Maximum Likelihood Estimation (ME-MLE) implemented in ThunderSTORM[51]. Of note, Deep-STORM led to high-quality reconstruction in a fraction of the time necessary for ME-MLE to produce the reconstructed image.

In addition, a SQUIRREL[52] analysis of these reconstructions showed that Deep-STORM had better agreement with the equivalent wide-field image, highlighting better linearity in the reconstructions when compared to reconstructions based on ME-MLE localisation (Fig. 5b, d).

**Image-to-image translation.** Image-to-image translation refers to a transformation from one type of image into another, such as by predicting a fluorescent label from bright-field images or by predicting a fluorescent label from another fluorescent label. Within ZeroCostDL4Mic, we implemented three networks, label-free prediction (fnet)[26], pix2pix[30] and CycleGAN[31] capable of performing image-to-image translations. Although label-free prediction (fnet) and pix2pix are based on supervised learning, CycleGAN trains in a self-supervised manner without the need for paired ground-truth data.

Therefore, performing image-to-image translation using fnet and pix2pix requires the user to provide a paired dataset for training (Fig. 6a). The fnet[26] network was developed to perform label-free predictions from bright-field and EM images. It is based on a U-Net architecture and training requires paired 3D stacks of two channels (e.g. fluorescence and bright-field images). To showcase the ZeroCostDL4Mic fnet notebook, we trained a fnet model to predict TOM20 mitochondrial labelling from bright-field images (Fig. 6b). In contrast, pix2pix[30] uses GANs[53] to

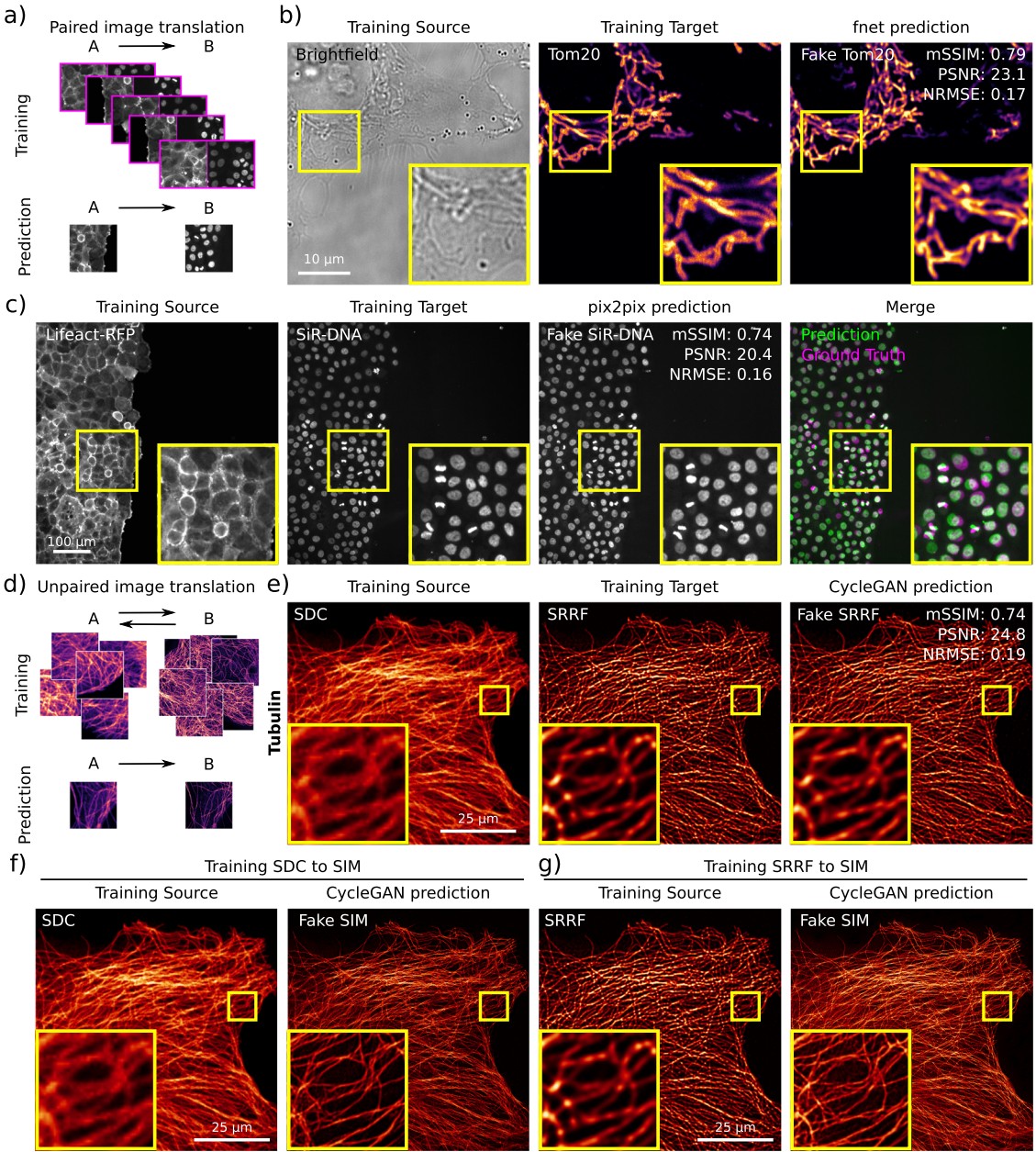

**Fig. 6 Image-to-image translation networks (fnet, pix2pix and CycleGAN).** Example of data generated using the ZeroCostDL4Mic fnet, pix2pix and CycleGAN notebooks. **a** Scheme illustrating the data required to train paired image-to-image translation networks (pix2pix and fnet). **b** Fnet was trained to predict the location of mitochondria (Tom20 staining, Training Target) from bright-field images (Training Source). Both the fnet prediction and the ground-truth images are displayed. The quality control metrics values computed directly in the fnet notebook are as follows: mSSIM (mean structural similarity index): 0.79, PSNR (peak signal-to-noise ratio): 23.1 and NRMSE (normalised root-mean-squared error): 0.17. **c** pix2pix was trained to predict nuclear stainings (SiR-DNA, Training Target) from actin stainings (lifeact-RFP, Training Source) in migrating DCIS.COM cells. A pix2pix prediction, the corresponding ground-truth images are displayed. The quality control metrics values computed directly in the pix2pix notebook are as follows: mSSIM: 0.74, PSNR: 20.4 and NRMSE: 0.16. **d** Scheme illustrating the data requirement to train unpaired image-to-image translation networks (CycleGAN). Importantly, these networks do not need to have access to a paired training dataset. **e, f** CycleGAN was trained to predict what images of microtubules acquired with an SDC (spinning-disk confocal) would look like when processed with SRRF (super-resolution radial fluctuations) (**e**) (quality control metrics values are as follows: mSSIM: 0.74, PSNR: 24.8 and NRMSE: 0.19) or imaged with a SIM (structured illumination microscopy) microscope (**f**). A CycleGAN model was also trained to transform SRRF images into SIM images (**g**). For the SDC to SRRF translation, the CycleGAN prediction and ground-truth SRRF images are displayed as well as the QC metrics values computed directly in the pix2pix notebook are displayed. For all panels, yellow squares highlight a region of interest that is magnified.

translate one type of image into another and training pix2pix requires paired 2D images. Here, we illustrate a possible use of pix2pix by training it to convert the fluorescence image of one label (actin) into that of another label (nucleus) in migrating DCIS.COM cells (Fig. 6c).

Image-to-image translation networks such as CycleGAN[31] can capture the characteristics of one image domain and understand how these characteristics are related to another image domain, all in the absence of any paired training examples (Fig. 6d). Therefore, training CycleGAN only requires a representative set

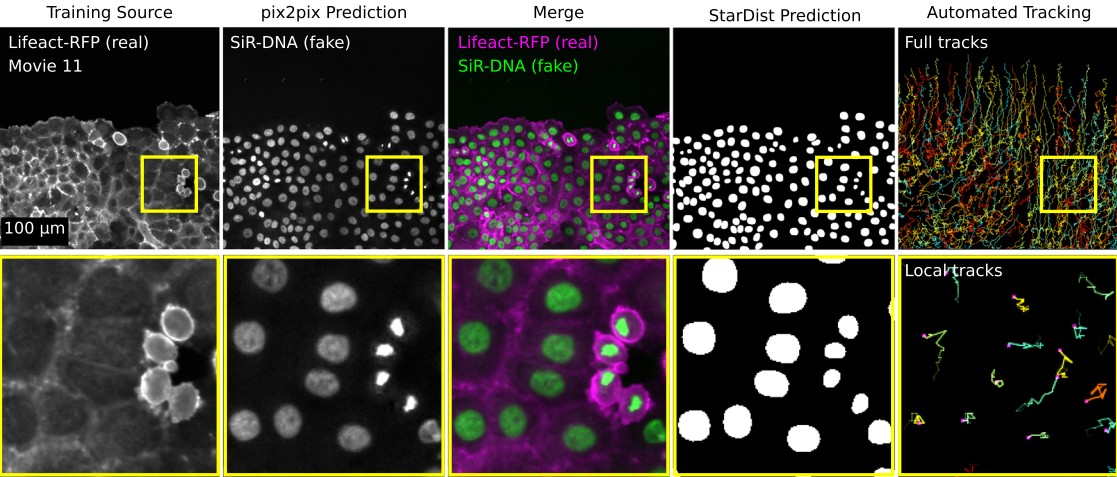

**Fig. 7 Example illustrating how ZeroCostDL4Mic notebook can be used together.** Figure highlighting how ZeroCostDL4Mic notebooks can be combined to create a data analysis pipeline. Here, we wanted to automatically track the migration pattern of DCIS.COM cells labelled with lifeact-RFP. Therefore, we first used pix2pix to predict the actin staining into nuclei staining (as in Fig. 6c) and StarDist to detect the nuclei. From the StarDist prediction, cells were tracked automatically using TrackMate[38] (as in Fig. 2d; see also Supplementary Movie 11). A representative field-of-view is displayed.

of images from one domain and a set from the other without the need for any particular correspondence and, due to this, it provides unique flexibility in training. As for pix2pix, CycleGAN uses GANs to perform the image-to-image translation task and requires 2D images to train. Here, we use CycleGAN to predict what a fluorescent label would look like when imaged using other imaging modalities. In particular, we trained CycleGAN to predict what images of microtubules acquired with a spinning-disk confocal would look like when processed with SRRF[54,55] or imaged with a SIM microscope (Fig. 6e, f). Using our notebook, we also trained CycleGAN to transform SRRF images into SIM images (Fig. 6g).

**ZeroCostDL4Mic within larger analysis pipelines**. ZeroCostDL4Mic notebooks are self-contained as they are sufficient to train, evaluate and use DL networks. However, they can also be connected with other image-analysis tools and predictions generated within the notebooks can be further analysed elsewhere. For instance, the StarDist notebook can easily be connected to TrackMate[38] to enable automated cell tracking[56] (Fig. 2). In fact, most models trained with ZeroCostDL4Mic can also be downloaded and used outside of ZeroCostDL4Mic (e.g., StarDist[5,6], CARE[7], Noise2Void[8] in Fiji[21] and U-Net[2,28] and Deep-STORM[29] in DeepImageJ[23]) and we expect cross-platform compatibilities to improve in the future. This capability allows users to easily benefit from the large pre-existing image-analysis ecosystem around ImageJ/Fiji[21,57]. For segmentation tasks, this capability also allows users to annotate more training images by using model predictions as a starting point[56].

ZeroCostDL4Mic notebooks can also be easily used sequentially. For instance, we combined image-to-image translation and nuclear segmentation tasks to track cells based on an actin label automatically (Fig. 7 and Supplementary Movie 11). Several groups have recently published tools that fulfil similar cross-modality tasks[58,59], highlighting a need for such tools in the bioimaging community. Here, we demonstrate that ZeroCostDL4Mic notebooks can be used in a modular fashion to quickly and easily recapitulate sought-after solutions for the community.

**Quality control**. One key feature of the ZeroCostDL4Mic notebooks is that they allow for a detailed quality assessment of the trained models before they are deployed on unseen data (Supplementary Note 2). This is essential in optimising the network performance for a particular application, determining its limitations and preventing the significant introduction of artefacts, a commonly raised concern for DL applications in microscopy[60,61]. With this in mind, we implemented a quantitative QC step in all notebooks, which allows the assessment and improvement of model performance (Supplementary Figs. 5 and 6). These metrics allow the user to improve the performance of a given model by tuning its hyperparameters or exploring the range of applicability to different data from which it was trained (generalisation). The QC section typically has two parts. In the first part, the performance metrics shown to users are loss curves for model training and validation (Fig. 8a). These allow users to determine if the tested model overfits during training, identifiable by an increasing divergence between the validation and training loss. This divergence appears if the model learns features too specific to the training dataset instead of general features applicable to all similar datasets, therefore preventing it from generalising to unseen data, a common problem for DL networks[62].

However, even a model that performs well on the validation data can produce unwanted results when used on unseen data, ultimately making the model unreliable. In the second part of the notebooks' QC section, these issues can be detected by comparing the predictions from unseen data to the equivalent ground-truth data. The metrics used in individual notebooks to quantify the differences between predictions and ground truth vary to reflect the differences in the type of data these models operate on. For networks producing a grey-scale image, e.g. CARE, Noise2Void, pix2pix, CycleGAN and label-free prediction (fnet), the metrics used are SSIM (structural similarity)[63] and RSE (root square error) (Fig. 8b). For networks producing a binary or semantic segmentation, e.g., U-Net, the metric used is IoU (intersection over union) (Fig. 8c). The StarDist notebook also makes use of the IoU as well as other metrics, including the F1 score and the Panoptic quality[64]. The YOLOv2 notebook uses the mean average precision score (mAP)[65,66] and F1 score, reflecting the validity of the bounding box positions and the corresponding classification (Fig. 8d). These metrics and our implementations are described in detail in Supplementary Information (Supplementary Note 2).

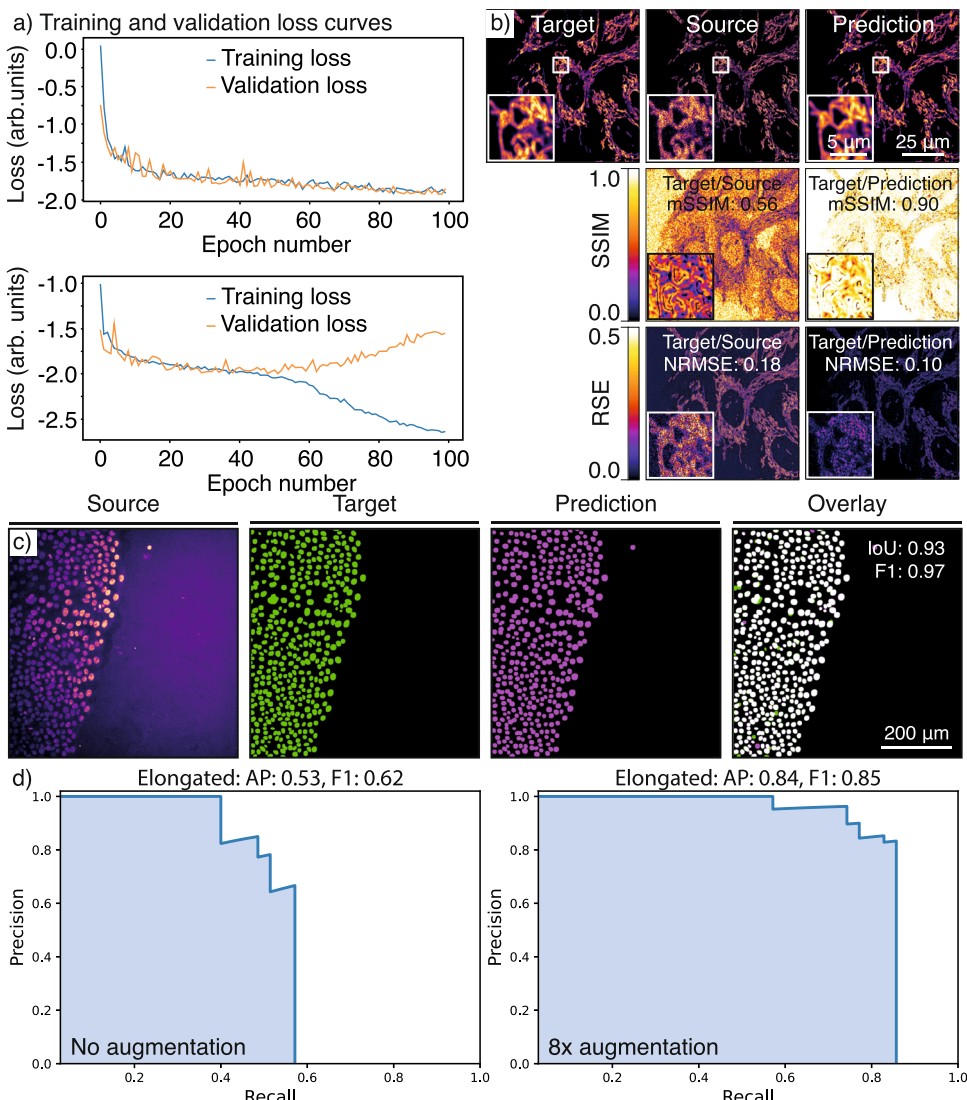

**Fig. 8 Quality control of trained models. a** Overfitting models: Graphs showing training loss and validation loss curves of a CARE network with different hyperparameters. The upper panel shows a good fit of the model to unseen (validation) data (main training parameters, number_of_epochs: 100, patch_size: 256, number_of_patches: 10, Use_Default_Advanced_Parameters: enabled), the lower panel shows an example of a model that overfits the training dataset (main training parameters, number_of_epochs: 100, patch_size: 80, number_of_patches: 200, Use_Default_Advanced_Parameters: enabled). **b** RSE (root-squared error) and SSIM (structural similarity index) maps: An example of quality control for CARE denoising model performance. The quality control metrics values computed directly in the notebook are as follows: mSSIM (mean structural similarity index): 0.56 and NRMSE (normalised root-mean-squared error): 0.18 for target vs source and mSSIM: 0.90 and NRMSE: 0.10 for target vs prediction. **c** IoU (intersection over union) maps: An example of quality control metrics for a StarDist segmentation result, where IoU: 0.93 and F1: 0.97. **d** Precision–recall (p–r) curves: p–r curves for the dataset shown in Supplementary Fig. 13, highlighting the effect of augmentation on the performance metrics of the YOLOv2 model, where AP (average precision) for elongated improved from 0.53 to 0.84 upon 8× augmentation while F1 improves from 0.62 to 0.85.

**Data augmentation and transfer learning**. In ZeroCostDL4Mic, we also enabled several important functionalities that facilitate and improve the applicability of each DL approach. First, we implemented the possibility to enable data augmentation (Augmentor[32], imgaug) which can artificially expand the image diversity of a dataset. It commonly consists of applying a set of image transformations to both source and target data in the training dataset, such as rotation or vertical/horizontal flipping but more complex transformations can also be used, such as shearing. For instance, merely flipping (horizontal and vertical) and rotating all the images in a dataset by 90° will increase the size of a dataset by 8. Data augmentation may improve the generalisation of a model by amplifying diversity in the dataset. This may be especially useful if the available dataset is small,

which can occur if it is expensive or time-consuming to generate. For instance, when training YOLOv2, with our test dataset, we found that the model performance improved when performing data augmentation on the training images and bounding boxes (Fig. 9a–c).

In addition, we included the option to perform transfer learning[33] in all the ZeroCostDL4Mic notebooks. Transfer learning allows us to take advantage of pretrained models, from model "zoos" for instance, by re-using previously learned features within these models and speeding up and improving the training process (Supplementary Note 4). To illustrate the performance improvement that can be achieved using transfer learning, we compared the results obtained by training a StarDist model from scratch to the results obtained when re-training a readily available

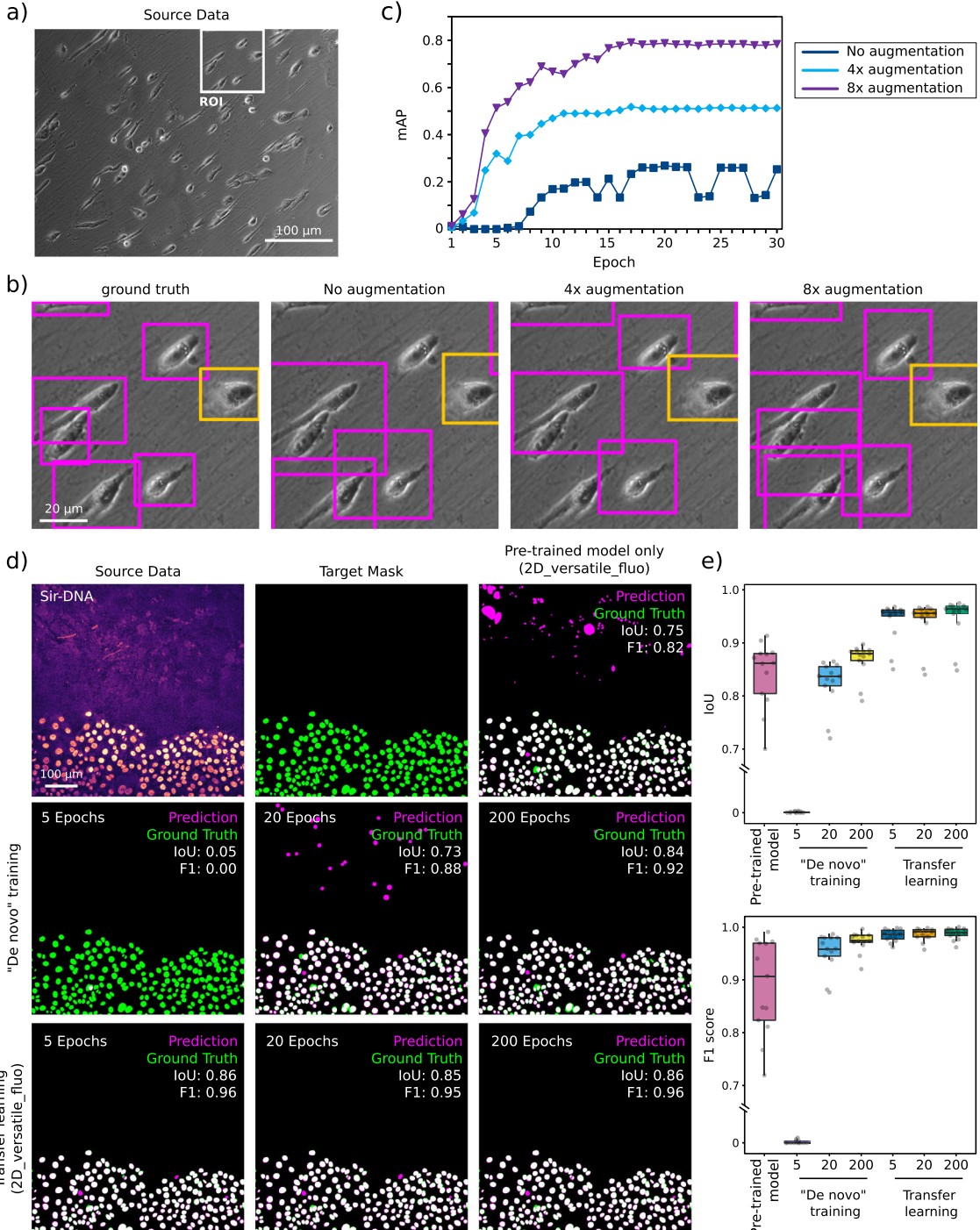

**Fig. 9 Data augmentation and Transfer learning can improve performance. a–c** Data augmentation can improve prediction performance. YOLOv2 cell shape detection applied to bright-field time-lapse dataset. **a** Raw bright-field input image. **b** Ground-truth and YOLOv2 model predictions (after 30 epochs) with increasing amounts of data augmentation. The original dataset contained 30 images which were first augmented by vertical and horizontal mirroring and then by 90° rotations. **c** mAP (mean average precision) as a function of epoch number for different levels of data augmentation. **d, e** These panels display an example of how transfer learning using a pretrained model can lead to very high-quality StarDist prediction even after only 5 epochs. This figure also highlights that using a pretrained model, even when trained on a large dataset, can lead to inappropriate results. **d** Examples of StarDist segmentation results obtained using models trained using 5, 20 or 200 epochs and using a blank model ("De novo" training) or the 2D-versatile-fluo as a starting point (transfer learning). **e** StarDist QC metrics obtained with the models highlighted in (**d**) ($n = 13$ images). The IoU (intersection over union) scores are calculated over the whole image, while the F1 scores are calculated on a per-object basis. Results are displayed as boxplots which represent the median and the 25th and 75th percentiles (interquartile range); outliers are represented by dots[96]. Note that the axes of both graphs are cut. Source data for panel (**c**) and (**e**) are provided in the Source Data file.

model (model trained by others using related but different images). When the provided pretrained model is used to perform prediction on our data, it led to a reasonable nuclear segmentation but also to the generation of large artefacts rendering it unusable (Fig. 9d, e). Typically, to obtain high-quality predictions from models trained from scratch, we needed to train our StarDist models for more than 200 epochs. However, when using transfer learning, very high-quality predictions can be made using a model trained for as little as five epochs (Fig. 9d, e).

## Discussion

By bringing previously validated methods into a streamlined format that allows easy, cost-free access and customised DL use for microscopy data, we believe that ZeroCostDL4Mic provides an important step towards broadening the use of DL approaches beyond the community of computer scientists to the biology laboratories that generate the imaging data. We hope to make DL available to all researchers regardless of their laboratory's scale and means. We believe that this democratisation will contribute to the acceptance and validation of DL methods in biomedical research.

ZeroCostDL4Mic complements current community efforts to simplify access to DL in microscopy. Other platforms, however, suffer from either a lack of training capacity (StarDist ImageJ plugin, DeepImageJ[23], CellPose[22], NucleAIzer[17], Cellprofiler[20] or Ilastik[19]), a narrow focus on a single task (i.e., image segmentation with CDeep3M[10], DeepCell Kiosk[15] or DeepMIB[16]) or rely on local servers or paid-for services (as the authors of U-Net have implemented[14], or with ImJoy[18]). Some methods can already be trained on the widely used Fiji platform[21] (DenoiSeg[11], Noise2-Void 2D[8]). However, for most users, the latter methods are only feasible for use if their machines have GPU acceleration without which training times can take 10's to 100's of hours. We believe that ZeroCostDL4Mic fills these gaps and enables affordable and versatile DL-deployment capabilities. It also differs from existing solutions by providing a wide variety of DL tasks within a standardised and user-friendly platform to carry out end-to-end DL workflows: (1) installing relevant computational components, (2) loading training datasets, (3) training models with tailored data, (4) quantitatively validating the performance of models and (5) deploying validated models on new data. ZeroCostDL4Mic also enables researchers to improve their understanding of DL and experiment with hyperparameter optimisation while making informed decisions when choosing appropriate networks for specific applications. These steps help to both leverage the benefits and understand the limitations of DL approaches in research.

Currently, ZeroCostDL4Mic relies on the computing power provided for free by Google Colab. We have characterised in detail the limitations of using Colab for DL and found that RAM availability often limits the number of images that can be used for training DL networks (see Supplementary Table 5). Besides, the runtime duration can, in principle, limit the number of epochs used for training. Importantly, we also showed that each network presented here could be trained efficiently within these boundaries and that transfer learning, as implemented here, can alleviate training time limits. Therefore, we believe that ZeroCostDL4Mic is ideal for carrying out small-scale studies with microscopy data (a few 10's of GB of data). Larger-scale analysis pipelines may require the investment in paid-for cloud-based platforms (i.e., Paperspace, AWS or FloydHub) or local infrastructure.

Although we have focused our development on Google Colab, the ZeroCostDL4Mic notebooks are not strictly dependent on them since they can be ported to any platform that supports Jupyter notebooks. For instance, we demonstrate that the

ZeroCostDL4Mic Deep-STORM and StarDist 2D notebooks can be adapted to run on Deepnote or FloydHub respectively (see Supplementary Note 6 and Supplementary Fig. 4). Paid-for cloud-based solutions often provide more resources than Google Colab, which will allow to train models using more data and larger batch and patch sizes. This can lead to the creation of higher performance and/or more general models. Access to these platforms will only require a small financial investment to train DL models. For instance, FloydHub access costs $9 per month plus $1.2 per GPU hour (as of December 2020). However, as it can take multiple training sessions to find a suitable set of parameters for an individual model and dataset, we recommend first-time users familiarise themselves with training models and explore network suitability using Google Colab initially before moving to paid-for platforms.

One remaining challenge lies in handling, curating, and annotating datasets, especially when performing segmentation tasks: the appropriate preparation of training datasets is always associated with human hours cost, disregarding the platform used for training DL networks. However, we would like to highlight that creating segmentation training datasets can be significantly accelerated by initially training models first with a small number of images[67,68]. These models can then generate masks that can be refined by users (i.e., in Fiji[21]) before being used for training to obtain a high-performance model[5,7,8,11,23,69]. This bootstrapping approach can be carried out iteratively, thereby increasing the network performance progressively while increasing the amount of training data available[56].

Another challenge associated with DL is enabling model versioning to ensure reproducibility. Indeed, a DL model's performance is affected by the training dataset, the network and training parameters but also all the underlying dependencies. To mitigate this issue, in ZeroCostDL4Mic, we provide each trained model with a thorough report that contains all parameters used for training (the type of the data, network parameters, essential package versions) and the model performance (assessed via the quality control section). This logging allows users to easily keep track of parameter changes and training data modifications during model optimisation. Importantly, to contribute to setting good practices on reporting DL model training in the literature, this report is human-readable and can be included as-is in a typical "Methods" section.

Altogether, ZeroCostDL4Mic has the potential to accelerate the uptake of DL for new users and promote their capacity to use powerful image-analysis strategies, thereby readily benefitting from computer science innovations. Together with the help of the broader research community, we expect to grow the number of networks available in ZeroCostDL4Mic quickly. We also expect ZeroCostDL4Mic to become a standard framework that developers can use to showcase and evolve their networks, adapting them to data analysis tasks optimised for their specific image-processing problems. Indeed, since the initial release of ZeroCostDL4Mic, several groups have adapted their novel tools with a similar structure and interface to our notebooks and by exploiting Google Colab[70,71]. Similarly, we will continuously develop, maintain and adapt ZeroCostDL4Mic to be up-to-date with the latest DL paradigms. This focus will incorporate the capacity to maintain compatibility with the rapidly evolving DL libraries available today and the ability to export models that could be used in either DeepImageJ[23], CSBDeep[7] or other prediction engines.

## Methods

**Cell culture**. U-251 glioma cells were grown in DMEM/F-12 (Dulbecco's Modified Eagle's Medium/Nutrient Mixture F-12; Life Technologies, 10565-018) supplemented with 10 % foetal bovine serum (FCS) (Biowest, S1860). U-251

glioma cells expressing endogenously tagged paxillin-GFP were generated using CRISPR/Cas9[72].

MCF10 DCIS.COM (DCIS.COM) lifeact-RFP cells were cultured in a 1:1 mix of DMEM (Sigma-Aldrich) and F-12 (Sigma-Aldrich) supplemented with 5% horse serum (16050-122; GIBCO BRL), 20 ng/ml human EGF (E9644; Sigma-Aldrich), 0.5 mg/ml hydrocortisone (H0888- 1 G; Sigma-Aldrich), 100 ng/ml cholera toxin (C8052-1MG; Sigma-Aldrich), 10 µg/ml insulin (I9278-5ML; Sigma- Aldrich), and 1% (vol/vol) penicillin/streptomycin (P0781- 100 ML; Sigma-Aldrich). DCIS.COM lifeact-RFP cells were generated using lentiviruses, produced using pCDH-LifeAct mRFP, psPAX2, and pMD2.G constructs[73].

HeLa ATCC cells were seeded on fibronectin-coated eight-well chamber slides (Sarstedt, Germany, $1.5 \times 10^4$ cells/well). Cells were grown for 16 h at 37 °C and 5% $CO_2$ in Dulbecco's modified Eagle's medium containing 4.5 g/l glucose, 10% FCS and 1% L-alanyl-L-glutamine (Thermo Fisher, GlutaMAX). To fix the HeLa cells, we employed a protocol shown to preserve the cytoskeleton and organelles (adapted from ref. [74]). The culture medium was directly replaced with PHEM buffer containing 3% methanol-free formaldehyde (Thermo Fisher, USA) and 0.2% EM-grade glutaraldehyde (Electron Microscopy Sciences, USA) and incubated the samples for 1 h at room temperature. Cells were washed thrice with PBS, quenched with 0.2% sodium borohydride in PBS for 7 min and washed again thrice with PBS.

A2780 cells were cultured in RPMI 1640 supplemented with 10% FCS. The cells were grown at 37 °C in a 5% $CO_2$ incubator.

MDA-MB-231 (triple-negative human breast adenocarcinoma) cancer cells were grown in DMEM (Sigma-Aldrich) supplemented with 10% FCS at 37 °C and 5% $CO_2$.

Rat hippocampal neurons from embryonic day 18 pups[75] were cultured on 18-mm coverslips at a density of 6000 cells/cm$^2$ following established guidelines of the French Animal Care and Use Committee (French Law 2013-118 of February 1, 2013) and approval of the local ethics committee (agreement 2019041114431531-V2 #20242). In these neuronal cultures, a small number of glial cells, such as the one shown in Fig. 2, Supplementary Fig. 5 and Supplementary Movie 10 are present and were labelled and imaged.

U2OS cells were purchased from DSMZ (Leibniz Institute DSMZ-German Collection of Microorganisms and Cell Cultures, Braunschweig DE, ACC 785). MDA-MB-231 cells were provided by ATCC. DCIS.COM cells were provided by J. F. Marshall (Barts Cancer Institute, Queen Mary University of London, London, England, UK). HeLa cells were provided by ECACC. U-251 glioma cells were provided by J. Ivaska (University of Turku, Turku, Finland). A2780 cells were a kind gift of P. Caswell (University of Manchester, Manchester, UK).

**Python programming**. We developed our platform on Jupyter Notebooks as a Python interactive environment. The notebooks were developed with Google Colab, which includes several pre-installed packages. Important packages used in ZeroCostDL4Mic include Numpy[76], Keras (https://github.com/keras-team/keras), TensorFlow[77], Augmentor[32], tifffile (https://github.com/cgohlke/tifffile), elasticdeform[28] and imgaug.

**U-Net training dataset**. The training datasets used for segmentation in the U-net ZeroCostDL4Mic notebooks are publicly available EM datasets. For 2D segmentation, this was a neuronal membrane segmentation dataset from the ISBI challenge 2012[78,79] and for 3D segmentation the mitochondrial segmentation dataset available from EPFL (https://www.epfl.ch/labs/cvlab/data/data-em/). Datasets for segmentation tasks can also be created manually. This requires target images that have been segmented by an expert using drawing tools, e.g. in ImageJ/Fiji[21,57], to draw outlines around the structures of interest. For training in the notebook, the source (raw EM image) and target (8-bit mask obtained from expert drawing) images were placed in separate folders, with each source image having a corresponding target image with the same name.

**StarDist training dataset**. DCIS.COM lifeact-RFP cells were labelled using 0.5 µM SiR-Hoechst (SiR-DNA, Tetu-bio, Cat number: SC007) for 2 h. The cells were then imaged live for 14 h on a spinning-disk confocal microscope (one picture every 10 min). The spinning-disk confocal microscope used was a Marianas spinning-disk imaging system with a Yokogawa CSU-W1 scanning unit on an inverted Zeiss Axio Observer Z1 microscope (Intelligent Imaging Innovations, Inc.) equipped with a ×20 (NA 0.8) air, Plan-Apochromat objective (Zeiss).

To generate the StarDist training dataset, mask images were generated manually in Fiji. Briefly, the outlines of each nucleus were drawn using the freehands selection tool and added to the ROI manager. Once all outlines were stored in the ROI manager, the LOCI plugin (https://imagej.net/LOCI) was used to create an ROI map. These ROI map images were then used as the mask images to train StarDist.

To automatically track cells, we sequentially used StarDist and the Fiji plugin TrackMate[56]. Briefly, using our StarDist notebook, nuclei from live-cell imaging data were detected, and tracking files containing the coordinate of their centre (marked by a dot) were generated (StarDist notebook section 6). These tracking files were then used as input for TrackMate[38]. We also provide in the ZeroCostDL4Mic GitHub page a Fiji macro to batch analyse a folder containing

multiple tracking files. Cell tracks were then analysed using the online tool Motility lab (http://www.motilitylab.net/)[80].

**Noise2Void training datasets**. The 2D dataset provided with our notebooks was generated by plating U-251 glioma cells expressing endogenously tagged paxillin-GFP on fibronectin-coated polyacrylamide gels (stiffness 9.6 kPa)[72]. Cells were then recorded live using a spinning-disk confocal microscope equipped with a long working distance of ×63 (NA 1.15 water, LD C-Apochromat) objective (Zeiss). The 3D dataset provided with our notebooks was generated by recording A2780 ovarian carcinoma cell, transiently expressing lifeact-RFP (to visualise the actin cytoskeleton), migration on fibroblast-generated cell-derived matrices[81]. The cell-derived matrices were labelled using Alexa Fluor 488-recombinant fibronectin and the images acquired using a spinning-disk confocal microscope equipped with a 63x oil (NA 1.4 oil, Plan-Apochromat, M27 with DIC III Prism) objective (Zeiss). For both datasets, the spinning-disk confocal microscope used was a Marianas spinning-disk imaging system with a Yokogawa CSU-W1 scanning unit on an inverted Zeiss Axio Observer Z1 microscope controlled by SlideBook 6 (Intelligent Imaging Innovations, Inc.). Images were acquired using a Photometrics Evolve, a back-illuminated EMCCD camera ($512 \times 512$ pixels).

**CARE training datasets**. Briefly, DCIS.COM lifeact-RFP cells were plated on high-tolerance glass-bottom dishes (MatTek Corporation, coverslip 1.5) and were allowed to reach confluence. Cells were then fixed and permeabilised simultaneously using a solution of 4% (wt/vol) paraformaldehyde and 0.25% (vol/vol) Triton X-100 for 10 min. Cells were then washed with PBS, quenched using a solution of 1 M glycine for 30 min, and incubated with phalloidin-488 (1/200 in PBS; Cat number: A12379; Thermo Fisher Scientific) at 4 °C until imaging (overnight). Just before imaging using SIM, samples were washed three times in PBS and mounted in Vectashield (Vectorlabs). The SIM system used was Delta-Vision OMX v4 (GE Healthcare Life Sciences) fitted with a ×60 Plan-Apochromat objective lens, 1.42 NA (immersion oil RI of 1.516) used in SIM illumination mode (five phases and three rotations). Emitted light was collected on a front-illuminated pco.edge sCMOS (pixel size 6.5 µm, read-out speed 95 MHz; PCO AG) controlled by SoftWorx. In the dataset, the high signal-to-noise ratio images were acquired from the phalloidin-488 staining using acquisition parameters optimal to obtain high-quality SIM images (in this case, 50 ms of exposure time, 10% laser power). In contrast, the low signal-to-noise ratio images were acquired from the LifeAct-RFP channel using acquisition parameters more suitable for live-cell imaging (in this case, 100 ms of exposure time, 1% laser power). The dataset provided with the 2D CARE notebooks are maximum intensity projections of the collected data.

**Label-free prediction (fnet) training dataset**. The dataset provided for training label-free prediction notebook was designed to predict a mitochondrial marker from bright-field images. Before the acquisition, fixed HeLa ATCC cells were permeabilised and blocked using 0.25% Triton X-100 (Sigma-Aldrich, Germany) and 3% IgG-free bovine serum albumin (BSA, Carl Roth, Germany) in PBS for 1.5 h. Cells were labelled for TOM20 using 5 µg/ml rabbit anti-TOM20 primary antibody (sc-11415, Santa Cruz, USA) and 10 µg/ml donkey-anti-rabbit-secondary antibody (Alexa Fluor 594 conjugated, A32754, Thermo Fisher, USA) in PBS containing 0.1% Triton X-100 and 1% BSA for 1.5 h each. Samples were rinsed twice and washed thrice with PBS (5 min) after each incubation step. Image stacks were acquired on a Leica SP8 confocal microscope (Leica Microsystems, Germany) bearing a ×63, 1.40 NA oil objective (Leica HC PL APO). The pixel size was set to 90 nm in $XY$-dimensions, and 150 nm in $Z$ (32 slices) and fluorescence image stacks were recorded using 561-nm laser excitation and collected by a (photo-multiplier tube) PMT. The corresponding transmitted light image stack was recorded in parallel using a transmitted light PMT. We acquired 25 3D stacks with dimensions of $1024 \times 1024 \times 32$. To create the training set each image stack was split into four stacks with dimensions of $512 \times 512 \times 32$, giving a dataset of 100 images of which 92 were used for training and 8 unseen for testing and quality control (see Table). The raw data were converted into.tif file format and split into stacks of the respective channels (fluorescence and transmitted light). To prepare a training set, stacks were split into individual folders by channel. To create matched training pairs, the signal files (transmitted light) and their respective targets (fluorescence) must be in the same order in the irrespective folders. It is therefore advisable to number source-target pairs or to give the files the same names.

**Deep-STORM training and example dataset**. For Deep-STORM, training data and test data can be generated via SMLM data simulations that mimic the experimental data type that needs to be subsequently analysed. This is directly possible within the notebook that we provide. The example experimental data that we provide was obtained from a glial cell in a culture of rat hippocampal neurons, fixed and stained with phalloidin-Alexa Fluor 647[82], then imaged on a Nikon N-STORM microscope using ×100, NA 1.49 TIRF objective ($256 \times 256$ pixels with 160-nm pixel size, acquiring 59,900 frames at 15 ms/frame) controlled by NIS-Elements (version 4.4). In order to simulate a higher density of emitters, we binned the data in groups of four frames using Fiji (Grouped z-project/Sum slices plugin)

**Table 1 Simulation parameters used to generate training data for Deep-STORM datasets.**

| Dataset | BIN4 | BIN10 | TUB |
|---|---|---|---|
| FOV size (nm) | 10,240 | 10,240 | 15,800 |
| Pixel size (nm) | 160 | 160 | 158 |
| ADC/photon-conversion factor | 16 | 16 | 12.7 |
| Read-out noise (ADC) | 1040 | 1800 | 370 |
| Offset (ADC) | 10,550 | 26,500 | 4090 |
| Emitter density (#/um$^2$) | 2.8 | 4 | 3 |
| STD of emitter density (#/um$^2$) | 0 | 0.8 | 0.5 |
| Number of frames | 20 | 20 | 20 |
| PSF sigma (nm) | 153 | 150 | 160 |
| STD of PSF sigma (nm) | 29 | 25 | 30 |
| Number of photons | 3500 | 5500 | 2800 |
| STD of number of photons | 850 | 2000 | 900 |

leading to 14,975 frames (BIN4), or groups of ten frames leading to 5990 frames (BIN10). The binning was performed on a 32-bits dynamic range.

In order to train the model for the reconstruction of the BIN4 and the BIN10 dataset, 20 frames of a $64 \times 64$ pixels field-of-view were simulated with parameters matching the experimental datasets closely. The parameter lists are shown in Table 1.

The localisation files obtained from our Deep-STORM notebook were subsequently imported in ThunderSTORM[51] for drift correction using cross-correlation analysis[83].

For DNA-PAINT imaging of the tubulin cytoskeleton, U2OS cells were fixed and immuno-labelled with mouse-anti-β-tubulin primary antibody (32-2600, Thermo Fisher) and DNA-conjugated (P1-docking strand) secondary antibody. DNA-PAINT imaging was performed using a Nikon N-STORM system bearing a Nikon Apo TIRF $\times100$ oil immersion objective (1.49 NA). In all, 2 nM P1-ATTO655 (8 nt duplex, Eurofins) in PBS pH 8.2 + 500 mM NaCl was added to the sample, resulting in a labelling density significantly higher than used for conventional DNA-PAINT imaging. Fluorophores were excited in highly inclined and laminated optical sheet mode using ~1 kW/cm² 647-nm laser illumination. In total, 2000 frames were recorded at 10-Hz frame rate and an EM gain of 200.

Parameters for simulating the training data (TUB) were estimated from the raw movie or ThunderSTORM ME-MLE reconstruction of a few frames and are listed in Table 1. The training was performed for 30 epochs and 1750 steps per epoch using a batch size of 4. Both localisations obtained from Deep-STORM and ThunderSTORM analysis were corrected for drift using the cross-correlation function in ThunderSTORM. Emission events that are split onto subsequent frames were merged, applying a 40 nm distance threshold (no dark frame allowed).

**YOLOv2 training dataset.** The YOLOv2 dataset is composed of live-cell movies of breast cancer cells (MDA-MB-231) migrating on cell-derived matrices[81] generated by human fibroblasts[84]. MDA-MB-231 cells were seeded at a density of 5000 cells per ml on cell-derived matrices and allowed to spread for four hours. Cells were then filmed using an inverted wide-field microscope (AxioCam MRm camera, EL Plan-Neofluar 20/0.5 NA objective (Carl Zeiss)) equipped with a heated chamber (37 °C) and CO$_2$ controller (5%). Images were collected every 10 min.

To create the annotations, 30 individual images of size $1380 \times 1040$ were saved as .png files and loaded into https://www.makesense.ai/, an online annotation platform (for documentation, see: https://github.com/SkalskiP/make-sense/). A new project was started with the "Object Detection" option. Next, a list of labels was created using the "+" option and typing the names of the classes to be identified. The option "Going on my own" was selected after the creation of the labels and without selecting the "COCO SSD…" or "POSE-NET…" boxes. In the next step, bounding boxes were drawn using the cursor and object classes were added by clicking on the "Select Label" option on the right side of the window and selecting from the previously assembled labels list, now available as a dropdown list. After all images in the dataset have annotated, the annotations were downloaded by selecting "Export Labels" and checking the box "A.zip package containing the files in VOC.XML format". The annotation files were then downloaded in a zip folder with the original images' filenames and the.xml file suffix. These files were used as targets for the training of YOLO network in our notebook. We used the imgaug library to augment the dataset and the bounding boxes by rotation and flipping.

**CycleGAN training dataset.** U2OS cells were plated on fibronectin-coated glass-bottom dishes (MatTek Corporation) for 2 h before methanol fixation at −20 °C for 5 min. Fixed samples were washed three times using PBS and stained with an anti-tubulin antibody (clone 12G10, Developmental Studies Hybridoma Bank) for 35 min at room temperature. Samples were then washed thrice with PBS and

incubated with an anti-mouse secondary antibody conjugated to Alexa488 (Thermo Fisher Scientific. Cat number: R37114). Stained samples were washed thrice with PBS and kept at 4 °C, in PBS, until imaging.

The SDC dataset was acquired using a Marianas spinning disk equipped with a Yokogawa CSU-W1 scanning unit on an inverted Zeiss Axio Observer Z1 microscope (NA 1.4 oil, Plan-Apochromat, M27) objective and controlled by SlideBook 6 (Intelligent Imaging Innovations, Inc.). Images were acquired using a Photometrics Evolve, a back-illuminated EMCCD camera ($512 \times 512$ pixels). For each field of view, 200 images were acquired. The SDC images used to train CycleGAN were generated by performing average projections of the collected images. To maintain a uniform pixel size across the SDC, SIM, and Fluctuation-based super-resolution (FBSR) images, the SDC images were magnified by four using a bilinear interpolation.

The fluctuation-based super-resolution dataset was acquired by processing the SDC images using the latest implementation of NanoJ-SRRF[55] within the ImageJ software[21,57]. This new version of SRRF (super-resolution radial fluctuations) is available upon request and will be openly available for download soon. The SRRF settings were chosen so that the least amount of errors were present in the reconstructed images (estimated using SQUIRREL[52]) and were as followed: "vibration correction", on; radius, 2; sensitivity, 2; magnification, 4; temporal analysis, average; intensity weighting, on; macro-pixel patterning correction, on.

The SIM dataset was acquired using a DeltaVision OMX v4 (GE Healthcare Life Sciences) fitted with a $\times60$ Plan-Apochromat objective lens, 1.42 NA (immersion oil RI of 1.512) used in SIM illumination mode (five phases and three rotations). Emitted light was collected on a front-illuminated pco.edge sCMOS (pixel size 6.5 μm, read-out speed 95 MHz; PCO AG) controlled by SoftWorx. Each dataset was augmented by five by randomly cropping (original size $2048 \times 2048$ px, crop size $1280 \times 1280$), flipping and rotating the original images. This augmentation pipeline was generated using Augmentor[32], for which we also provide a ZeroCostDL4Mic notebook.

**pix2pix training dataset.** DCIS.COM lifeact-RFP cells were incubated for 2 h with 0.5 μM SiR-DNA (SiR-Hoechst, Tetu-bio, Cat Number: SC007) before being imaged live for 14 h (1 picture every 10 min) using a spinning-disk confocal microscope. The spinning-disk confocal microscope used was a Marianas spinning-disk imaging system with a Yokogawa CSU-W1 scanning unit on an inverted Zeiss Axio Observer Z1 microscope (Intelligent Imaging Innovations, Inc.) equipped with a $\times20$ (NA 0.8) air, Plan-Apochromat objective (Zeiss).

**Statistics and reproducibility.** Unless otherwise specified, all experiments were performed once. Here, we are assessing the performance of computer algorithms and not the variability of biological systems.

**Ethics declarations.** We confirm that we complied with all the relevant ethical regulations for animal testing and research.

**Reporting summary.** Further information on research design is available in the Nature Research Reporting Summary linked to this article.

## Data availability

All our example datasets are available for download in Zenodo (Links also available on our GitHub page): CARE 3D[85], CARE 2D[86], Noise2Void 3D[87], Noise2Void 2D[88], Deep-STORM[89], CycleGAN[90], pix2pix[91], YOLOv2[92], StarDist 2D[93], label-free prediction fnet[94]. The datasets used to train 2D U-net were originally published as part of the ISBI 2012 segmentation challenge[32,79] and were retrieved for this work from https://github.com/zhixuhao/unet. The dataset used for 3D segmentation in U-Net 3D is publicly available from the page of the École polytechnique fédérale de Lausanne (EPFL): https://www.epfl.ch/labs/cvlab/data/data-em/. Source data are provided with this paper.

## Code availability

ZeroCostDL4Mic is available as Supplemental Software or can be accessed from our GitHub page[95] https://github.com/HenriquesLab/ZeroCostDL4Mic. This resource is fully open-source, providing users with tutorials, Jupyter Notebooks for Google Colab, and many real-life example datasets for training and testing.

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

## Acknowledgements

First and foremost, we would like to thank Dr. Martin Weigert from the Swiss Federal Institute of Technology (EPFL) and Dr. Uwe Schmidt from the Max Planck Institute of Molecular Cell Biology and Genetics (MPI-CBG), who pioneered a considerable portion of the technology this work is based on and whose ethos in making deep learning more accessible for microscopy helped inspire this work. Dr Schmidt has kindly given key feedback during the preparation of the paper. All the network architectures and tasks presented here originate from already published work, having been edited and prepared for Google Colab to simplify their uptake by novice users. When using the ZeroCostDL4Mic platform, please cite the original publications associated with each network. We would like to also thank Estibaliz Gómez de Mariscal and Ignacio Arganda-Carreras for providing code and support to export and use U-Net (2D) and Deep-STORM models trained in ZeroCostDL4Mic in DeepImageJ. This work was funded by grants from the UK Medical Research Council (MR/K015826/1) (R.H.), the Wellcome Trust (203276/Z/16/Z) (R.H.), the Gulbenkian Foundation (R.H.), This project has also received funding from the European Research Council (ERC) under the European Union's Horizon 2020 research and innovation programme (grant agreement No. [101001332]) (R.H.) and an European Molecular Biology Organization (EMBO) Installation Grant (EMBO-2020-IG-4734) (R.H.). R.F.L. would like to acknowledge the support of the MRC Skills development fellowship (MR/T027924/1). This work was also supported by grants awarded by the Academy of Finland (to G.J. and P.K.M.), the Sigrid Juselius Foundation (to G.J. and P.K.M.), the University of Turku foundation and Turku Doctoral Program in Molecular Medicine (TuDMM)(to S.H.P.), Åbo Akademi University Research Foundation (CoE CellMech; G.J.) and by Drug Discovery and Diagnostics strategic funding to Åbo Akademi University (G.J.). We thank Dr Aki Stubb for providing us with the raw data used to showcase Noise2Void 2D. The Cell Imaging and Cytometry Core facility (Turku Bioscience, University of Turku, Åbo Akademi University, and Biocenter Finland) and Turku Bioimaging are acknowledged for their services and instrumentation and expertise. M.L is supported by Victoriastiftelsen (FI). A.K., T.-O.B. and F.J. funded by the German Research Foundation (DFG) under the code JU3110/1-1 and German Federal Ministry of Research and Education under the code 01IS18026C, ScaDS2. M.H. and C.S. acknowledge funding by the German Science Foundation (grant nr. SFB1177), C.S. additionally acknowledges support by the European Molecular Biology Organization (short-term fellowship 8589). C.L. acknowledges the support of CNRS through the ATIP AO2016 grant. S.H. and E.K. were supported by a Wellcome Trust & Royal Society Sir Henry Dale Fellowship grant number 206670/Z/17/Z to S.H. We additionally thank Chan Zuckerberg Biohub and its donors for funding Loic A. Royer and Ahmet Can Solak's work. This work was supported by the Francis Crick Institute, which receives its core funding from Cancer Research UK (FC001999), the UK Medical Research Council (FC001999), and the Wellcome Trust (FC001999). Note: CoLaboratory™ and Google Drive™ are trademarks of Google LLC - ©2018 Google LLC All rights reserved.

## Author contributions

G.J. and R.H. conceived the project; L.v.C., R.F.L, J.J, D.K., E.N., T.-O.B., G.J. and R.H. wrote source code based on the work of A.K., T.-O.B., F.J., E.N., Y.S. and L.A.R. among others; L.v.C., R.F.L., J.J., C.S., C.L. and G.J. performed the image acquisition of the test and example data; L.v.C., R.F.L., J.J., C.S., M.L., S.H.-P., P.K.M., E.K., S.H., A.K., T.-O. B., C.L., M.L.J., D.K., E.N., Y.S., M.H. and G.J. tested the platform; L.v.C., R.F.L., J.J., C.S., G.J. and R.H. wrote the paper with input from all co-authors.

## Competing interests

We provide a platform based on Google Drive and Google Colab to streamline the implementation of common Deep Learning analysis of microscopy data. Despite heavily relying on Google products, we have no commercial or financial interest in promoting and using them. In particular, we did not receive any compensation in any form from Google for this work. The authors declare no competing interests.
