## [Peer Review File · Nature Communications]

Reviewers' Comments:

Reviewer #1:

Remarks to the Author:

"ZeroCostDL4Mic: an open platform to use Deep-Learning in Microscopy" by von Chamier, Laine, Henriques and colleagues describes software based on Google Colab - a free cloud platform supported by Google designed for interactive computing - that enables users to train deep learning models and run these models on their data. The software consists of a GUI that enables users to connect their Google Drive to active Colab instances (and hence make their data available to the software) and Jupyter notebooks that guide users through model training, quality control, and inference. The authors make a collection of pre-trained models available through this library - these include models for image restoration (CARE and Noise2Void), super resolution (DeepSTORM), image-to-image translation (pix2pix and cycleGAN), and EM & nuclear segmentation (U-Net and StarDist).

All told this is a nice paper that makes a valuable contribution to this space. Reproducibility is an issue that computational biologists have been dealing with for a long time, and it is pleasing to see so many methods reproduced in one location. That said, I have some suggestions that I think would improve the clarity of the presented work and make it easier for readers - both novice and expert - to navigate this rapidly moving landscape. I describe these suggestions below.

-Novelty. One of the most significant issues with this paper is lack of novelty. This is acknowledged with respect to algorithms, as all of the presented methods have been published previously. But it is also true with respect to the underlying software architecture. Using Google Colab as a means to distribute computational methods has been done before in other contexts (the most notable example that comes to mind is kallisto bustools by Pachter and colleagues - <https://www.kallistobus.tools/tutorials>). The schema that the authors have set up (Google Colab, Google Drive, and GitHub) is also one that is frequently used in the data science community. Google Colab notebooks for two of the author's methods (pix2pix and cycleGAN) are already present within the TensorFlow documentation, guides for connecting google drive to Colab instances exist within Google Colab's documentation, and numerous blog posts exist describing how to link Colab, Drive, and GitHub to create a development environment. That's not to say that any of these pre-existing works nullify the author's contribution. But I think it would be more accurate to frame this work as bringing existing data science workflows into the life sciences as opposed to creating a brand new conceptual framework for writing software.

-Context. I think the paper could do a better job of placing its work in context of ongoing community efforts. The paper briefly mentions other software efforts (ImJoy, ilastik, and ImageJ), but this is an incomplete cross section of this space. Following the bioinformatics traditions, several groups have made pre-trained models available through web portals (e.g. CellPose from HHMI, NucleAIzer by Horvath and colleagues, and DeepCell from Caltech). There is also CDeep3M from UCSD which allows users to train and deploy models on an AWS instance. Briefly mentioning these other works would allow readers (and potential users) to better place this work in context and understand how the author's software differs - both in positive and negative aspects.

-Vision and Limitations. One of the more serious issues I have with this paper is the author's vision for the deep learning/biological imaging interface. I don't think this is well communicated (although it could be inferred from the paper), and this lack of clarity leads to significant blind spots in the paper with respect to the software's limitations. Based on my understanding of the author's work, a fair view of this vision is that users would collect training data specific to their own project, oversee its annotation, train a model on Colab, process existing data with this trained model on Colab, and download this trained model for later use. The presence of pre-trained models allows users to use transfer learning to speed up the model training process and possibly yield more accurate models (although the utility of transfer learning for model accuracy is in question - see for instance <https://arxiv.org/abs/1805.08974>). This vision centers around a single

user's needs and has a single person (or a small number of people) overseeing the effort. This vision may make sense for some use cases. For instance, for reconstruction methods like CARE, it is likely best that users train models on their own data.

I believe this vision has several issues that I feel should be brought to the author's attention. First is the issue of data. Deep learning is driven by data and it is impossible to develop models without it. An under-appreciated fact is that the difficulty of creating annotated training data has actually shaped methods development. While some tasks like image restoration or image-to-image translation have an intrinsically low annotation burden (they simply require collecting pairs of images), the same is not true for tasks like segmentation. With that in mind, I find it surprising that there's little mention of the data challenges involved in model development. To maximize user adoption, I feel there needs to be some discussion of the data challenges surrounding deep learning and some solution offering, even if it is pointing users to existing annotation tools developed by previously mentioned groups (e.g. ImJoy, CellPose, DeepCell, etc). Given that training datasets change over time and that models are derived from said datasets, there is a need for some form of model versioning for model developers, much like versioning is required for writers of software. Absent versioning, users will find that a myriad of issues (poorly documented models, no chain of custody between data and models, over writing pre-existing useful models, etc.) will arise in a short period of time. That's not to say the authors need to solve all these problems for this paper to be a valuable contribution. But the paper's blind spot to the data challenges surrounding using deep learning in practice does impact how useful users will ultimately find this software.

Also problematic is the issue with the scale of training data. Deep learning methods work best for big data; for most tasks, the same model architecture trained with the same algorithm for the same number of steps on a larger, more diverse dataset will see a performance boost. Big data comes with a price however, and for this work the most relevant price is memory. Google Colab instances are limited to 12GB of memory; purchasing Colab Pro (9.99 USD/month) can boost this to 25GB but this nuance can be ignored given the constraint of zero cost. Practically speaking, that means that the author's software is incapable of taking advantage of big data due to memory limits and that models that result from it will have suboptimal performance. The memory limits of the available GPUs also form a similar constraint, given that mixed precision training is not available for the GPUs available to users. This means users are limited to models with smaller capacity and training them with smaller batch sizes, both of which can limit performance. Datasets for which memory is an issue already exist - two that come to mind are DeepCell's live cell imaging dataset (Caltech) and the PanNuke dataset for nuclear segmentation and cell type detection in H&E images (University of Warwick). The vision issue surrounding this limitation is simple - who is this software for? Is it for novice users or experts? The former may be less concerned with these issues (although they will certainly be impacted) but the latter almost certainly will be. I find the discussion in the supplement inadequate - the limits Colab's hardware options place on the kinds of models that users can develop makes me skeptical about the impact the author's claim this work will have.

The same issue arises for inference. How much data can one process using Colab? This is unclear. While instances are supposed to remain active for 12-24 hours, a number of issues limit throughput. The type of the available GPU is stochastic - assuming one is available - and the ones that are available limits the options available to increase inference speed (e.g. performing inference at reduced precision like float16 or int8). Moreover, Google Colab throttles heavily active users - even for the pro version - to ensure accessibility for the broader community. The author's do not investigate the inference rates achievable with their software. Given the ever increasing size of imaging datasets, I think such an analysis is necessary. Moreover, I don't think it was ever Google's intention to pay the computing costs for the entire bio-image community, which is what the authors are effectively suggesting.

I think most of these issues could be resolved with a more thoughtful discussion and analysis of

the limitations posed by the author's software architecture. Some discussion is found in the supplement, but I believe this merit's a discussion in the main text.

-Accuracy. Related to the previous point, the authors make an argument in Supplementary Note 1 and 5 that models are generally specific to the acquisition instrument and that they often don't generalize across microscopes. While generalization is an issue, the claim that generalization can't be achieved is untrue, as is the claim surrounding the computational power required to train said models. Work in the literature has shown that image normalization (<https://journals.plos.org/ploscompbiol/article?id=10.1371/journal.pcbi.1005177>) and diversity of training data (<https://www.nature.com/articles/s41592-019-0612-7>, <https://www.biorxiv.org/content/10.1101/2020.02.02.931238v1.abstract>) and image resizing (<https://www.sciencedirect.com/science/article/pii/S2405471220301174>) is sufficient to achieve generalization. Generation of annotated training data that spans cell types and acquisition instruments is the most difficult aspect of this work, but it is achievable and the models derived from these efforts can usually be used safely. While this has yet to be proven true for tasks like image restoration (which are more ill-posed), it is certainly true for cell segmentation tasks. The pervasive deployment of deep learning models on data generated from mobile devices, which have a diversity of cameras, is further proof of this point.

Moreover, the author's statement that training models for these larger datasets is time consuming and resource intensive to the point that most lab's can't afford it is also incorrect. In our hands, training models on larger datasets (e.g. ~2000 512x512 images) can be done in several hours with a modest GPU (e.g. V100) - without performing time saving tricks like training at reduced precision.

The author's assertion about the climate impact of training these models is also incorrect. In addition to being faster, the more modern GPUs are more power efficient, which leads to a lower carbon footprint as well as lower prices (see for example <https://cloud.google.com/blog/products/ai-machine-learning/your-ml-workloads-cheaper-and-faster-with-the-latest-gpus>). Moreover, training in less time means less CPU utilization, which also reduces cost and footprint. Training models on a K80 on Colab is arguably worse for the environment than purchasing time on a more modern machine. The authors are likely confusing the time required for training vision models with the time required for language models. The self-supervised pre-training required by language models is both time and resource intensive.

-Cost. Is "ZeroCostDL4Mic" actually free? I would argue no. The notion that the cost is zero ignores the cost of human labor throughout the entire model development process. There is the cost involved in dataset annotation (which can easily cost >10,000 USD based on the application) and the cost incurred by using suboptimal models (whose results will require curation). There is also the opportunity cost of running training and inference jobs on out-dated hardware, as having models and model results sooner.

While the authors may disagree, the paper in its current form does not present an argument to the contrary. Moreover, in my view the current manuscript does not make a compelling case for Colab over a more proper cloud infrastructure, particularly given that preemptible GPU instances can be had for as little as 0.1 USD/hour. There is no analysis of the actual cost of producing a deep learning model in the cloud (provided training data is available) or the cost of performing inference. How much does training a single deep learning model in the cloud cost? How much is the cost of analyzing a single image? Is that price worth the limitations imposed by Google Colab? Only a prospective user can say, but having that information present would let them make an informed decision about how the author's work can meet their own analysis needs.

Other issues with the paper that should be addressed

-The memory footprint and inference speed of the models should be reported. Given that there is variance of GPU type (K80 vs P100, etc), this should be reported for each GPU type, not just the

type that was available to the authors at benchmarking time.

-The maximum dataset size permitted by the CPU and GPU memory limits of Google Colab should be reported for the model architectures used.

-While model chaining is something that is desirable, it is not new. The examples the authors give (translating label free images into nuclear images and subsequently performing segmentation and tracking) has been described previously (<https://www.biorxiv.org/content/10.1101/2020.03.05.979419v1.abstract>). Moreover, the chaining of segmentation and tracking models has also been described previously (<https://www.biorxiv.org/content/10.1101/803205v1>). This is a functionality allowed by the presence of multiple performant models in the same programming environment, rather than something unique to the author's software - this should be made clear. It is also unclear how the authors plan to make model chaining accessible through Colab in a way that does not involve programming.

-The ability to use semi-synthetic data for training image restoration models, as pioneered by Manor and colleagues (<https://www.biorxiv.org/content/10.1101/740548v8.abstract>) would likely make these methods more accessible.

I found the notebooks presented rather clumsy for the purpose of using a pre-trained model. So much so that if given the choice between the Colab or a web portal/ImageJ plugin, I would use the latter option. An alternative could be to have notebooks solely for inference with a pre-trained model where appropriate.

-The author's should investigate Tensorflow Hub as a storage location for pre-trained models to increase their accessibility.

-It would be nice if benchmarks for pre-trained models were available on the project's front page.

-I found the benchmarking methods presented in this paper to be somewhat confusing and underwhelming. The SSIM measure for image restoration can be biased by the presence of background pixels. It seems to me that a better alternative would be to compute it only for foreground pixels - although I could be mistaken. The use of mAP and IoU for benchmarking segmentation is also unclear. Is IoU computed on a per object basis, or just in aggregate over all the pixels over foreground. The mAP can also be a misleading measure of segmentation performance, as it includes information about precision at cut off values that are not used in practice. A more informative approach I would suggest is the F1 score, augmented by statistics about error types (splits, merges, etc). This was first reported by Caicedo and colleagues (<https://onlinelibrary.wiley.com/doi/full/10.1002/cyto.a.23863>) and expanded on further by Moen and colleagues (<https://www.biorxiv.org/content/10.1101/803205v1>).

-The pip commands in the notebooks that install the packages do not appear to be versioned. This is problematic, as versioning is an essential part of reproducibility. A model trained under one version of TensorFlow can produce different results when inference is performed in a different version. I would recommend versioning if possible, which can be done by specifying them in a requirements.txt file. Docker would be the gold standard, but it is not clear to me how to make Docker work in the context of the author's framework.

-Computation precision (float32, vs float16, vs bfloat16) may be a reason why the authors see poor performance on TPU devices. This likely merit's further investigation.

-The inductive bias of object detection models (e.g. positing that cells can fit inside an unrotated bounding box with limitations on the aspect ratio) is likely the origin of their poor performance. Investigation of other object detection methods (e.g. RetinaNet or a more modern YOLO model) is warranted. Also, the authors may want to make use of a resource generated by a recent CVPR

paper

(https://openaccess.thecvf.com/content_CVPRW_2020/html/w57/Anjum_CTMC_Cell_Tracking_With_Mitosis_Detection_Dataset_Challenge_CVPRW_2020_paper.html)

Reviewer #2:

Remarks to the Author:

The authors present a software library (in fact a collection of existing software packages) which use deep learning (DL) for different aspects of image analysis. By bringing together disparate packages and unifying them to have a common interface (via Jupyter Notebooks) and implementing them to run on a common computing platform (Google CoLab), this work aims to simplify and illustrate the use of DL for image analysis for researchers who may not be familiar with the technical details of these methods.

In my view, this work is a very positive development for the microscopy field, and will accelerate the uptake of DL methods, leading to new biology results and the refinement / development of further image analysis methods. A quick look at the github page for this project shows a well-organized software library with an active userbase – this is already a strong validation of the concept the authors have developed in ZeroCostDL4Mic.

The choice of Google CoLab as a computing platform is the one aspect of the work which I would ask the authors to address. Commercial enterprises such as Google do offer free services such as CoLab, but platforms such as this may be withdrawn at any time if they are not profitable, or if the company decides on a different strategy. Therefore, I would ask the authors to consider the question of what would happen if, tomorrow, Google decided to shut down the CoLab service. Would the ZeroCostDL4Mic library be usable? Would the publication in Nature Communications still have value, or would it become obsolete when the CoLab service ends? Could ZeroDLCost4Mic be run in the same way on a different computing platform? It could be that the authors have already answered these questions and I have missed it, but those are the only significant points that I wish to raise in this review.

A minor point is also that the CoLab service is indeed "Zero cost" when free resources are available, but for larger tasks users may be required to sign up for "CoLab Pro", which is a pay-for-use service offered by Google. I feel that this point is worth mentioning, so as to avoid giving the impression that unlimited free image processing resources are available from Google.

Typos etc.:

1. Main text line 161: change "allowing to" to "allowing one to" or equivalent
2. Main text line 188: Missing ")" after "Video 2-11".
3. Main text line 200: Missing "," after "ilastik". Start new sentence after "Fiji/ImageJ".
4. Main text line 231: Add footnote including link to Zenodo.
5. Online methods line 370, 375: Acronym "FBSR" not defined. "SRRF" also not defined.
6. Supplemental information line 172: "EM" not defined.

This manuscript presents a new and valuable image analysis resource. Works such as this, which

simplify and make accessible pre-existing methods, is often not recognized in the literature even though they can be as important as the initial development of the methods themselves. I recommend this work for publication, with minor revisions as described above.

Reviewer #3:

Remarks to the Author:

In their manuscript "ZeroCostDL4Mic: an open platform to use Deep-Learning in Microscopy", the authors ported a set of tools, which use deep neural networks for image processing task to notebooks to be used in Google Colab. The software can be run using the GPU infrastructure provided by Google Colab which is provided to the public for free. The authors distribute their notebooks as open source through a publicly accessible GitHub repository and placed an emphasis on documentation of the different notebooks to guide new users through the steps.

I find the GitHub repository is providing a set of useful tools for the biomedical community. The steps to accomplish the results and to use the tools are aimed to be clear and I expect it will result in a good acceptance of the tools. In its current form the manuscript reads like an advertisement rather than a scientific manuscript. I think this could be improved by focusing on benchmarking important features and demonstrations. The odd structure of the manuscript makes it sometimes difficult to find all the information about the contributions and some new developments from the authors, who I believe sometimes added features to the different networks. To facilitate reading the manuscript, it would be very helpful to have the main text subdivided into the different networks and an have evaluations of how the networks fare on Colab.

My major concerns are regarding the scientific presentation of the work and the evaluations performed so far. The manuscript is often hyperbolic and many arguments made would need quantifications or more thorough research. The limitations of Google Colab are only mentioned at the very end of the supplement, which would mean many readers will not be aware of those. It should be made clearer to the readers that the free offerings come with several limitations.

Major:

1) The description the authors provide about the different Google Colab implementations do not provide enough information for an end-user to decide if a particular notebook is an appropriate solution for a potential end-user. Are the notebooks one-to-one implementations of the previous implementations or are there differences in some cases?

2) What size of data can actually be processed on Google Colab with the different notebooks? There are different potential limitations that might differ for the notebooks, like data loader, throughput, paired with disconnection time of Colab. How is the processing throughput on Colab compared to throughput e.g. with a local installation (RAM and CPUs are very limited on Colab)? Do all of the networks require a GPU, for prediction and for training? How is the utilization of the underlying hardware (minimum requirements)? Which also means, may a regular graphics card be sufficient in some of those cases?

3) Related to point 2, it would be important for the authors to explain more about working or processing on Colab. E.g. the installation times of the notebooks? (since they need to be installed each time a notebook is started, the time of each installation should be clearly stated).

4) The manuscript is based on the assumption that deep learning is great, and the networks here are the ideal choice but the assessment of the performance provided here is limited. Assessments about the metrics of IoU or SSIM are embedded in images (e.g. Supplementary Fig. 9 and Supplementary Fig. 15) rather than written in plain text or spelled out in graphs, which makes it very difficult to find the relevant information when reading the manuscript or to compare with other methods in the literature. Are all of those state-of-the-art IoU values?

5) Hyperparameters: in most cases I can't find which hyperparameters were used to train the models. E.g. Supplementary Figure 9, which hyperparameters that lead to overfitting and which that did not lead to overfitting?

Other comments:

- The authors mention a common organization theme between the notebooks, which makes it easier for others to implement their network, but it's unclear what this actually implies. Do the authors provide a template for this?

- L80-83/L198-214: Better embedding into the current literature would help the readers instead of a blanket statement that this 'considerable simplifies the use of DL for microscopy'. The approach by the authors is not the first or only approach that provide easy end-to-end solutions for DL.

- Supplementary Fig. 11: This figure displays an example of how adjusting parameters (batch size, number of steps and number of training epochs) can affect model performance in StarDist nuclear segmentation. Alternative interpretation: If you train enough epochs, the other 2 parameters don't matter? How much do those epochs take in wall clock time on Colab? Do they run uninterrupted until the end, or does the user get disconnected from the notebook in between?

- L74-79: I could not find the actual quantification. Some of it could be in Supplementary Fig 15, if so, the metrics should be represented in the text and/or graph and appropriately quantified, rather than imprinted on an image. (Also concerns: L61-74) How much time takes the new or re-training, how much does the IoU improve? Importantly: How much time does it take to (manually?) generate the new training data?

- Colab has a policy to limit throughput of data, which the authors describe as unreliability to access GPUs, but indeed this is Google's policy which is against using their infrastructure too heavily. It is meant as a developer environment, not production scale. This should be made clearer for the readers.

- L174-175 the computational resources provided by Google Colab is mostly a GPU, but is otherwise very limited in RAM and CPUs, which would be important to explore if this limits the throughput of the individual notebooks or not

Dear reviewers and editorial team,

We thank you for your time in evaluating our manuscript "ZeroCostDL4Mic: an open platform for Deep-Learning in Microscopy". We are grateful for the detailed comments provided by the three referees. In this revision we have addressed all the comments raised, allowing us to formalise a greatly improved manuscript. Below we detail our response and changes made to the manuscript based on your feedback:

REVIEWER COMMENTS

Reviewer #1 (Remarks to the Author):

R1.0. "ZeroCostDL4Mic: an open platform to use Deep-Learning in Microscopy" by von Chamier, Laine, Henriques and colleagues describes software based on Google Colab - a free cloud platform supported by Google designed for interactive computing - that enables users to train deep learning models and run these models on their data. The software consists of a GUI that enables users to connect their Google Drive to active Colab instances (and hence make their data available to the software) and Jupyter notebooks that guide users through model training, quality control, and inference. The authors make a collection of pre-trained models available through this library - these include models for image restoration (CARE and Noise2Void), super resolution (DeepSTORM), image-to-image translation (pix2pix and cycleGAN), and EM & nuclear segmentation (U-Net and StarDist).

All told this is a nice paper that makes a valuable contribution to this space. Reproducibility is an issue that computational biologists have been dealing with for a long time, and it is pleasing to see so many methods reproduced in one location. That said, I have some suggestions that I think would improve the clarity of the presented work and make it easier for readers - both novice and expert - to navigate this rapidly moving landscape. I describe these suggestions below.

We thank the reviewer for their valuable suggestions and constructive criticisms which helped us improve our manuscript significantly. In addition, we would like to thank the reviewer for listing additional preprints for us to consider. It is stimulating to interact with others who also value preprints. We now discuss and reference the suggested work in the present revised manuscript.

We would just like to add one clarification to the statement above (which may simply be semantic). With ZeroCostDL4Mic, we do not provide pre-trained models but rather provide Jupyter notebook implementations that are optimised for google colab, for a range of neural networks to enable easy training. We provide, along with that, a set of example datasets that can be used to generate models or test the framework.

Below is an overview of the many changes that we made to our manuscript following the excellent comments made by the reviewers:

- We significantly revised the text to better include the mentions of pre-existing work that were suggested by the reviewers, notably prior use of Colab and a more exhaustive representation of the DL deployment landscape for bioimage analysis. This is particularly developed in the introduction.
- We migrated a lot of the information previously in SI into the main text to discuss each implemented network as well as our quality control approach and transfer learning/data augmentation. This gives a more in-depth overview of the current capabilities of ZeroCostDL4Mic.
- We added 5 new supplementary tables that characterise in detail the performance, limitations and range of use of Colab for the DL network that we provide (in particular evaluating the breaking points, and both training and inference speeds). This also helps placing our approach in better context with other platforms.
- We demonstrate the possibility of using our notebooks beyond Google Colab, notably using Deepnote.

R1.1. -Novelty. One of the most significant issues with this paper is lack of novelty. This is acknowledged with respect to algorithms, as all of the presented methods have been published previously. But it is also true with respect to the underlying software architecture. Using Google Colab as a means to distribute computational methods has been done before in other contexts (the most notable example that comes to mind is kallisto bustools by Pachter and colleagues - <https://www.kallistobus.tools/tutorials>). The schema that the authors have set up (Google Colab, Google Drive, and GitHub) is also one that is frequently used in the data science community.

We thank the reviewer for pointing out these excellent tools developed by Pachter and colleagues, they are now highlighted in our revised manuscript (ref. 32 in the manuscript). We also clarify that Google Colab is indeed frequently used in the data science community. It is, however, not yet extensively used **to analyse microscopy data**.

In the results section:

“By using Google Colab, ZeroCostDL4Mic provides free access to the high-performance computing resources needed to run the broad range of DL networks implemented here (Fig. 1). Google Colab is widely used in the data science community for developing DL projects^{31–33}. However, to productively make use of these resources, users typically need to possess expert knowledge which has drastically limited its uptake by the biomedical research community. By establishing a user-friendly and efficient interface with Google Colab, we aim to leverage this free and highly accessible cloud-computing system to deploy state-of-the-art DL models for microscopy.”

We also point out that one of the main novelties behind our approach is that we provide self-explanatory notebooks with complete workflows (now described in Supp Fig. 2) to train and use a wide variety of DL tasks: install the various computational components, load the training dataset, train a model using custom data, quantitatively validate the performance of the model and deployment on new data. ZeroCostDL4Mic also enables researchers to improve their understanding of DL, experiment with optimising DL parameters and choosing appropriate networks for a specific application. We believe that these steps are essential to both exploit the benefits and understand the limitations of DL approaches in research. Additionally, the community is quickly taking up the approach and, already, some developers are providing ZeroCostDL4Mic-inspired Google Colab notebooks of their DL networks (i.e., Speiser et al. 2020 bioRxiv; Khadangi et al. 2020, bioRxiv). This, to us, is a testimony of the importance of our work.

R1.2. Google Colab notebooks for two of the author's methods (pix2pix and cycleGAN) are already present within the TensorFlow documentation, guides for connecting google drive to Colab instances exist within Google Colab's documentation, and numerous blog posts exist describing how to link Colab, Drive, and GitHub to create a development environment. That's not to say that any of these pre-existing works nullify the author's contribution. But I think it would be more accurate to frame this work as bringing existing data science workflows into the life sciences as opposed to creating a brand new conceptual framework for writing software.

We fully agree with the reviewer, it is precisely our intention to port these approaches to the life science community. We have now amended the text to provide additional context and to make this point clearer.

In Discussion,

“By bringing previously validated methods into a streamlined format that allows easy, cost-free access and customised DL use for microscopy data, we believe that ZeroCostDL4Mic provides an essential step towards broadening the use of DL approaches beyond the community of computer scientists to the biology laboratories that generate the imaging data. We hope to make DL available to all researchers regardless of their laboratory's scale and means. We believe that this democratisation is vital for the acceptance and validation of DL methods in biomedical research.”

Just to clarify, for pix2pix and cycleGAN, we have implemented their PyTorch versions as described by the original authors and not the TensorFlow versions (<https://github.com/junyanz/pytorch-CycleGAN-and-pix2pix>).

R1.3. -Context. I think the paper could do a better job of placing its work in context of ongoing community efforts. The paper briefly mentions other software efforts (ImJoy, ilastik, and ImageJ), but this is an incomplete cross section of this space. Following the

bioinformatics traditions, several groups have made pre-trained models available through web portals (e.g. CellPose from HHMI, NucleAIzer by Horvath and colleagues, and DeepCell from Caltech). There is also CDeep3M from UCSD which allows users to train and deploy models on an AWS instance. Briefly mentioning these other works would allow readers (and potential users) to better place this work in context and understand how the author's software differs - both in positive and negative aspects.

We thank the reviewer for pointing out these missing pieces in the context of our work. We have now greatly expanded the introduction and the discussion to further describe many of the relevant tools that have been and are being developed to facilitate the use of DL in bioimage analysis. We now cite the suggested articles in the manuscript.

In Discussion,

"ZeroCostDL4Mic complements current community efforts to simplify access to DL in microscopy. Other platforms, however, suffer from either a lack of training capacity (StarDist ImageJ plugin, DeepImageJ¹⁹, CellPose¹⁸, NucleAIzer²², Ilastik¹⁵), a narrow focus on a single task (i.e., image segmentation with CDeep3M¹² or DeepMIB¹³) or rely on local servers or paid-for services (as the authors of U-Net have implemented¹¹, or with ImJoy¹⁴). We believe that ZeroCostDL4Mic fills these gaps and enables affordable and versatile DL deployment capabilities."

In addition, in Fig. 1 we now provide a scheme that, we hope, helps put in context the various ways with which DL models can be generated and utilized.

R1.4. -Vision and Limitations. One of the more serious issues I have with this paper is the author's vision for the deep learning/biological imaging interface. I don't think this is well communicated (although it could be inferred from the paper), and this lack of clarity leads to significant blind spots in the paper with respect to the software's limitations. Based on my understanding of the author's work, a fair view of this vision is that users would collect training data specific to their own project, oversee its annotation, train a model on Colab, process existing data with this trained model on Colab, and download this trained model for later use. The presence of pre-trained models allows users to use transfer learning to speed up the model training process and possibly yield more accurate models (although the utility of transfer learning for model accuracy is in question - see for instance <https://arxiv.org/abs/1805.08974>).

This vision centers around a single user's needs and has a single person (or a small number of people) overseeing the effort. This vision may make sense for some use cases. For instance, for reconstruction methods like CARE, it is likely best that users train models on their own data.

This is indeed a key point. In order to clarify the description of our vision of the work and its limitations, we added extensive descriptions of the context within which we think that our work will be valuable, and where it might appear limiting. In particular, the following paragraph in the main text is key here:

In results,

“Considering the resources available with Colab, we believe that ZeroCostDL4Mic is well suited for:

1. Prototyping image analysis workflows and pipelines without financial investment.
2. Executing small to medium size projects (a few 10's of GB of data) compared to large scale projects often encountered in machine vision research.
3. Short term projects not requiring a permanent investment in DL infrastructure.
4. As a resource for DL enthusiasts and students to learn about DL methods and state-of-the-art architectures, such as U-Net^{2,23} or (generative adversarial networks) GANs^{26,27}.

However, larger-scale (> 20 GB of data) and longer-term analysis pipelines may benefit from the investment in paid-for cloud-based platforms (like Paperspace³⁴, Amazon Web Services (AWS)³⁵ Deepnote³⁶) or local infrastructure, therefore tuning the resources to the needs of the specific in-house application. For these cases, ZeroCostDL4Mic is easily adjustable to run outside Google Colab (see Supplementary Note 6 for running ZeroCostDL4Mic notebooks within Deepnote and Supplementary Fig. 4).”

Additionally, we would also argue that the use case that the reviewer describes here (single-user and small team applications) in fact represent a significant number of applications in the biomedical research field. These applications will be well catered for by ZeroCostDL4Mic. This assessment has been corroborated by all author collaborators and many users from the biology field who have been in touch with us.

R1.5. I believe this vision has several issues that I feel should be brought to the author's attention. First is the issue of data. Deep learning is driven by data and it is impossible to develop models without it. An under-appreciated fact is that the difficulty of creating annotated training data has actually shaped methods development. While some tasks like image restoration or image-to-image translation have an intrinsically low annotation burden (they simply require collecting pairs of images), the same is not true for tasks like segmentation. With that in mind, I find it surprising that there's little mention of the data challenges involved in model development. To maximize user adoption, I feel there needs to be some discussion of the data challenges surrounding deep learning and some solution offering, even if it is pointing users to existing annotation tools developed by previously mentioned groups (e.g. ImJoy, CellPose, DeepCell, etc).

We fully agree with the reviewer on this important point. Now that tools such as ZeroCostDL4Mic enable users to easily train their own DL network, the main challenge is data annotation and curation, just as it would be with any platform/solutions that enable training regardless of the approach taken. We now explicitly highlight the burden of human hours due to annotation/curation of data in our discussion, in order to ensure that users have a better understanding of the requirements for appropriate network training:

"ZeroCostDL4Mic provides users access to free computational resources to train DL networks. One remaining challenge lies in handling, curating, and annotating datasets, especially when performing segmentation tasks: the appropriate preparation of training datasets is always associated with human hours cost, disregarding the platform used for training DL networks. However, we would like to highlight that creating segmentation training datasets can be significantly accelerated by initially training models first with a small number of images^{69,70}. These models can then generate masks that can be refined by users (i.e., in Fiji¹⁷) before being used for training to obtain a high-performance model^{5,7,8,16,19,71}. This bootstrapping approach can be carried out iteratively, thereby increasing the network performance progressively while increasing the amount of training data available⁵⁹."

R1.6. Given that training datasets change over time and that models are derived from said datasets, there is a need for some form of model versioning for model developers, much like versioning is required for writers of software. Absent versioning, users will find that a myriad of issues (poorly documented models, no chain of custody between data and models, over writing pre-existing useful models, etc.) will arise in a short period of time. That's not to say the authors need to solve all these problems for this paper to be a valuable contribution. But the paper's blind spot to the data challenges surrounding using deep learning in practice does impact how useful users will ultimately find this software.

This point is well taken. To improve on the issue of versioning and help users keep track of models and training sessions, we now provide an automatic export of a detailed model

report for each trained model saved as a PDF file that contains all parameters used for training (e.g., the type of data, network parameters, essential package versions) and the model performance (assessed via the quality control section). This logging allows users to easily keep track of parameter changes and training data modifications during model optimisation. Importantly, in order to contribute to setting good practices on reporting DL model training in the literature, this report is human-readable and can be included as-is in a typical *materials and methods* section.

To further address the reviewer's concern about versioning transparency, we have now provided all notebooks with a version number which is automatically compared to the latest release of ZeroCostDL4Mic. When using the notebook the user will now see a message notifying them of whether they are using the latest version or not.

In addition, to bring this issue forward we decided to highlight it in our discussion.

"Another challenge associated with DL is enabling model versioning to ensure reproducibility. Indeed, a DL model's performance is affected by the training dataset, the network and training parameters but also all the underlying dependencies. To mitigate this issue, in ZeroCostDL4Mic, we provide each trained model with a thorough report that contains all parameters used for training (the type of data, network parameters, essential package versions) and the model performance (assessed via the quality control section). This logging allows users to easily keep track of parameter changes and training data modifications during model optimisation. Importantly, to contribute to setting good practices on reporting DL model training in the literature, this report is human-readable and can be included as-is in a typical material and method section. "

R1.7. Also problematic is the issue with the scale of training data. Deep learning methods work best for big data; for most tasks, the same model architecture trained with the same algorithm for the same number of steps on a larger, more diverse dataset will see a performance boost. Big data comes with a price however, and for this work the most relevant price is memory. Google Colab instances are limited to 12GB of memory; purchasing Colab Pro (9.99 USD/month) can boost this to 25GB but this nuance can be ignored given the constraint of zero cost. Practically speaking, that means that the author's software is incapable of taking advantage of big data due to memory limits and that models that result from it will have suboptimal performance.

To give more details about what practically limits the use of Colab for our applications, we now provide supplementary tables showing the maximal size of the training datasets that can be used to train the various DL networks provided in ZeroCostDL4Mic (See Supplementary Tables 5 and 6, respectively for training and inference). Of note, all these calculations were performed assuming that 12GB of RAM is available, but, during our experimentations, we also often got allocated sessions with 25GB of RAM without purchasing a pro account.

Here we would, however, like to present a different point of view than that described by the reviewer. Indeed, large and diverse training datasets are required to generate models that achieve top performances at generalizing to new data. This is the idea behind projects such as Cellpose or NucleAIzer, where trained models are generated to work on as many types of images as possible. Users can then use these models directly on their data without training. We agree that generating these DL models is likely to require resources that are beyond the capabilities of Google Colab (due to the size of the training dataset). However, with ZeroCostDL4Mic we enable users to train DL models designed for a specific purpose using their own data. Users can then deploy their model on data which is new but similar to the data it was trained on. This relaxes the requirements on how largely generalizable the model needs to be, as long as performance within the boundaries of the intended applications can be demonstrated (as we ensure via QC). We would argue that this is a common use-case for typical users we envision (see our response to R1.4) since they will require their models to perform well for a given type of dataset, not necessarily a wide range of data. When training such specific models (as opposed to those general models that the reviewer may be making reference to), much smaller training datasets can be used. For instance, we never use more than 80 training images to retrain StarDist (typically 40-60 images can yield excellent performance). We would thus argue that Google Colab is very well suited for such tasks.

R1.8. The memory limits of the available GPUs also form a similar constraint, given that mixed precision training is not available for the GPUs available to users. This means users are limited to models with smaller capacity and training them with smaller batch sizes, both of which can limit performance. Datasets for which memory is an issue already exist - two that come to mind are DeepCell's live cell imaging dataset (Caltech) and the PanNuke dataset for nuclear segmentation and cell type detection in H&E images (University of Warwick). The vision issue surrounding this limitation is simple - who is this software for? Is it for novice users or experts? The former may be less concerned with these issues (although they will certainly be impacted) but the latter almost certainly will be. I find the discussion in the supplement inadequate - the limits Colab's hardware options place on the kinds of models that users can develop makes me skeptical about the impact the author's claim this work will have.

As detailed in the previous comment, we believe that ZeroCostDL4Mic is very well suited to cater to single studies from the microscopy bioimaging community dealing with small to medium-scale projects (a few 10's of GB of data). We would argue that 1) in biological studies, these cases are the norm rather than the exception (very few biomedical research institutes have access to dedicated core facilities that can help with image analysis) 2) they are the one who would benefit the most from tools that are easy to use (experts can easily adapt available python tools to their own need).

To make the limitations of the hardware available in ZeroCostDL4Mic more transparent to readers/users we have also added a table (Supplementary Table 5 and 6) outlining the dataset size limitations for each notebook given the GPUs available.

As described by the reviewer, larger-scale (> 20 GB of data) and longer-term analysis pipeline would benefit from platforms that are more powerful than Google Colab such as paid-for cloud services (i.e., Paperspace, AWS, or Deepnote) or local infrastructure. In these cases, the ZeroCostDL4Mic notebooks we provide can still be very useful as they can easily be ported to any platforms that support Jupyter notebooks. For instance, we demonstrate that the ZeroCostDL4Mic Deep-STORM notebook can be used on Deepnote (see new Supplementary Note 6 and Supp. Fig. 8).

We now explicitly highlight the main use-case where ZeroCostDL4Mic will be most valuable:

“Considering the resources available with Colab, we believe that ZeroCostDL4Mic is well suited for:

1. Prototyping image analysis workflows and pipelines without financial investment.
2. Executing small to medium size projects (a few 10's of GB of data) compared to large scale projects often encountered in machine vision research.
3. Short term projects not requiring a permanent investment in DL infrastructure.
4. As a resource for DL enthusiasts and students to learn about DL methods and state-of-the-art architectures, such as U-Net^{2,23} or (generative adversarial networks) GANs^{26,27}.

”

R1.9. The same issue arises for inference. How much data can one process using Colab? This is unclear. While instances are supposed to remain active for 12-24 hours, a number of issues limit throughput. The type of the available GPU is stochastic - assuming one is available - and the ones that are available limits the options available to increase inference speed (e.g. performing inference at reduced precision like float16 or int8). Moreover, Google Colab throttles heavily active users - even for the pro version - to ensure accessibility for the broader community. The author's do not investigate the inference rates achievable with their software. Given the ever increasing size of imaging datasets, I think such an analysis is necessary.

We agree it is very useful for users to get an idea of inference rates so we now provide as supplementary tables (Supplementary Table 6) the maximum number of images that can be processed (inference) in a single Google Colab session as well as the typical inference speed obtained using different GPUs or CPU. Here the main limitation of Google Colab is the Gdrive space provided for free. Still, thousands of images can typically be processed.

R1.10. Moreover, I don't think it was ever Google's intention to pay the computing costs for the entire bio-image community, which is what the authors are effectively suggesting.

We cannot speak for Google, but we expect the tool to remain stable as it sets a reference for an important community and is an essential promotional tool for them. For instance, Google is slowly expanding the geographical area where Colab Pro is available (just became available in Canada). That said, even if Colab were to close down tomorrow, our platform would still be highly valuable as it can be adapted (with very minimal changes) to run on any platform that supports Jupyter notebooks (as demonstrated on Deepnote). Furthermore, we have clarified which purpose we believe Colab can serve in the bioimaging community and where alternatives could or should be used to overcome limitations. (see e.g. our responses to R1.3., R1.4., R1.7. and R1.8.)

R1.11. I think most of these issues could be resolved with a more thoughtful discussion and analysis of the limitations posed by the author's software architecture. Some discussion is found in the supplement, but I believe this merit's a discussion in the main text.

As described above we have now considerably extended the main text (including the introduction and the discussion) and we thoroughly characterized the limitations of the platform (see Supp. Tables 3, 4, 5, and 6).

R1.12. -Accuracy. Related to the previous point, the authors make an argument in Supplementary Note 1 and 5 that models are generally specific to the acquisition instrument and that they often don't generalize across microscopes. While generalization is an issue, the claim that generalization can't be achieved is untrue, as is the claim surrounding the computational power required to train said models. Work in the literature has shown that image normalization (<https://journals.plos.org/ploscompbiol/article?id=10.1371/journal.pcbi.1005177>) and diversity of training data (<https://www.nature.com/articles/s41592-019-0612-7>, <https://www.biorxiv.org/content/10.1101/2020.02.02.931238v1.abstract>) and image resizing (<https://www.sciencedirect.com/science/article/pii/S2405471220301174>) is sufficient to achieve generalization. Generation of annotated training data that spans cell types and acquisition instruments is the most difficult aspect of this work, but it is achievable and the models derived from these efforts can usually be used safely. While this has yet to be proven true for tasks like image restoration (which are more ill-posed), it is certainly true for cell segmentation tasks. The pervasive deployment of deep learning models on data generated from mobile devices, which have a diversity of cameras, is further proof of this point.

We agree with this assessment and thank the reviewer for highlighting further valuable literature around the topic. It is true that it is possible to achieve generalization if enough training data is available. So we have now toned down this aspect in the text. In the case of segmentation, well-generalising models are indeed especially useful for cells and nuclei (some of the StarDist pre-trained models already perform well in a wide range of cases).

However, this type of general model is unlikely to be able to cover the broad variety of data type that need to be performed when analyzing biological images (fluorescent, EM images, etc.). Most of the time, image analysis needs to be carefully adapted to the biological phenomena that are observed.

R1.13. Moreover, the author's statement that training models for these larger datasets is time consuming and resource intensive to the point that most lab's can't afford it is also incorrect. In our hands, training models on larger datasets (e.g. ~2000 512x512 images) can be done in several hours with a modest GPU (e.g. V100) - without performing time saving tricks like training at reduced precision.

Here, we want to point out that a V100 cost around \$10,000. This is not something that a lab focusing on biological experiments would easily be able to invest in and set-up to analyze a specific experimental dataset. Among the extensive community that we have been interacting with on this project, this kind of purchase seems to only make sense in laboratories dedicated to the development of DL technologies or core facilities aiming to provide such services for their users. For occasional users (i.e. analyzing a specific set of experimental data), it seems to be much more appropriate to use free (or paid-for) cloud service rather than invest in local infrastructure.

Again, here we provide a platform that will cater to some but not all applications and dataset sizes/types. We feel that the added text introduces these key points better in this version of the manuscript and therefore would not mislead potential users.

R1.14. The author's assertion about the climate impact of training these models is also incorrect. In addition to being faster, the more modern GPUs are more power efficient, which leads to a lower carbon footprint as well as lower prices (see for example <https://cloud.google.com/blog/products/ai-machine-learning/your-ml-workloads-cheaper-and-faster-with-the-latest-gpus>). Moreover, training in less time means less CPU utilization, which also reduces cost and footprint. Training models on a K80 on Colab is arguably worse for the environment than purchasing time on a more modern machine. The authors are likely confusing the time required for training vision models with the time required for language models. The self-supervised pre-training required by language models is both time and resource intensive.

We thank the reviewers for bringing this interesting point forward. We agree with the arguments listed above and we believe that our point was misunderstood. In the text, we indicate that re-training existing high-quality models using transfer learning had a lower carbon footprint compared to training new models from scratch. This is because the training time is considerably shorter, in perfect agreement with the above comments.

R1.15. -Cost. Is “ZeroCostDL4Mic” actually free? I would argue no. The notion that the cost is zero ignores the cost of human labor throughout the entire model development process. There is the cost involved in dataset annotation (which can easily cost >10,000 USD based on the application) and the cost incurred by using suboptimal models (whose results will require curation). There is also the opportunity cost of running training and inference jobs on out-dated hardware, as having models and model results sooner.

We fully agree with the reviewer and as indicated above, now that tools such as ZeroCostDL4Mic enable biologists to easily train DL models, the next challenge is the data annotation and curation. This comment is directly linked to the previous comment R1.5 and as highlighted above, we have now included these points in the discussion section.

In Discussion,

“ZeroCostDL4Mic provides users access to free computational resources to train DL networks. One remaining challenge lies in handling, curating, and annotating datasets, especially when performing segmentation tasks: the appropriate preparation of training datasets is always associated with human hours cost, disregarding the platform used for training DL networks.”

R1.16. While the authors may disagree, the paper in its current form does not present an argument to the contrary. Moreover, in my view the current manuscript does not make a compelling case for Colab over a more proper cloud infrastructure, particularly given that preemptible GPU instances can be had for as little as 0.1 USD/hour. There is no analysis of the actual cost of producing a deep learning model in the cloud (provided training data is available) or the cost of performing inference. How much does training a single deep learning model in the cloud cost? How much is the cost of analyzing a single image? Is that price worth the limitations imposed by Google Colab? Only a prospective user can say, but having that information present would let them make an informed decision about how the author’s work can meet their own analysis needs.

It is true that numerous cloud computing platforms are available online with fewer restrictions than what Colab provides (AWS, paperspace for instance). But it remains nonetheless true that Colab is entirely free and readily accessible without assessing, agreeing and setting up a payment plan with the companies involved. This remains a major advantage for the labs that we’ve engaged with and that have engaged with us: being able to go from reading the paper to training a model in ~1h, by researchers and students alike, without the worry of expensive and varying monthly bills (which many departments have issues dealing with for accounting purposes). All in all, we feel that a free and one-click access tool really has a major advantage for the initial assessment of DL methods and the other applications that we already described in the comments above. As a comparison, one alternative solution is using Aivia Web, which provides similar capabilities as

ZeroCostDL4Mic (although for fewer DL tasks) and which costs £5,000 per year. It is very challenging to assess how much a training session costs on other platforms such as AWS or paperspace because we cannot evaluate the pricing, CPU time, time required for model optimisation etc. In addition, paid for Google Colab alternatives would still require notebooks such as the one we provide with ZeroCostDL4Mic to run.

In terms of time cost, we now provide more detailed information on the performance of different GPUs available in Colab for training and inference in the added supplementary tables (Supp. Tables 3 and 4)

R1.17. Other issues with the paper that should be addressed:

-The memory footprint and inference speed of the models should be reported. Given that there is variance of GPU type (K80 vs P100, etc), this should be reported for each GPU type, not just the type that was available to the authors at benchmarking time.

We now provide this as Supplementary Tables 3 and 4.

We provide these benchmarks for the P100 and T4 as well as for CPU only. We could not perform these benchmarks with the other GPU sometimes available via colab as in the last two months we could not access them regularly, despite multiple attempts.

R1.18. -The maximum dataset size permitted by the CPU and GPU memory limits of Google Colab should be reported for the model architectures used.

As already detailed above, this information is now detailed in Supp. Tables 5 and 6. The limit here is typically dependent on the RAM provided by Google Colab and is independent of the GPU type. Of note, all these calculations were performed assuming that 12GB of RAM is available. However, during our experimentations, we also often got allocated sessions with 25GB of RAM without the need to purchase a Colab Pro account.

R1.19. -While model chaining is something that is desirable, it is not new. The examples the authors give (translating label free images into nuclear images and subsequently performing segmentation and tracking) has been described previously (<https://www.biorxiv.org/content/10.1101/2020.03.05.979419v1.abstract>). Moreover, the chaining of segmentation and tracking models has also been described previously (<https://www.biorxiv.org/content/10.1101/803205v1>). This is a functionality allowed by the presence of multiple performant models in the same programming environment, rather than something unique to the author's software - this should be made clear. It is also unclear how the authors plan to make model chaining accessible through Colab in a way that does not involve programming.

We thank the reviewer for pointing out these manuscripts to us. We now cite them (ref 60 and 61). Model chaining can be simply used in ZeroCostDL4Mic by using our notebooks sequentially which by default requires no coding. It is indeed advantageous to have all the data and models in the same environment (Google Drive) to do this, without the need to migrate data across platforms.

R1.20. -The ability to use semi-synthetic data for training image restoration models, as pioneered by Manor and colleagues (<https://www.biorxiv.org/content/10.1101/740548v8.abstract>) would likely make these methods more accessible.

We also use synthetic data to train Deep-STORM for which we built a realistic single-molecule localization microscopy data simulator directly within the notebook to automatically generate the training model. Here, we also provide an extensive description of how to use the simulator and how the parameters relate to physical processes, as described in our Wiki¹.

Regarding the denoising/image restoration models, we aim to implement the original neural networks as they were developed by the authors. If the training dataset is too small, data augmentation strategies (available in the notebook) can be used to improve the results.

This is however an important point and the use of synthetic data for DL training remains an active area of research. When new approaches become available, we will rapidly incorporate these into our notebooks.

R1.21. I found the notebooks presented rather clumsy for the purpose of using a pre-trained model. So much so that if given the choice between the Colab or a web portal/ImageJ plugin, I would use the latter option. An alternative could be to have notebooks solely for inference with a pre-trained model where appropriate.

To clarify, ZeroCostDL4Mic does not aim at providing pre-trained models but rather to help users train their own models, therefore we established that our current workflow was the most appropriate for users in this situation. This was also corroborated by all of our beta testers at various stages of the elaboration of the project. Their feedback helped us make ZeroCostDL4Mic as user-friendly as possible. The idea of using a single notebook for the whole workflow came from interacting with them. It is also important to note that for most DL networks we provide, once a model has been trained, it can be downloaded and used to perform inference locally using Fiji (using CSBDeep plugin or DeepImageJ).

¹ <https://github.com/HenriquesLab/ZeroCostDL4Mic/wiki/Deep-STORM>

R1.22. -The author's should investigate Tensorflow Hub as a storage location for pre-trained models to increase their accessibility.

This is an excellent point. As we do not directly provide pre-trained models we will not store them directly on Tensorflow Hub (also several of our neural networks rely on PyTorch so we prefer a platform that is agnostic of the backend used). However, we are working very closely with the bioimage.io (<https://bioimage.io/#/>) model zoo initiative to share our notebooks and training datasets. If we decide to provide pre-trained models in the future, it will likely be via the same platform. Some of our trained StarDist models are also available on Zenodo.

R1.23. -It would be nice if benchmarks for pre-trained models were available on the project's front page.

We apologize but we are unsure what the reviewer is referring to here. Again, we do not provide pre-trained models, but notebooks for training models. Although providing models is something we are looking into (as discussed in the comment above), this is not the main and most important advantage of our platform.

R1.24. -I found the benchmarking methods presented in this paper to be somewhat confusing and underwhelming. The SSIM measure for image restoration can be biased by the presence of background pixels. It seems to me that a better alternative would be to compute it only for foreground pixels - although I could be mistaken.

We used the SSIM metric for our quality control section in several notebooks because it is widely used and accepted in the literature in the DL field for microscopy, notably in CSBDeep and because it is one of the gold-standard methods in image processing (the SSIM paper by Wang et al., 2004 has been cited over 17,000 times). We would argue that the distinction between foreground and background pixels is not trivial and might introduce bias into the calculation of the metric and is also not usually done when applying SSIM (see Supplementary Note 2 for more detailed information on how data is normalised and processed when using SSIM). Additionally, we would like to point out that we provide two additional direct quality control metrics in the form of root-squared error (RSE) and peak signal to noise ratio (PSNR) as alternative metrics for the user to consult when assessing their models.

R1.25. The use of mAP and IoU for benchmarking segmentation is also unclear. Is IoU computed on a per object basis, or just in aggregate over all the pixels over foreground. The

mAP can also be a misleading measure of segmentation performance, as it includes information about precision at cut off values that are not used in practice. A more informative approach I would suggest is the F1 score, augmented by statistics about error types (splits, merges, etc). This was first reported by Caicedo and colleagues (<https://onlinelibrary.wiley.com/doi/full/10.1002/cyto.a.23863>) and expanded on further by Moen and colleagues (<https://www.biorxiv.org/content/10.1101/803205v1>).

The answer here depends on the network in question. We are unsure what specific network the reviewer is referring to here as the mAP score is calculated only in YOLOv2 notebook while the manuscripts cited above are perhaps more relevant for nuclear segmentation (for instance StarDist).

In the U-Net notebook IoU is performed on the whole image as the objects are not separated. In StarDist, in the previous version of the manuscript, IoU was also performed on the whole image as it is a good indication of the segmentation quality as a whole. Now we have also included other metrics so that users can also better appreciate the quality of the segmentation on a per-object basis. These metrics include 'false positive', 'true positive', 'false negative', 'precision', 'recall', 'accuracy', 'f1 score', 'n_true', 'n_pred', 'mean_true_score', 'mean_matched_score', 'panoptic_quality' and are detailed in SI. In the YOLO notebook IoU is calculated per-object bounding box which is only used to determine whether each predicted object has enough overlap with a ground-truth object to be considered a detection. In the previous version of the YOLOv2 notebook, the average precision (AP) per object class and the following mAP were calculated. Given the valid comments by the reviewer regarding limitations of the AP metric, we have updated our YOLO notebook so that the F1 score and several other metrics ('false positives', 'true positives', 'false negatives', 'precision', 'recall' and 'accuracy') are also included for each object class.

R1.26. -The pip commands in the notebooks that install the packages do not appear to be versioned. This is problematic, as versioning is an essential part of reproducibility. A model trained under one version of TensorFlow can produce different results when inference is performed in a different version. I would recommend versioning if possible, which can be done by specifying them in a requirements.txt file. Docker would be the gold standard, but it is not clear to me how to make Docker work in the context of the author's framework.

With ZeroCostDL4Mic we aim to provide the best possible experience and performance to our users and this often involves the use of the latest version of available packages (to benefit from the latest features and performance improvements). However, as indicated by the reviewer, versioning can then become an issue to reproduce specific results. To address this and as already described above, we now provide each trained model with a detailed PDF report that contains all parameters used for training including essential package versions, and the model performance (assessed via the quality control section). This logging allows users to easily keep track of parameter changes and training data

modifications during model optimization. Importantly, in order to contribute to setting good practices on reporting DL model training in the literature, this report is human-readable and can be included as-is in a typical material and method section.

R1.27. -Computation precision (float32, vs float16, vs bfloat16) may be a reason why the authors see poor performance on TPU devices. This likely merit's further investigation.

All computations are performed using the precision and architecture chosen by the original authors. As we see good performances using GPU and as we are not aiming to use TPU to train these networks we did not investigate this point further.

R1.28. -The inductive bias of object detection models (e.g. positing that cells can fit inside an unrotated bounding box with limitations on the aspect ratio) is likely the origin of their poor performance. Investigation of other object detection methods (e.g. RetinaNet or a more modern YOLO model) is warranted. Also, the authors may want to make use of a resource generated by a recent CVPR paper (https://openaccess.thecvf.com/content_CVPRW_2020/html/w57/Anjum_CTMC_Cell_Tracking_With_Mitosis_Detection_Dataset_Challenge_CVPRW_2020_paper.html)

This is an excellent point. ZeroCostDL4Mic is an ever-growing platform and we are indeed very interested to implement other object detection methods in the near future (retinaNet, YOLOv4, and Detectron2 are on our list for instance).

Reviewer #2 (Remarks to the Author):

R2.0. The authors present a software library (in fact a collection of existing software packages) which use deep learning (DL) for different aspects of image analysis. By bringing together disparate packages and unifying them to have a common interface (via Jupyter Notebooks) and implementing them to run on a common computing platform (Google CoLab), this work aims to simplify and illustrate the use of DL for image analysis for researchers who may not be familiar with the technical details of these methods.

In my view, this work is a very positive development for the microscopy field, and will accelerate the uptake of DL methods, leading to new biology results and the refinement / development of further image analysis methods. A quick look at the github page for this project shows a well-organized software library with an active userbase – this is already a strong validation of the concept the authors have developed in ZeroCostDL4Mic.

We thank the reviewer for their positive and valuable comments.

R2.1. The choice of Google CoLab as a computing platform is the one aspect of the work which I would ask the authors to address. Commercial enterprises such as Google do offer free services such as CoLab, but platforms such as this may be withdrawn at any time if they are not profitable, or if the company decides on a different strategy. Therefore, I would ask the authors to consider the question of what would happen if, tomorrow, Google decided to shut down the CoLab service. Would the ZeroCostDL4Mic library be usable? Would the publication in Nature Communications still have value, or would it become obsolete when the CoLab service ends? Could ZeroDLCost4Mic be run in the same way on a different computing platform? It could be that the authors have already answered these questions and I have missed it, but those are the only significant points that I wish to raise in this review.

We thank the reviewer for bringing this important point forward. Yes, we believe that our platform would still be highly valuable even if Google Colab would shut down. Indeed, ZeroCostDL4Mic can be adapted (with very minimal changes) to run on any platform that supports Jupyter notebooks. For instance, we now show that our Deep-STORM notebook can be used on Deepnote. Detailed information on this is available in Supp. Note 6 and Supp. Fig. 4.

Supplementary Fig. 4. Using Deepnote to run a ZeroCostDL4Mic notebook. Screenshot highlighting the different steps of the DL workflow running within Deepnote (<https://deepnote.com>).

R2.2. A minor point is also that the CoLab service is indeed "Zero cost" when free resources are available, but for larger tasks users may be required to sign up for "CoLab Pro", which is a pay-for-use service offered by Google. I feel that this point is worth mentioning, so as to avoid giving the impression that unlimited free image processing resources are available from Google.

All the limitations are clearly stated and described in Supp. Note 5. To further emphasize this as a response to the reviewer's comment, we have also now expanded the part in the main manuscript where we describe in detail the resources provided by Google Colab (for free). In addition, and as requested by Reviewer 1, we provide as supplementary tables the maximal size of the training datasets that can be used to train the various DL networks provided (Supp. Table 5). We also indicate the maximum number of images that can be analyzed once a DL network has been trained (Supp Table 6).

R2.3. Typos etc.:

We thank the reviewer for reporting these typos and other issues.

1. Main text line 161: change "allowing to" to "allowing one to" or equivalent

2. Main text line 188: Missing ")" after "Video 2-11".
3. Main text line 200: Missing "," after "ilastik". Start new sentence after "Fiji/ImageJ".
4. Main text line 231: Add footnote including link to Zenodo.

These have now been corrected

5. Online methods line 370, 375: Acronym "FBSR" not defined. "SRRF" also not defined.

This has now been corrected

6. Supplemental information line 172: "EM" not defined.

This has now been corrected

R2.4. This manuscript presents a new and valuable image analysis resource. Works such as this, which simplify and make accessible pre-existing methods, is often not recognized in the literature even though they can be as important as the initial development of the methods themselves. I recommend this work for publication, with minor revisions as described above.

We thank the reviewer for their positive and valuable comments.

Reviewer #3 (Remarks to the Author):

R3.0. In their manuscript "ZeroCostDL4Mic: an open platform to use Deep-Learning in Microscopy", the authors ported a set of tools, which use deep neural networks for image processing task to notebooks to be used in Google Colab. The software can be run using the GPU infrastructure provided by Google Colab which is provided to the public for free. The authors distribute their notebooks as open source through a publicly accessible GitHub repository and placed an emphasis on documentation of the different notebooks to guide new users through the steps.

R3.1. I find the GitHub repository is providing a set of useful tools for the biomedical community. The steps to accomplish the results and to use the tools are aimed to be clear and I expect it will result in a good acceptance of the tools. In its current form the manuscript reads like an advertisement rather than a scientific manuscript. I think this could be improved by focusing on benchmarking important features and demonstrations.

We thank the reviewer for their valuable comments. We have now largely rewritten the main manuscript to take these comments into account. A large part of what used to be in supplementary information has now been transferred and merged to the main text. This includes more information on the ZeroCostDL4Mic framework, on Google Colab and its limitations as well as a description of the various DL tasks available. As a result, the manuscript now has a more traditional structure highlighting the methods, limitations and application showcasing.

R3.2. The odd structure of the manuscript makes it sometimes difficult to find all the information about the contributions and some new developments from the authors, who I believe sometimes added features to the different networks.

In the new version of the manuscript, we have also focused on highlighting the novel features that were added to each notebook. We hope that this information is now easier to find.

R3.3. To facilitate reading the manuscript, it would be very helpful to have the main text subdivided into the different networks and an have evaluations of how the networks fare on Colab.

We thank the reviewers for this excellent suggestion. As indicated above, we have now divided the main text into the DL tasks that are available within ZeroCostDL4Mic. We have also included several new Supplementary Tables that detailed the networks install, training, and prediction times (Supp. Tables 5 and 6).

R3.4. My major concerns are regarding the scientific presentation of the work and the evaluations performed so far. The manuscript is often hyperbolic and many arguments made would need quantifications or more thorough research.

We have now substantially modified the main manuscript to address this point. In addition, we provide an extensive characterization of the performance and breaking points of the various DL networks provided.

R3.5. The limitations of Google Colab are only mentioned at the very end of the supplement, which would mean many readers will not be aware of those. It should be made clearer to the readers that the free offerings come with several limitations.

In the new version of the manuscript, we include a full section dedicated to Google Colab. In this section, we describe in detail the limitations associated with the use of Google Colab. We also provide, as supplementary tables, the maximal size of the training datasets that can be used to train the various DL networks included (Supp Table 5). We also indicate the maximum number of images that can be analyzed once a DL network has been trained (Supp Table 6). The Supp Note (Supp. Note 5) dedicated to the limitations of Colab is also clearly highlighted.

R3.6. Major:

1) The description the authors provide about the different Google Colab implementations do not provide enough information for an end-user to decide if a particular notebook is an appropriate solution for a potential end-user. Are the notebooks one-to-one implementations of the previous implementations or are there differences in some cases?

For all the notebooks, we implemented the original network architecture but within our standard workflow that every ZeroCostDL4Mic notebook follows. For many of the networks, we obtained significant support from the original developers in order to ensure that we implemented their work faithfully. Also, the notebook workflow was optimised thanks to the help and support of the beta testers. More specifically, we added the graphical interface, the possibility to perform data augmentation, transfer learning as well as, importantly, quality control. We also added in some notebooks such as Deep-STORM and StarDist extra functionalities. For instance, our Deep-STORM notebook contains an SMLM data simulator (to train Deep-STORM easily), the possibility to export prediction as coordinates files, and the possibility to perform drift corrections. The StarDist notebook can export tracking files that can be used in Trackmate to perform automated cell tracking.

R3.7. 2) What size of data can actually be processed on Google Colab with the different notebooks?

We thank the reviewer for this excellent question. We now provide supplementary table detailed information on the maximum numbers of images that can be used for training or when performing predictions (Supp Table 5 and 6).

R3.8. There are different potential limitations that might differ for the notebooks, like data loader, throughput, paired with disconnection time of Colab. How is the processing throughput on Colab compared to throughput e.g. with a local installation (RAM and CPUs are very limited on Colab)?

As seen in Supp. Table 6, what limits the throughput of our notebooks is the amount of RAM provided by Colab. Of note, all these calculations were performed assuming that 12GB of RAM is available. However, during our experimentations, we also very often got allocated sessions with 25GB of RAM.

R3.9. Do all of the networks require a GPU, for prediction and for training? How is the utilization of the underlying hardware (minimum requirements)?

Only pix2pix, CycleGAN and Label-free prediction (fnet) absolutely require GPU for training and making predictions. However, for the remaining notebooks, performing training using only the CPU is very slow. We now provide quantification of the speed improvement gained by using GPU over CPU in Supp Table 3 and 4.

R3.10. Which also means, may a regular graphics card be sufficient in some of those cases?

A regular graphics card may be sufficient to train specific DL networks such as Noise2Void 2D. However, as seen in Supp. Table 3 and 4, other networks require much more computational power. This is especially true for the networks dealing with 3D datasets. It is important to note that most of these networks currently require CUDA installation to enable GPU acceleration, and this is currently compatible only with NVIDIA graphic cards. It is also important to point out that another advantage of Google Colab is that it provides a clean environment at every session where all the necessary dependencies are installed. This enables users to use the provided DL network out-of-the-box without the need to install any dependencies on their own computers. This is especially important as not all the provided networks require the same version of TensorFlow, CUDA to work, and others require PyTorch or KERAS to function.

R3.11. 3) Related to point 2, it would be important for the authors to explain more about working or processing on Colab. E.g. the installation times of the notebooks? (since they need to be installed each time a notebook is started, the time of each installation should be clearly stated).

We now provide in Supp. Table 3 the installation time for each notebook. They range from 10s to 1 min.

R3.12. 4) The manuscript is based on the assumption that deep learning is great, and the networks here are the ideal choice but the assessment of the performance provided here is limited. Assessments about the metrics of IoU or SSIM are embedded in images (e.g. Supplementary Fig. 9 and Supplementary Fig. 15) rather than written in plain text or spelled out in graphs, which makes it very difficult to find the relevant information when reading the manuscript or to compare with other methods in the literature. Are all of those state-of-the-art IoU values?

We thank the reviewer for bringing this issue forward. We have now included the metrics displayed in each figure in the associated legends. We hope this helps make them more readable. In addition, we have added two new graphs to figure 9 to better highlight the changes in metrics following the use of transfer learning (New figure 9 is copy and pasted below for the reviewers' convenience). The IoU values obtained are indeed state-of-the-art and are similar to those obtained by others. It is important to note that most metric values are specific to a particular dataset. Therefore the values we obtain here are not directly comparable to those published by other authors (using a different dataset). The reviewer raises an important point: the lack of recognized evaluation datasets (one for each task) for analyzing bioimage using DL.

Fig. 9: Data augmentation and Transfer learning can improve performance. (a-c) Data augmentation can improve prediction performance. YOLOv2 cell shape detection applied to brightfield time-lapse dataset. **(a)** Raw brightfield input image. **(b)** Ground-truth and YOLOv2 model predictions (after 30 epochs) with increasing amounts of data augmentation. The original dataset contained 30 images which were first augmented by vertical and horizontal mirroring and then by 90 degrees rotations. **(c)** Mean average precision (mAP) as a function of epoch number for different levels of data augmentation. **(d-e)** These panels display an example of how transfer learning using a pre-trained model can lead to very high-quality StarDist prediction even after only 5 epochs. This figure also highlights that using a pre-trained model, even when trained on a large dataset, can lead to inappropriate results. **(d)** Examples of StarDist segmentation results obtained using models trained using 5, 20 or 200 epochs and using a blank model (“De novo” training) or the 2D-versatile-fluo as a starting point (transfer learning). **(e)** StarDist QC metrics obtained with the models highlighted in **(d)** ($n = 13$ images). The IoU scores are calculated over the whole image, while the F1 scores are calculated on a per-object basis. Results are displayed as boxplots which represent the median and the 25th and 75th percentiles (interquartile range); outliers are represented by dots⁶⁸. Note that the axes of both graphs are cut.

R3.13. 5) Hyperparameters: in most cases I can't find which hyperparameters were used to train the models. E.g. Supplementary Figure 9, which hyperparameters that lead to overfitting and which that did not lead to overfitting?

All the hyperparameters used for training are indicated in Supp. Table 2. We thank the reviewers for noticing that the information related to Supplementary Figure 9 (new Figure 8) were missing. We now have added these parameters in the figure legend.

In the caption of Figure 8:

“The upper panel shows a good fit of the model to unseen (validation) data (main training parameters, number_of_epochs:100, patch_size: 256, number_of_patches: 10, Use_Default_Advanced_Parameters: enabled), the lower panel shows an example of a model that overfits the training dataset (main training parameters, number_of_epochs:100, patch_size: 80, number_of_patches: 200, Use_Default_Advanced_Parameters: enabled).”

R3.14. Other comments:

- The authors mention a common organization theme between the notebooks, which makes it easier for others to implement their network, but it's unclear what this actually implies. Do the authors provide a template for this?

Yes, we provide a template for this purpose (See provided Softwares or our Wiki pages). To further clarify what we mean by a common organization, we now describe this in Supp Figure 2.

Supplementary Fig. 2. ZeroCostDL4Mic notebooks common workflow.

R3.15. - L80-83/L198-214: Better embedding into the current literature would help the readers instead of a blanket statement that this ‘considerable simplifies the use of DL for microscopy’. The approach by the authors is not the first or only approach that provide easy end-to-end solutions for DL.

We have considerably expanded our introduction and discussion. We have now included references to other software/papers that aim to also simplify the access of DL for microscopy (see revised manuscript).

R3.16. - Supplementary Fig. 11: This figure displays an example of how adjusting parameters (batch size, number of steps and number of training epochs) can affect model performance in StarDist nuclear segmentation. Alternative interpretation: If you train enough epochs, the other 2 parameters don’t matter?

This is indeed possible for this particular dataset. We would however be cautious about generalizing this rule. In addition, training for too long may lead to model overfitting and poorer performances. In our opinion this further highlight the importance of having a QC section that allows the assessment of model performances.

How much do those epochs take in wall clock time on Colab? Do they run uninterrupted until the end, or does the user get disconnected from the notebook in between?

We have added a table where the time it takes for an EPOCH to be completed is indicated (for a range of GPU and CPU). In colab, EPOCHs run uninterrupted until the end provided that the overall session lasts less than 12h.

R3.17. - L74-79: I could not find the actual quantification. Some of it could be in Supplementary Fig 15, if so, the metrics should be represented in the text and/or graph and appropriately quantified, rather than imprinted on an image. (Also concerns: L61-74) How much time takes the new or re-training, how much does the IoU improve? Importantly: How much time does it take to (manually?) generate the new training data?

As indicated above, we are now displaying the metric in graphs (Figure 9e). As indicated above, the metrics obtained are now also indicated in the figure legend. The IoU improves from 0.75 (using a pre-trained model) to 0.86 (retraining the model using 5 EPOCH). For the same image, the F1 score improves from 0.82 (using a pre-trained model) to 0.95 (retraining the model using 5 EPOCH) More importantly than the small metric increase is the disappearance of the large-scale artifacts present on the top of the image which would render this type of segmentation virtually useless for automated analyses. Here the retraining took 4 min.

We are not sure what the reviewer is referring to when mentioning “ the new training data”. Image annotation is indeed a time-consuming process. We are now discussing this point in the new discussion. In this regard, it is important to note that the availability of pre-trained models can greatly accelerate the annotation process. In the case of StarDist, the model provided in the StarDist Fiji plugin was used to accelerate the annotation of our training dataset. Indeed, this model was used to generate masks that we refined. Then we used our annotated data to retrain the same StarDist model (in the ZeroCostDL4Mic notebook) to generate a new model that generates higher quality predictions (on our dataset). We estimate that it took around 4h to annotate our dataset. If no pretrained model is available, a similar approach can be used. A small training dataset (around 20 images) needs to be manually annotated (around 4h). This small dataset can then be used to train an initial model that can be used to accelerate the annotation of more training images. These models can then generate masks that can be refined by users (i.e., in Fiji) before being used for training to obtain a high-performance model. This bootstrapping approach can be made

iteratively, thereby increasing the network performance progressively while increasing the amount of training data available.

R3.18. - Colab has a policy to limit throughput of data, which the authors describe as unreliability to access GPUs, but indeed this is Google's policy which is against using their infrastructure too heavily. It is meant as a developer environment, not production scale. This should be made clearer for the readers.

We have now added several sentences in the main text to make the limitation of Colab (including this one) clearer.

R3.19. - L174-175 the computational resources provided by Google Colab is mostly a GPU, but is otherwise very limited in RAM and CPUs, which would be important to explore if this limits the throughput of the individual notebooks or not

We now provide a table indicating the maximal number of images that can be safely used to train a DL network using ZeroCostDL4Mic (Supp Table 5 and 6). As guessed by the reviewer, and already indicated above, the RAM is what limits the throughput of our notebooks.

Reviewers' Comments:

Reviewer #1:

Remarks to the Author:

Unfortunately the author's revision does an inadequate job of addressing the two serious concerns I had with their paper - placing their work in context of ongoing community efforts and highlighting the limitations of building a software ecosystem on colab. While the revisions attempt to address these issues, they fall short. Admittedly, my enthusiasm for this paper is somewhat diminished by the need for two rounds of revision to adequately address these points.

Correcting these issues are essential for this paper to be published because these issues are tied to the paper's value to the community. The work the author's present as it stands is not a major technical advance from either an algorithmic or software engineering point of view. No new deep learning methodology is presented. Superiority from a software engineering perspective is also not demonstrated. Concepts like model chaining have been demonstrated in other works and colab has been used for bioinformatics analyses in other contexts. These two metrics are the traditional measures by which computational work is judged, and on both axes there are serious shortcomings.

However, I (and other reviewers) feel the author's work has significant value as a community resource, as it provides an interactive survey of the field. This value was reflected in my prior review. The repository the authors have put together has utility to users curious about deep learning methods, as well as those with small, pilot datasets. Beyond that, this work falls short. I strongly believe the above issues should be addressed prior to publication.

The author's survey of the biological image analysis/deep learning space is inadequate. In its current version, it has three notable omissions that should be corrected.

-NucleAIzer: Peter Horvath's work deserves to be included in the introduction, as achieving the top score in the kaggle data science bowl is a non-trivial advance. His team also made their model available as a web service, which highlights the additional software engineering work necessary to make deep learning methods widely available

-DeepCell: This was highlighted in my previous review, and it is surprising that the authors chose not to include this work in the introduction or the discussion. The author's current ecosystem (Expository Jupyter notebooks with annotation in ImageJ) is quite similar to what was presented in the DeepCell team's first paper

(<https://journals.plos.org/ploscompbiol/article?id=10.1371/journal.pcbi.1005177>). The DeepCell team created the first web portal hosting deep learning models for cellular image analysis and their most recent work - which uses cloud computing to power a web portal as well as a FIJI plugin (<https://www.biorxiv.org/content/10.1101/505032v4>) - is relevant to this paper.

-CellProfiler: Even more troublesome than the prior omissions is omitting Anne Carpenter's contribution to this space. From organizing a data science bowl, to contributing training data, to making deep learning methods accessible, this space would not have evolved as fast as it has without her efforts. New readers deserve to be made aware of her work and her contributions.

These contributions should be added to the introduction as well as the discussion. Simply adding citations is not sufficient; the contributions of each of these works and how it differs from the authors should be explicitly stated in the introduction and discussion.

The discussion and framing of the limitations created by colab's are also inadequate. As the authors state, instances are short lived (12 hours), limited in memory (12 GB), and in GPUs (K80, P100, T4). Agreeing to these limitations is what provides free access to colab's resources. The cost of these limitations is substantial. As part of a group that actively develops deep learning models, I would not be able to use colab given the size of the training datasets we use. The instances do not have enough memory to load all of our images, and the GPU's memory limitation imposes severe constraints on batch and image sizes (increased batch size is often required for stable training). As

such, for our use case, it is not possible to get performant models through what colab (and hence ZeroCostDL4Mic offers). No where would I be able to infer this from the main text until the very end. The paper's language makes numerous references to how easy, user-friendly, and powerful the author's software is to use; the information telling me it's not possible to train models for use cases that have higher data requirements for free (as the paper title suggests) is buried in the supplement. The author's rosy assessment of their own work and the reality that ZeroCost4DL is limited to data poor use cases are in direct opposition. The authors claim that the notebooks can be easily set up to run on more powerful clusters, but this is not the case. Instances available through DeepNote are still limited in their memory (<https://docs.deepnote.com/environment/selecting-hardware> - even the pro machines only have 5 GB of memory and the product is still in beta; moreover there is a 750 hour limit on compute time). Running them on a GCE or AWS instance requires additional DevOps work (e.g. installation of CUDA, cuDNN, etc.) to create a suitable machine image, etc. The author's haven't presented evidence that this work has been done.

A similar issue exists for inference. Imaging experiments produce a lot of data; as an example I have two colleagues each with datasets that exceed 50,000 megapixel images; processing them in a timely fashion is an issue for both of them. According to the SI (and my own experience), this is too much for colab to handle. The only solution I can think of if they were to use ZeroCostDL4Mic would be to process in smaller batches spread - doing this in practice is not an easy or user friendly task. The author's claim that most life scientists have datasets limited in size is not likely to be true in my estimation. Even if it is, once they become aware of what deep learning can do, the demand is likely to increase by a large degree. ZeroCostDL4Mic is not equipped to handle this volume. There is also the larger issue of what happens when there is large demand on colab; excessive use will almost certainly lead to community wide throttling. Colab was never meant to provide the computational needs of the entire bio imaging community.

The use case I mention is not unique; any machine learning project that goes beyond the hobbyist phase with respect to data will reach the limit of what colab can offer fairly quickly. From a training perspective, the models derived from these hobbyist efforts might be useful, but they are limited in their capacity to generalize. A similar statement can be said for inference. Also missing from the paper's discussion is the financial cost of fixing these issues. The author's reason for not providing an estimate of the cost per model and cost per inference is not adequate - an order of magnitude estimate is more than sufficient to give the reader a sense of the numbers. A preemptible deep learning capable instance is ~1 dollar/hour on google cloud; based on this, each model costs ~1-10 dollars to train and each image costs ~0.0002 dollars for inference. These are not large numbers, and accurately frame what "ZeroCost" should be compared to. While the authors claim in their response that machines like V100s are expensive, the proper way to assess cost in cloud computing is with the marginal cost of computation (e.g. 70 cents per hour rather than 10,000 dollars for a V100). While 10 dollars creates a barrier for model R&D, for cases where training parameters are well defined it is quite affordable - even more so in light of the cost required to generate data. Vision models are a far cry from language models with respect to their development costs. Recent libraries like Tensorflow Cloud make the process of cloud training much simpler than it was a year ago.

In addition to these issues, there are serious shortcomings in the author's work from a software engineering perspective. Unit testing is absent for much of the code present in the Jupyter notebooks. Moreover, the software environment for each notebook is different. For example, with respect to the deep learning ecosystems, some notebooks are in Tensorflow, others are in PyTorch. This poses a substantial barrier to performing tasks like transfer learning and model chaining, as models cannot be shared between ecosystems (although outputs from one notebook can be fed as inputs to another). An additional weakness from the software perspective is annotation software; none is included in the authors work. Annotating using different software packages (e.g. FIJI) and bringing the data into the notebook has its own challenges and is another mark against the author's ecosystem being user friendly - having to manually shuttle back and

forth between two software packages introduces additional, unnecessary labor. This is an important point given that the authors do not include any pre-trained models in their software.

These issues are straight forward to address, and there are two possible paths.

-Bring the software engineering and performance up to a standard necessary to justify the paper's rosy language. This would include incorporating tests throughout all notebooks; bringing all of the notebooks into the same deep learning ecosystem; updating the cells of each notebook to represent the best practices (e.g. migrating to tensorflow 2, using tensorflow datasets rather than keras ImageDataGenerators, cleaning up unused, commented out code, etc.); implementing an annotation package - one capable of dot annotation, bounding box annotation, semantic segmentation annotation, and instance segmentation annotation; implementing a versioned, searchable database for models and training data; and implementing a robust pipeline for training models in a fee-for-service cloud (which should include mixed precision training to minimize costs to the end user).

-The alternative is that the authors adjust the framing and tone so that the paper accurately reflects their work. It needs to be clear in the paper's introduction that this work is meant as an introduction to novice users and is a survey of what the field has done. It also needs to be made clear from the beginning that this software is not meant to be a solution for model training and inference, and that more serious users should explore other pieces of work. Claims about the software's ease of use need to be walked back - particularly in light of the lack of annotation software. Discussion around "remaining challenges" needs to be reworded; the current version implies the author's work solves challenges around model training and inference. As mentioned previously it does not. Claims around iterative development of training data needs to be refactored; the works the authors cite would be categorized as using self/weak supervised learning rather than human-in-the-loop annotation. Information about how much these approaches reduce the number of labels necessary for performant models is not provided, and the point that quality annotations are needed to benchmark models is also missing. Moreover, it is not clear that the author's software would be able to support a human-in-the-loop effort - annotation software that facilitates easy editing of labels is not included, and colab again creates a limit to model and dataset size. Last, an order of magnitude estimate describing the marginal cost to train a typical model and the marginal cost to process one image should be included, so that readers can have an idea of what "ZeroCost" should be compared to.

Either path provides a reasonable route to publication, although admittedly the second would require substantially less work.

Other issues that merit attention:

-I personally find the exposition on the different use cases in the results section hard to follow. Given the target audience, I think most readers would not be interested in the more technical aspects of the analyses. I think a better organizational scheme would be to have a 1 page figure highlighting each of the different tasks and associated models with their outputs. The results section would be reduced to a vignette about each model, the problem it solves, and some insight with respect to utility/generalization (e.g. what could be used out of the box, and what needs careful attention to detail). Much of the discussion surrounding QC, metrics, etc can be moved to the supplement.

-Number of images (e.g. 1024x1024) would be an easier to interpret measure than GBs in the discussion around limitations. It would likely be worthwhile to separate the limitations for training and for inference.

-The UNet model looks like it only does binary classification, which is odd. Most UNets I have seen will do interior/edge/background, or have multiple classes for semantic segmentation.

-Figure 9b could use a cleaner example. The object detection error is confusing - I understand the desire to avoid cherry picking performance but in this case I think it would be fine.

Reviewer #2:

Remarks to the Author:

The authors have addressed the comments from my earlier review. I recommend the revised manuscript for publication.

Reviewer #3:

Remarks to the Author:

The authors have largely addressed my concerns appropriately. In the process they have however introduced an issue I think should be corrected.

The authors write:

"However, larger-scale (> 20 GB of data) and longer-term analysis pipelines may benefit from the investment in paid-for cloud-based platforms (like Paperspace ³⁴ , Amazon Web Services (AWS) ³⁵ Deepnote ³⁶) or local infrastructure, therefore tuning the resources to the needs of the specific in-house application. For these cases, ZeroCostDL4Mic is easily adjustable to run outside Google Colab (see Supplementary Note 6 for running ZeroCostDL4Mic notebooks within Deepnote and Supplementary Fig. 4)."

While this is a much better assessment of the limitations of the Colab implementations, the authors now suggested an alternative here, proposing to mitigate the issue. While I understand that the transition of Jupyter Notebooks is easier to this platform, it has to be made clear that using Deepnote is not resolving scalability by any means. In fact Deepnote is even far more restricted in terms of computational resources than Colab. Deepnote only provides 5GB of disk space and has currently no GPU implementation at all. It is specified when or how they plan to make GPUs available. This is contradictory to the authors original claims that GPUs are essential for the DL approaches and that the DL applications could not be performed locally because the necessary GPUs would be too expensive for the community.

It is fair to say, this mitigates concerns regarding the lifetime of the project once Google decides to stop their offerings with Colab but I don't think Deepnote can be suggested as a way to perform large scale analysis, large commercial providers do have very different infrastructure available with multi-GPUs, large memory nodes etc.

Dear reviewers and editorial team,

Thank you for your time evaluating our manuscript "ZeroCostDL4Mic: an open platform for Deep-Learning in Microscopy". We are grateful for the detailed comments provided by the three referees. In this revision, we have addressed all the comments raised, allowing us to improve our manuscript further. Below we detail our response and changes made to the manuscript based on your feedback:

Point by point reply to criticism raised

Reviewer #1 (Remarks to the Author):

Unfortunately the author's revision does an inadequate job of addressing the two serious concerns I had with their paper - placing their work in context of ongoing community efforts and highlighting the limitations of building a software ecosystem on colab. While the revisions attempt to address these issues, they fall short. Admittedly, my enthusiasm for this paper is somewhat diminished by the need for two rounds of revision to adequately address these points.

Correcting these issues are essential for this paper to be published because these issues are tied to the paper's value to the community. The work the author's present as it stands is not a major technical advance from either an algorithmic or software engineering point of view. No new deep learning methodology is presented. Superiority from a software engineering perspective is also not demonstrated. Concepts like model chaining have been demonstrated in other works and colab has been used for bioinformatics analyses in other contexts. These two metrics are the traditional measures by which computational work is judged, and on both axes there are serious shortcomings.

We thank the reviewer for raising these points. The criticism presented by the reviewer was already largely remarked in the previous revision. As such, we took these points into account during both revisions and made significant changes to the manuscript, which included an explicit acknowledgement of the published literature by other groups throughout the introduction and conclusions. Also, we highlight the previous use of Google Colab for Deep Learning (DL) by others and are perfectly aware that the software engineering involved in our work is new in itself. Therefore we explicitly avoided making such claims. And in fact, we were cautious and intentionally meant to implement the DL networks as faithfully as possible to the original authors' version. Again, the major novelty of ZeroCostDL4Mic comes from providing an entry-level DL platform purposely designed for non-expert researchers, featuring a uniform interface layout across a large number of different networks. We believe that the increasing number of citations of the preprint and the significant number of users identified in public forums showcases that ZeroCostDL4Mic is indeed a valuable resource for its intended life sciences academic community.

However, I (and other reviewers) feel the author's work has significant value as a community resource, as it provides an interactive survey of the field. This value was reflected in my prior review. The repository the authors have put together has utility to users curious about deep learning methods, as well as those with small, pilot datasets. Beyond that, this work falls short. I strongly believe the above issues should be addressed prior to publication.

We think there is an important point that might not have appeared clear enough to reviewer #1. We apologise if that was not the case in previous communications: our work stems from a community need, highlighted by collaborators, beta-testers and users. The ZeroCostDL4Mic platform's development is user-driven: we continuously implement DL networks and optimise the user experience based on community feedback. In this light, we believe ZeroCostDL4Mic has become a useful tool for academic research, of which there is already clear evidence. Beyond its educational value (ZeroCostDL4Mic is already significantly used as a learning tool), we feel strongly that it fills a need for medium-size training data projects as we highlighted clearly in our manuscript.

The author's survey of the biological image analysis/deep learning space is inadequate. In its current version, it has three notable omissions that should be corrected.

-NucleAIzzer: Peter Horvath's work deserves to be included in the introduction, as achieving the top score in the kaggle data science bowl is a non-trivial advance. His team also made their model available as a web service, which highlights the additional software engineering work necessary to make deep learning methods widely available

-DeepCell: This was highlighted in my previous review, and it is surprising that the authors chose not to include this work in the introduction or the discussion. The author's current ecosystem (Expository Jupyter notebooks with annotation in ImageJ) is quite similar to what was presented in the DeepCell team's first paper (<https://journals.plos.org/ploscompbiol/article?id=10.1371/journal.pcbi.1005177>). The DeepCell team created the first web portal hosting deep learning models for cellular image analysis and their most recent work - which uses cloud computing to power a web portal as well as a FIJI plugin (<https://www.biorxiv.org/content/10.1101/505032v4>) - is relevant to this paper.

-CellProfiler: Even more troublesome than the prior omissions is omitting Anne Carpenter's contribution to this space. From organizing a data science bowl, to contributing training data, to making deep learning methods accessible, this space would not have evolved as fast as it has without her efforts. New readers deserve to be made aware of her work and her contributions.

These contributions should be added to the introduction as well as the discussion. Simply adding citations is not sufficient; the contributions of each of these works and how it differs from the authors should be explicitly stated in the introduction and discussion.

We agree that the works the reviewer describes are indeed significant for the Deep Learning space and essential to provide context to the current manuscript. We want to point out that both NucleAIzzer and DeepCell papers were already cited in the document but maybe not discussed in sufficient detail. These tools are now explicitly mentioned in the introduction and discussion.

We have now also included the reference to CellProfiler and DeepCell Kiosk papers in the revised manuscript. We would also like to highlight that the manuscript describing DeepCell Kiosk from Bannon *et al.* was released after submitting our revised manuscript and could not have been included until now.

Nevertheless, it is important to note that the solutions highlighted by the reviewer occupy a specific space in the analysis of microscopy images using DL, focusing on nuclei and or cell segmentation. These solutions are not directly comparable to what we are presenting here, which is a general platform for DL agnostic of ecosystem and image analysis tasks. This is the main reason why we did not discuss them in the manuscript extensively. The fantastic work by Anne Carpenter is also mentioned in the text explicitly.

The discussion and framing of the limitations created by colab's are also inadequate. As the authors state, instances are short lived (12 hours), limited in memory (12 GB), and in GPUs (K80, P100, T4). Agreeing to these limitations is what provides free access to colab's resources. The cost of these limitations is substantial. As part of a group that actively develops deep learning models, I would not be able to use colab given the size of the training datasets we use. The instances do not have enough memory to load all of our images, and the GPU's memory limitation imposes severe constraints on batch and image sizes (increased batch size is often required for stable training). As such, for our use case, it is not possible to get performant models through what colab (and hence ZeroCostDL4Mic offers). No where would I be able to infer this from the main text until the very end. The paper's language makes numerous references to how easy, user-friendly, and powerful the author's software is to use; the information telling me it's not possible to train models for use cases that have higher data requirements for free (as the paper title suggests) is buried in the supplement. The author's rosy assessment of their own work and the reality that ZeroCost4DL is limited to data poor use cases are in direct opposition. The authors claim that the notebooks can be easily set up to run on more powerful clusters, but this is not the case. Instances available through DeepNote are still limited in their memory (<https://docs.deepnote.com/environment/selecting-hardware> - even the pro machines only have 5 GB of memory and the product is still in beta; moreover there is a 750 hour limit on compute time). Running them on a GCE or AWS instance requires additional DevOps work (e.g. installation of CUDA, cuDNN, etc.) to create a suitable machine image, etc. The author's haven't presented evidence that this work has been done.

As we clearly state, ZeroCostDL4Mic does not aim to be a universal solution for DL but rather an entry platform. It aims to help researchers with little expertise, to start exploring DL for their research quickly. We believe academics outside of well funded elite academic environments will benefit from it in particular. Importantly, we would like to highlight that all models used to generate the manuscript's data were trained and validated on the platform. The validation step unambiguously confirmed that the models generated achieved good performance, even within the restrictions of Google Colab.

In these settings, it is unclear to us why this demonstration of the use of the platform is not perceived as adequate.

We explicitly provide the scope of usability in the manuscript. Our results section provides a clear breakdown of the type of use cases that will benefit from the platform, which includes references to small- and medium projects (rather than big-data projects). We also highlight the platform as a prototyping and learning tool. The criticism that readers will not know this until reading the supplementary information is overstated, in our opinion. To be clear, we even quote that ZeroCostDL4Mic will not be practical for training datasets above 20GB.

Thus, we argue that our platform is a genuine training/inference platform, limited to a range of data scale and applications (which we practically and quantitatively define in a number of Supp. Tables). Of course, the resources provided for free by Google Colab are finite, which is also clearly acknowledged and quantified in the main part of the manuscript. But importantly, we show that ZeroCostDL4Mic is adaptable to run on more powerful cloud or local solutions to address this issue. Our more expert users in practice already do this. The project wiki pages now increasingly provide detailed guidance on how to use alternative solutions to Google Colab.

In the revised manuscript, we now further demonstrate how to adapt our notebooks in a few minutes to run on FloydHub (<https://www.floydhub.com/>), beyond the demonstration on Deepnote presented in our previous version of the manuscript. FloydHub is an online platform that provides higher computing performance than Google Colab for a small fee (See new supplementary note 6). It is important to note that platforms such as FloydHub charge users for each second used. In this context, it makes much more sense to get started with Google Colab and to move to a paid-for platform only when absolutely required.

A similar issue exists for inference. Imaging experiments produce a lot of data; as an example I have two colleagues each with datasets that exceed 50,000 megapixel images; processing them in a timely fashion is an issue for both of them. According to the SI (and my own experience), this is too much for colab to handle. The only solution I can think of if they were to use ZeroCostDL4Mic would be to process in smaller batches spread - doing this in practice is not an easy or user friendly task. The author's claim that most life scientists have datasets limited in size is not likely to be true in my estimation. Even if it is, once they become aware of what deep learning can do, the demand is likely to increase by a large degree. ZeroCostDL4Mic is not equipped to handle this volume. There is also the larger issue of what happens when there is large demand on colab; excessive use will almost certainly lead to community wide throttling. Colab was never meant to provide the computational needs of the entire bio imaging community.

As requested by the reviewer in the previous revision round, we carefully quantified the data limitations of Google Colab. This information is available in the SI. As with any computational resources, there are always limitations that restrict what is possible. In particular, the case mentioned by the reviewer, although not improbable, represents a rare scenario when considering the use cases we encountered so far. To guide users in this aspect, we highlight that for datasets larger than 20GB it is beneficial to use tools such as those that reviewer #1 mentions. As repeatedly stated in the manuscript, the ZeroCostDL4Mic project aims to be a

starting resource for curious life scientists who want to begin using DL. This is the target audience that we are aiming for. We further clarified this in the current manuscript.

Of note, the StarDist network can accommodate the use of Gigapixel images for inference. We will incorporate this option in our notebooks soon.

Also, many of the models trained within ZeroCostDL4Mic can be used on a local machine using Fiji. If the throughput provided by Colab is not sufficient (duration, storage space, etc..), inference can be easily performed locally.

The point raised by the reviewer that scientists will only produce more and more data is valid, a reason why a platform such as ZeroCostDL4Mic is invaluable to assess whether such amount of data is required in the first place. Additionally, our platform will enable scientists to learn about and gain confidence in using DL methods for free using Google Colab. Once this is achieved, they can quickly move to other paid-for cloud environments, such as Floydhub. This has always been the intention and is clearly stated in the manuscript on multiple occasions.

The use case I mention is not unique; any machine learning project that goes beyond the hobbyist phase with respect to data will reach the limit of what colab can offer fairly quickly. From a training perspective, the models derived from these hobbyist efforts might be useful, but they are limited in their capacity to generalize.

We fully agree with the reviewer, as clearly indicated in the manuscript. Specifically, thanks to the quantitative estimation of the breaking point of each notebook, we frame the limitations of the notebooks very clearly for the user and even give a conservative upper limit of the amount of dataset of ~20GB (representing ~10,000 images of 1024x1024 16 bits images) beyond which our platform starts being limiting. We agree that thoroughly testing a model's performance is necessary, and this is why we enabled a quality control step in all our notebooks.

We would also like to add that the use of the term "hobbyist" to define the type of user of the platform is demeaning, given that the platform is already being used and cited by professional academic researchers.

A similar statement can be said for inference. Also missing from the paper's discussion is the financial cost of fixing these issues. The author's reason for not providing an estimate of the cost per model and cost per inference is not adequate - an order of magnitude estimate is more than sufficient to give the reader a sense of the numbers. A preemptible deep learning capable instance is ~1 dollar/hour on google cloud; based on this, each model costs ~1-10 dollars to train and each image costs ~0.0002 dollars for inference. These are not large numbers, and accurately frame what "ZeroCost" should be compared to. While the authors claim in their response that machines like V100s are expensive, the proper way to assess cost in cloud computing is with the marginal cost of computation (e.g. 70 cents per hour rather than 10,000 dollars for a V100). While 10 dollars creates a barrier for model R&D, for cases where training parameters are well defined it is quite affordable - even more so in light of the cost required to

generate data. Vision models are a far cry from language models with respect to their development costs. Recent libraries like Tensorflow Cloud make the process of cloud training much simpler than it was a year ago.

We agree with the reviewer that the availability of inexpensive cloud computational resources will help the development of DL. We apologise for not providing explicit costing of what would be necessary to train and perform inference in the cloud on a paid-for system. We had not realised that the reviewer expected such costing to be present in the manuscript explicitly. We have now included the costing of FloydHub in the discussion.

While we agree that it is affordable to use cloud computing, this contrasts with the time and effort in skills development needed to set up such cloud computing. Currently, ZeroCostDL4Mic combined with Google Colab offers an excellent solution, simplifying DL access for academics interested in bioimage analysis. As now demonstrated, our notebook can be migrated to a different platform if more resources than what Colab provides are needed. There the cost will indeed be 10 USD per model, as we now demonstrate using FloydHub.

In addition to these issues, there are serious shortcomings in the author's work from a software engineering perspective. Unit testing is absent for much of the code present in the Jupyter notebooks. Moreover, the software environment for each notebook is different. For example, with respect to the deep learning ecosystems, some notebooks are in Tensorflow, others are in PyTorch. This poses a substantial barrier to performing tasks like transfer learning and model chaining, as models cannot be shared between ecosystems (although outputs from one notebook can be fed as inputs to another). An additional weakness from the software perspective is annotation software; none is included in the authors work. Annotating using different software packages (e.g. FIJI) and bringing the data into the notebook has its own challenges and is another mark against the author's ecosystem being user friendly - having to manually shuttle back and forth between two software packages introduces additional, unnecessary labor. This is an important point given that the authors do not include any pre-trained models in their software.

While we appreciate the usefulness of unit testing in large-scale project production, it is unclear how it would improve the user experience or functionality to end-users of the platform. Indeed we thoroughly validate our notebooks using our test datasets before each release. In our opinion, our strategy, while time-consuming, is appropriate to ensure the performance of the notebooks.

We however disagree on the point of ecosystems: we see that our platform is agnostic of any particular DL ecosystem as a strength since it allows us to implement such varied image analysis tasks as denoising and image-to-image translation. While we agree with the reviewer that network compatibility might provide benefits for transfer learning, we do not believe the lack of it to be a significant issue of the platform as we envision exceptionally few use cases where this would be useful. For instance, only a CARE pretrained network will likely help weight initialisation for a new CARE model, and this approach is, in fact, the easiest way to perform

transfer learning. This is valid for every network. Given that we maintain the networks as close as possible to the original publications, we believe the current option of feeding outputs from one network to another, made easy in the notebooks, is the best for users to combine multiple networks reliably. This approach also allows for the validation of each chained task to be performed independently.

The ZeroCostDL4Mic capacity to leverage high-quality tools such as Fiji and makesense.ai is a strength. Although re-implementing annotators within the platform is technically feasible, we see no point in duplicating a pre-existing and well-developed tool which can be interfaced with the platform in a straightforward manner. We have found that our user community appreciates using tools they trust and have been using for a long time to curate their own data.

Also, the possibility to use models trained in ZeroCostDL4Mic directly into Fiji is one of our most popular features, due to its convenience. To do so, we have teamed up with both the CSBdeep and DeepImageJ platform to ensure such compatibility.

Regarding the ecosystem, ZeroCostDL4Mic is a community effort, and we deeply engage with recent and ongoing developments. In this regard, ZeroCostDL4Mic has become a crucial part of the bioimage.io effort, which aims to create DL standards to be used across tools.

These issues are straight forward to address, and there are two possible paths.

-Bring the software engineering and performance up to a standard necessary to justify the paper's rosy language. This would include incorporating tests throughout all notebooks; bringing all of the notebooks into the same deep learning ecosystem; updating the cells of each notebook to represent the best practices (e.g. migrating to tensorflow 2, using tensorflow datasets rather than keras ImageDataGenerators, cleaning up unused, commented out code, etc.); implementing an annotation package - one capable of dot annotation, bounding box annotation, semantic segmentation annotation, and instance segmentation annotation; implementing a versioned, searchable database for models and training data; and implementing a robust pipeline for training models in a fee-for-service cloud (which should include mixed precision training to minimize costs to the end user).

What Rev. 1 suggests in this option would be a complete code change of all the individual networks that we have implemented. Not only would this require an unreasonable amount of time (we estimate that it may take one-two years for a full-time software engineer to achieve and thoroughly test this), but it is also in direct conflict with the spirit and the value of the work. Our framework constitutes a platform interfacing with the original authors' code implementation of the network. As Rev. 1 highlighted, we pull codes from the authors GitHub repositories. This has several practical and ethical purposes: we make the networks available as the authors provided and published them, necessary for reproducibility. We do not host code that is not originally ours and refer to each network's sources and contributors. Practically, the original authors deemed essential to develop their work in specific ecosystems, and our platform does not discriminate

between the underlying approaches. Instead, it makes networks available in a unified workflow for the users irrespective of the original ecosystem and original developer team. Finally, doing all this work will not change the user experience. It will not provide any new functional capacity to the target users and may still not satisfy the reviewer as the Google Colab limitations we clearly outline would remain.

Additionally to this, Rev. 1 expects us to build a fully working multi-modal Python-based annotation tool and a searchable database for models and training data. This request seems unreasonable to us, as each of these could command a separate publication altogether and is no easy feat, as, we would imagine, Rev.1 is well aware. But we agree that what Rev.1 describes would be fantastic tools for the community to have. This is why we collaborate with five other teams (Including DeepImageJ, Fiji, Imjoy, ilastik and the Human Protein Atlas project) to develop bioimage.io (<https://bioimage.io/#/>). Within bioimage.io we already provide our notebooks and training datasets in a searchable format. We will soon add pre-trained models (generated using our platform) that can be used across platforms. As you can see, what Rev.1 is asking of us is already being superseded by community efforts and ZeroCostDL4Mic is an integral part of it. Our tools are fully interfaced with the community's efforts for accessibility and transparency. We outlined all of this in our previous rebuttal, and it is very unclear why Rev.1 is asking this of us at this stage of the review process.

-The alternative is that the authors adjust the framing and tone so that the paper accurately reflects their work. It needs to be clear in the paper's introduction that this work is meant as an introduction to novice users and is a survey of what the field has done. It also needs to be made clear from the beginning that this software is not meant to be a solution for model training and inference, and that more serious users should explore other pieces of work. Claims about the software's ease of use need to be walked back - particularly in light of the lack of annotation software. Discussion around "remaining challenges" needs to be reworded; the current version implies the author's work solves challenges around model training and inference. As mentioned previously it does not. Claims around iterative development of training data needs to be refactored; the works the authors cite would be categorized as using self/weak supervised learning rather than human-in-the-loop annotation. Information about how much these approaches reduce the number of labels necessary for performant models is not provided, and the point that quality annotations are needed to benchmark models is also missing. Moreover, it is not clear that the author's software would be able to support a human-in-the-loop effort - annotation software that facilitates easy editing of labels is not included, and colab again creates a limit to model and dataset size. Last, an order of magnitude estimate describing the marginal cost to train a typical model and the marginal cost to process one image should be included, so that readers can have an idea of what "ZeroCost" should be compared to.

Either path provides a reasonable route to publication, although admittedly the second would require substantially less work.

In our opinion, the statement “ZeroCostDL4Mic is not meant to be a solution for model training and inference” is incorrect and unfounded, as proven by our numerous demonstration of performant models that can be used for legitimate microscopy studies and by the growing number of users within the community. Importantly, we would like to highlight that all of the models used to generate the manuscript’s data were trained and validated on the platform. The validation step unambiguously confirmed that the models provided good model performance, and all of this was performed within the restrictions of Google Colab. It is unclear why this demonstration of the use of the platform is not perceived as adequate.

Also, we explicitly state that ZeroCostDL4Mic is not meant to be a solution for extensive microscopy studies. We even quoted that for datasets above 20GB it would stop being practically useful. So we would argue that our platform is a genuine training/inference platform but is limited to a range of data scale and applications (which we practically and quantitatively define in several Supp. Tables). Of course, the resources provided for free by Google Colab are finite (which is also clearly acknowledged and quantified in the manuscript), but it is important to note that our platform can also be adapted to run on more powerful cloud solutions such as FloydHub, or on local machines. And this is already done by our more expert users in practice. We already guide users on how to set up local machines to run ZeroCostDL4Mic notebooks. In our GitHub Wiki page we also added further instructions on how to adapt our notebook so that they can be used on Deepnote and FloyHubs.

Other issues that merit attention:

-I personally find the exposition on the different use cases in the results section hard to follow. Given the target audience, I think most readers would not be interested in the more technical aspects of the analyses. I think a better organizational scheme would be to have a 1 page figure highlighting each of the different tasks and associated models with their outputs. The results section would be reduced to a vignette about each model, the problem it solves, and some insight with respect to utility/generalization (e.g. what could be used out of the box, and what needs careful attention to detail). Much of the discussion surrounding QC, metrics, etc can be moved to the supplement.

-Number of images (e.g. 1024x1024) would be an easier to interpret measure than GBs in the discussion around limitations. It would likely be worthwhile to separate the limitations for training and for inference.

-The UNet model looks like it only does binary classification, which is odd. Most UNets I have seen will do interior/edge/background, or have multiple classes for semantic segmentation.

-Figure 9b could use a cleaner example. The object detection error is confusing - I understand the desire to avoid cherry picking performance but in this case I think it would be fine.

We thank the reviewer for these additional comments. Concerning #1, the current structure of the manuscript was suggested by the other reviewers. We would be happy to make further

revisions at this stage, but we will let the editorial team decide what they think is the most appropriate action to take.

With regard to #2, we provide details on the sizes and number of images that the notebooks were tested on, as well as the different breaking points (limits of number of images that can be used for training for instance) within the SI tables. We have also included the number of images that 20GB roughly represents (~10,000 images) in the main text, as suggested.

Concerning #3, we have implemented the simplest version of U-Net which uses the same binary architecture as in the original publication as Ronneberger *et al.* We appreciate the comment about multiclass segmentation. We are indeed working on a notebook with a more extensive implementation of U-Net.

With regard to #4, we have indeed left this image in the current form for the mentioned concern against cherry-picking. We now provide a new and improved figure panel.

Reviewer #2 (Remarks to the Author):

The authors have addressed the comments from my earlier review. I recommend the revised manuscript for publication.

We thank the reviewer for accepting the revisions in the manuscript.

Reviewer #3 (Remarks to the Author):

The authors have largely addressed my concerns appropriately. In the process they have however introduced an issue I think should be corrected.

The authors write:

“However, larger-scale (> 20 GB of data) and longer-term analysis pipelines may benefit from the investment in paid-for cloud-based platforms (like Paperspace ³⁴ , Amazon Web Services (AWS) ³⁵ Deepnote ³⁶) or local infrastructure, therefore tuning the resources to the needs of the specific in-house application. For these cases, ZeroCostDL4Mic is easily adjustable to run outside Google Colab (see Supplementary Note 6 for running ZeroCostDL4Mic notebooks within Deepnote and Supplementary Fig. 4).”

While this is a much better assessment of the limitations of the Colab implementations, the authors now suggested an alternative here, proposing to mitigate the issue. While I understand that the transition of Jupyter Notebooks is easier to this platform, it has to be made clear that using Deepnote is not resolving scalability by any means. In fact Deepnote is even far more

restricted in terms of computational resources than Colab. Deepnote only provides 5GB of disk space and has currently no GPU implementation at all. It is specified when or how they plan to make GPUs available. This is contradictory to the authors original claims that GPUs are essential for the DL approaches and that the DL applications could not be performed locally because the necessary GPUs would be too expensive for the community.

It is fair to say, this mitigates concerns regarding the lifetime of the project once Google decides to stop their offerings with Colab but I don't think Deepnote can be suggested as a way to perform large scale analysis, large commercial providers do have very different infrastructure available with multi-GPUs, large memory nodes etc.

Firstly, we would like to thank the reviewer for the kind comments on the revised manuscript. Regarding the limitations of Deepnote as a representative alternative cloud-based platform, we agree with the reviewer that it has significant constraints even compared to Colab, certainly due to being in its infancy. The rationale behind providing this example was to address the reviewers valid original concerns about the adaptability of the notebooks should the Google Colab platform be retired by google's developers. To this point, we believe, in agreement with the reviewer, that we have demonstrated that ZeroCostDL4Mic can exist independently of Google Colab.

However, the reviewer makes a valid point that Deepnote is not a suitable option in its current form for users with larger Deep Learning projects or data requirements. Therefore, we have decided to implement a ZeroCostDL4Mic notebook within a more robust cloud computing resource, namely FloydHub to accommodate these additional concerns. This platform reliably offers scalable resources, including access to powerful GPUs and CPUs. Here, we showed that it was relatively straightforward to implement StarDist 2D training and inference and provide guidance to the users on setting this up in Supp. Note 6 and on our GitHub Wiki page.

Reviewers' Comments:

Reviewer #3:

Remarks to the Author:

After revision the authors have proposed that notebooks can now be utilized on a different platform, with the main purpose of scaling to large data. It is not particularly helpful considering that the target audience are scientists without programming knowledge, but they will now be left to convert the notebooks to Floydhub if they need to perform more extensive analysis.

I am also very concerned that the authors suggest this as a route to process large data but did not actually provide evidence that large-scale datasets can be processed with these notebooks on any of the platforms. The maximum image dimensions the authors describe which they process are at small stacks at 1024 by 1024 and it is well known that processing times for larger scale data do not scale linearly and image processing pipelines have to be adjusted or rewritten if used for images with large raster sizes. Other deep learning applications have shown image processing with individual images about several 100s of times as large as the images used here and terabytes of data being processed. Processing many small datasets is not the same as processing big data. In this context it is also not clear how the authors determined the breaking points in Supplementary Table 6. I have had trouble applying some of the provided colab notebooks on individual images with larger raster sizes.

This is in line with what Reviewer 1 describes as rosy language, where issues are neglected to an extent that the statements are incorrect, which still is a major issue with the current manuscript. As R1 pointed out in the last revision: Micra-Net is not a human in the loop approach. Also the authors have not actually implemented code to facilitate such an approach for their notebooks. However, they have not corrected their misstatements.

The authors also state that 'pre-trained models can lead to erroneous results when applied to a different dataset type, which, unfortunately, may lead to visually pleasing yet inaccurate results' The evidence here is restricted to one of their implementations where applying a pre-trained network generated inferior results. It may be an issue that the network architecture does not generalize very well. The authors cite two reviews (one from the authors themselves) to support their claim, which have very limited information in this direction and do not actually quantify such errors, or issues like overfitting and generalization.

On a related note, since the authors are concerned about model generalization, it seems that the authors in most (maybe all cases) don't use test data which is a different dataset from the training, but instead perform a simple training / test split, which means the data is typically nearly identical to the training data. For thorough evaluations of network generalization on biological imaging data this should be avoided, and instead separately generated test data be used.

Another issue arising after the revision is that the authors claim: 'Between the necessity to set up and access expensive and complex resources'. This however stands in clear contrast to 10 USD for training a new model. From my viewpoint the runtimes used for any of the datasets are all in this range and therefore do not impose a significant financial burden as compared to salaries, or costs of microscopy time. If those can actually be accomplished in Fiji on a local runtime (possibly even on a CPU), and such implementations are already existing in some cases why not point the readers to use these implementations, where they could run significantly longer without cost.